# Nonlinear circuits for naturalistic visual motion estimation

**James E Fitzgerald[1]\*, Damon A Clark[2,3]\***

[1]Center for Brain Science, Harvard University, Cambridge, United States; [2]Department of Molecular, Cellular and Developmental Biology, Yale University, New Haven, United States; [3]Department of Physics, Yale University, New Haven, United States

**Abstract** Many animals use visual signals to estimate motion. Canonical models suppose that animals estimate motion by cross-correlating pairs of spatiotemporally separated visual signals, but recent experiments indicate that humans and flies perceive motion from higher-order correlations that signify motion in natural environments. Here we show how biologically plausible processing motifs in neural circuits could be tuned to extract this information. We emphasize how known aspects of *Drosophila*'s visual circuitry could embody this tuning and predict fly behavior. We find that segregating motion signals into ON/OFF channels can enhance estimation accuracy by accounting for natural light/dark asymmetries. Furthermore, a diversity of inputs to motion detecting neurons can provide access to more complex higher-order correlations. Collectively, these results illustrate how non-canonical computations improve motion estimation with naturalistic inputs. This argues that the complexity of the fly's motion computations, implemented in its elaborate circuits, represents a valuable feature of its visual motion estimator.

**\*For correspondence:**
jamesfitzgerald@fas.harvard.edu
(JEF); damon.clark@yale.
edu (DAC)

**Competing interests:** The
authors declare that no
competing interests exist.

**Reviewing editor:** Matteo
Carandini, University College
London, United Kingdom

## Introduction

A major goal in neuroscience is to understand how the brain computes behaviorally relevant stimulus properties from streams of incoming sensory data (*Sejnowski et al., 1988*). Visual motion guides behaviors across the animal kingdom. To navigate, many vertebrates and invertebrates use visual data to estimate the velocity of full field motion, and they use that estimate to judge their motion with respect to their environment (*Sperry, 1950*; *Kalmus, 1964*; *Reichardt and Poggio, 1976*; *Orger et al., 2000*). Spatially localized motion perception (*Hubel and Wiesel, 1962*; *Barlow and Hill, 1963*; *Buchner, 1976*) is also important, as it can indicate the presence of predators or prey in the environment (*Reichardt et al., 1983*; *Gabbiani et al., 1999*; *Nordström et al., 2006*; *Zhang et al., 2012*), and spatial velocity gradients allow animals to judge relative distances (*Rogers and Graham, 1979*; *Srinivasan et al., 1991*; *Kral, 2003*; *Pick and Strauss, 2005*). In principle, different algorithms could be used to estimate different types of motion. However, data suggest that many animals compute local motion over an array of spatially localized elementary motion detectors, or EMDs, and then differentially pool those signals for use in different behaviors and neural operations (*Hubel and Wiesel, 1962*; *Barlow and Hill, 1963*; *Buchner, 1976*; *Britten et al., 1992*; *Gabbiani et al., 1999*; *Franz and Krapp, 2000*; *Rust et al., 2006*).

The Hassenstein-Reichardt correlator (HRC) was introduced nearly sixty years ago to model the EMD underlying the beetle's optomotor response (*Hassenstein and Reichardt, 1956*). It has since provided numerous insights into motion-guided behaviors across a variety of insect species. The HRC's successes are most striking in flies, where the HRC accurately predicts a wide variety of behavioral and neural responses (*Götz, 1968*; *Buchner, 1976*; *Egelhaaf and Borst, 1989*; *Haag et al., 2004*), and even adaptation to stimulus statistics (*Borst et al., 2005*). The HRC's importance also extends to primates and vertebrates, where the EMDs are often described in terms of the motion

**eLife digest** Many animals have evolved the ability to estimate the speed and direction of visual motion. They use these estimates to judge their own motion, so that they can navigate through an environment, and to judge how other animals are moving, which allows them to avoid predators or detect prey.

In the 1950s, a physicist and a biologist used measurements of beetle behavior in response to visual stimuli to develop a model for how the brain estimates motion. The model became known as the Hassenstein-Reichardt correlator (HRC). The HRC and related models accurately predict the behavioral and neural responses of insects and mammals to many types of motion stimuli.

However, there are visual stimuli that generate motion percepts in fruit flies (and humans) that cannot be accounted for by the HRC. Are these differences between real brains and the HRC simply imperfections in visual circuits, whose neurons cannot perform idealized mathematical operations, or are these deviations intentional, somehow improving motion estimates? In other words: are the observed deviations a bug or a feature of visual circuits?

To address this question, Fitzgerald and Clark evaluated how different models of motion detection performed when presented with natural scenes. Natural scenes are fundamentally different from most stimuli used in lab, since they contain a rich set of regularities that are not present in simple stimuli. Fitzgerald and Clark compared the ability of the HRC, along with new, more general models, to estimate the speed and direction at which images moved across a screen. This revealed that many models could out-perform the HRC by taking advantage of regularities in natural scenes. Those models that were tuned to perform well with natural scenes could also predict the paradoxical motion percepts that were not predicted by the HRC. This suggests that visual circuits may have evolved to perform well with natural inputs, and the paradoxical motion percepts represent a feature of the real circuit, rather than a bug.

Models that performed well with natural inputs treated light and dark visual information differently. This different treatment of light and dark is a property of most visual systems, but not of the HRC or related models. In the future, these models of motion processing may help us understand how biological details of the fruit fly's visual circuits help it to estimate motion.

energy model (*Adelson and Bergen, 1985*). In particular, although the HRC and motion energy models differ in terms of their intuition and neuronal bases, both models rely on the same mathematical fact about moving visual stimuli—motion causes pairs of spatially separated points to become correlated when a stimulus moves from one location towards the other (*Adelson and Bergen, 1985*; *van Santen and Sperling, 1985*). A simple algebraic identity shows that the HRC and motion energy models are computationally equivalent.

Research on *Drosophila*'s motion detection system has progressed quickly in recent years. With an influx of novel genetic, anatomical, and physiological tools, *Drosophila* researchers are able to perform experiments that have revealed an intricate neural circuit whose details were not anticipated. For example, separate pathways process the motion of light and dark moving edges (*Joesch et al., 2010*; *Clark et al., 2011*; *Maisak et al., 2013*), and different neurons within these pathways coordinate the motion response depending on the velocity of motion (*Ammer et al., 2015*). Furthermore, connectomic analysis has revealed that more spatial and temporal channels converge onto the fly's motion computing neurons than had been predicted by the HRC's two-input architecture (*Takemura et al., 2013*). Going forward, it is critical that the field discovers which of these circuit details are computationally relevant and which are not. Since many of these details go beyond the HRC's premise, we must consider alternate theories if we hope to understand how circuit details contribute to motion estimation.

A large body of theoretical and experimental work supports the hypothesis that visual systems are tailored for functionality in the animal's natural behavioral context (*Simoncelli and Olshausen, 2001*). For example, photoreceptors adapt effectively across the ecological range of light levels (*Juusola and Hardie, 2001*), the excess number of OFF vs ON retinal ganglion cells matches the excess information of dark vs light contrasts in natural images (*Ratliff et al., 2010*), and several learning algorithms predict receptive fields similar to early cortical neurons when applied to natural images

(*Olshausen and Field, 1996*; *Bell and Sejnowski, 1997*). These examples are special cases of the general hypothesis that the early visual system provides an efficient code for the natural visual environment, and recent research suggests that efficient coding accounts for certain aspects of higher-level coding and perception as well (*Tkačik et al., 2010*; *Hermundstad et al., 2014*; *Yu et al., 2015*).

Several recent studies have established connections between the biological algorithms used for visual motion estimation and the statistical demands of naturalistic motion estimation. Natural stimuli are intricately structured and light–dark asymmetric (*Geisler, 2008*), and a variety of low and high order correlations characterize motion in such an environment. Although the HRC and motion energy models only respond to pairwise correlations in their inputs (*Adelson and Bergen, 1985*; *van Santen and Sperling, 1985*), the Bayes optimal visual motion estimator also incorporates a variety of higher-order correlations of both even and odd order (*Potters and Bialek, 1994*; *Fitzgerald et al., 2011*). Accordingly, certain visual stimuli that contain only higher-order correlations induce motion percepts in both vertebrates and insects (*Chubb and Sperling, 1988*; *Quenzer and Zanker, 1991*; *Zanker, 1993*; *Orger et al., 2000*; *Hu and Victor, 2010*; *Clark et al., 2014*), and theoretical work shows that the correlations that characterize these stimuli can also improve motion estimation in natural environments (*Clark et al., 2014*). This demonstrates that neither the HRC nor the motion energy model can account for the totality of experimentally observed motion percepts and suggests that departures from these canonical models might improve motion estimation accuracy. Relatively little is known about the neural basis of these higher-order motion percepts, although several studies have suggested intriguing commonalities across insect and primate species (*Clark et al., 2014*; *Nitzany et al., 2014*).

Here we investigate whether the computational demands imposed by accurate motion estimation in natural environments can illuminate the unexpected details of *Drosophila*'s motion estimation circuit or account for non-Reichardtian motion perception in flies. We study a sequence of five computational models, each of which considers a conceptually new aspect of the motion estimation problem. Since each model succeeds in improving estimation accuracy, these results provide a range of nonlinear circuit mechanisms that flies and other animals might incorporate into their motion estimators. We describe how observed elements of *Drosophila*'s motion estimation circuitry could support such computations (*Table 1*). Importantly, four of the five models also predict the signs and approximate magnitudes of known non-Reichardtian motion percepts in flies. Since the models were tuned exclusively for estimation accuracy, these results support the view that non-Reichardtian motion percepts probe ethologically relevant aspects of biological motion estimators. More generally, our results posit normative interpretations for some unexpected aspects of the fly's motion estimation circuit and behavior and suggest that non-Reichardtian aspects of fly circuitry and behavior might be closely linked through the statistics of natural scenes.

## Results

### Flies incorporate motion signals that the HRC neglects

The HRC is the dominant model of motion computation in flies and other insects. In this paper we describe several generalizations of the HRC, but it is helpful to first review this canonical model. The HRC comprises three stages of processing. First, two different temporal filters (here, a low-pass filter ($f(t)$) and a high-pass filter ($g(t)$)) are applied to each of two spatially filtered visual input streams (*Figure 1A*, 'Materials and methods'). These four filtered signals are then paired and multiplied (*Figure 1A*). Finally, the HRC takes the difference between the two multiplied signals to obtain a mirror anti-symmetric motion estimator (*Figure 1A*). Because the HRC combines its two input channels via a multiplication operation, the average output of the HRC depends only on 2-point correlations in the visual stimulus. We thus refer to the HRC as a 2-point correlator, and we will return to this mathematical characterization of the HRC repeatedly throughout this work.

No motion estimator is perfect for every stimulus, and this paper explores the hypothesis that evolution has tuned *Drosophila*'s motion estimator for visual experiences that are likely to result from ordinary behavior in natural environments (Appendix 1). We assessed the accuracy of the HRC and other motion estimators by approximating naturalistic motion as the rigid translation of natural images (*Clark et al., 2014*), with a velocity distribution that mimicked *Drosophila*'s natural behavior (*Figure 1B*, 'Materials and methods') (*Katsov and Clandinin, 2008*). We spatiotemporally filtered the input signals to simulate the responses of two neighboring photoreceptors ('Materials and methods').

**Table 1.** The different models used in this paper, experimental results that support each model, and references for those results

| Model | Supporting evidence |
| --- | --- |
| Front-end nonlinearity | • Photoreceptors show nonlinear responses to contrast changes (*Laughlin, 1989*; *Juusola and Hardie, 2001*; *Juusola and Hardie, 2001*; *van Hateren and Snippe, 2006*) |
| | • Some neurons in the early visual system have nonlinear responses that make their output signals nearly uniform (*Laughlin, 1981*) |
| Weighted 4-quadrant model | • Visual processing is divided early into ON and OFF channels (*Joesch et al., 2010*; *Clark et al., 2011*; *Behnia et al., 2014*; *Meier et al., 2014*; *Strother et al., 2014*) |
| | • The two output channels (T4/T5) are sensitive to light and dark edges (*Maisak et al., 2013*), but their inputs are incompletely rectified (*Behnia et al., 2014*) |
| | • Stimuli targeting the four quadrants are differentially represented in neural substrates (*Clark et al., 2011*; *Joesch et al., 2013*) |
| Non-multiplicative nonlinearity | • Pure multiplication is not a trivial neural operation (*Koch, 2004*) |
| | • Inputs to T4/T5 are nonlinearly transformed (*Behnia et al., 2014*), which also contributes to the biologically implemented non-multiplicative nonlinearity |
| Unrestricted nonlinearity | • The direction-selective neurons T4 receive inputs from more than two types of neurons (*Takemura et al., 2013*; *Ammer et al., 2015*) |
| | • T4 receives inputs from both its major input channels at overlapping points in space (*Takemura et al., 2013*) |
| Extra input nonlinearity | • The direction-selective neuron T4 receives inputs from more than two discrete retinotopic locations (*Takemura et al., 2013*) |

We quantified the performance of each model as the mean squared error between the input velocity and model output. However, we report each model's accuracy as the correlation coefficient between its output and the true velocity (*Figure 1C*), an intuitive metric that is equivalent to the mean squared error for correctly scaled model outputs ('Materials and methods'). In isolation, the local HRC was weakly correlated with the velocity of motion (*Figure 1D*). Although the HRC's performance can be improved by averaging over space and time (*Dror et al., 2001*; *Clark et al., 2014*), this study explores how alternate nonlinear processing can improve motion estimation accuracy without sacrificing spatial or temporal resolution (*Clark et al., 2014*).

Researchers can probe a fly's motion estimate by measuring its behavioral optomotor turning response (*Hassenstein and Reichardt, 1956*; *Götz and Wenking, 1973*; *Buchner, 1976*; *Reichardt and Poggio, 1976*). We previously measured optomotor responses from flies walking on a spherical treadmill by recording their turning responses to various visual stimuli (*Figure 1E*) (*Clark et al., 2014*). We emphasized binary stimuli called gliders (*Hu and Victor, 2010*) (*Figure 1F*), which enforce spatiotemporal correlations to interrogate the fly's motion estimation algorithm. For example, 2-point gliders contain only 2-point correlations (*first two stimuli*, *Figure 1F*). *Drosophila* turned in response to these stimuli (*Clark et al., 2014*) (*black bars*, *left*, *Figure 1G*), and the HRC correctly predicted that flies would respond to both positive and negative 2-point correlations (*gray bars, left*, *Figure 1G*). On the other hand, 3-point gliders contain 3-point correlations without 2-point correlations (*last four stimuli*, *Figure 1F*). These stimuli generated motion responses in flies (*black bars*, *right*, *Figure 1G*) that the HRC could not explain (*gray bars*, *right*, *Figure 1G*). Thus, behavioral responses to glider stimuli show that the HRC is an incomplete description of fly motion estimation and provide a useful benchmark for evaluating alternate models.

In this study, we tune our models to optimize motion estimation accuracy, rather than to fit the behavioral data, for two main reasons. First, we want to explore the hypothesis that *Drosophila*'s glider responses follow from performance optimization within biologically plausible circuit architectures. Second, we seek models that will generalize well across visual stimuli, and the measured glider responses under-constrain possible motion estimation models. It's useful to illustrate

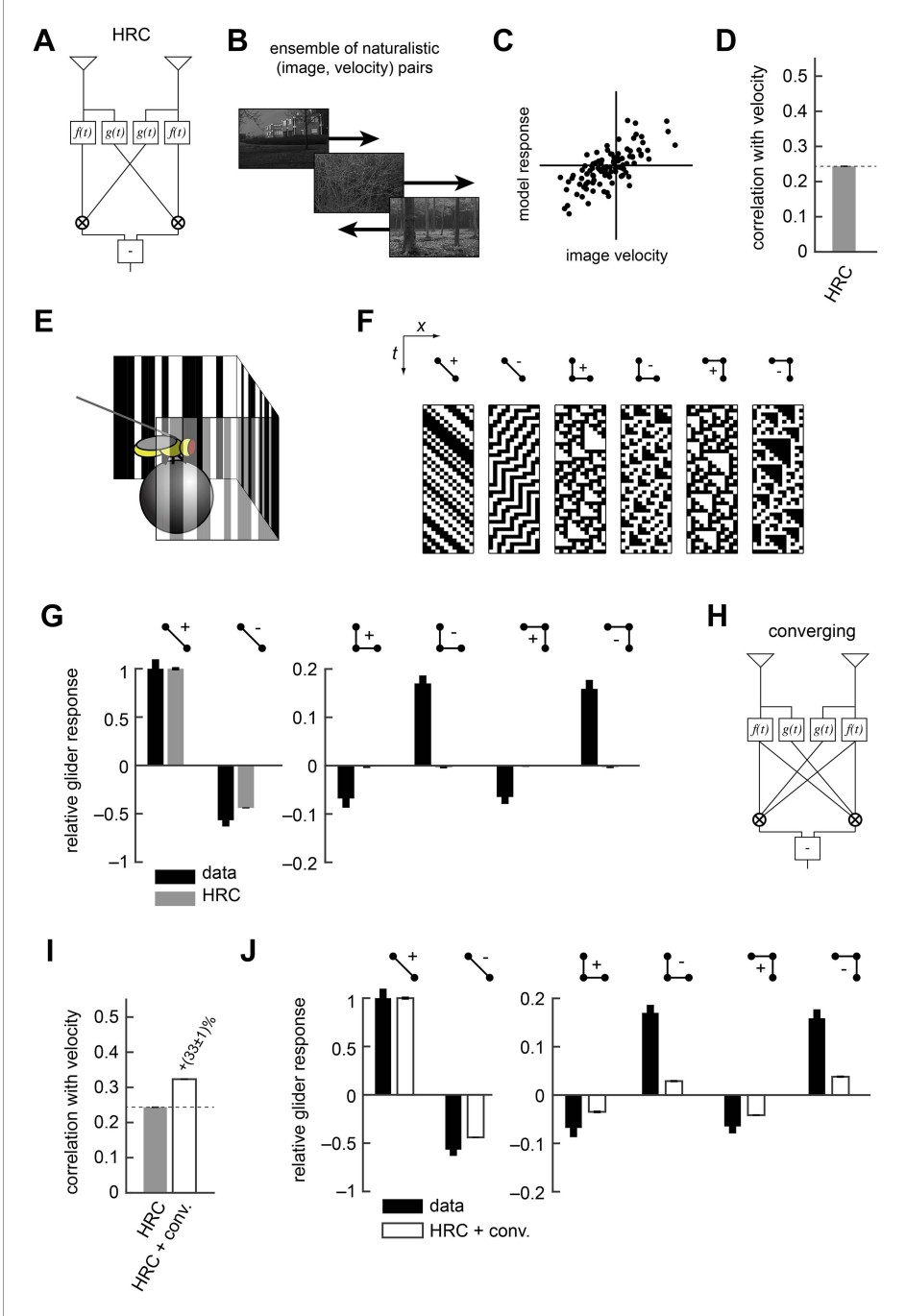

**Figure 1.** The Hassenstein-Reichardt correlator (HRC) model is an incomplete description of *Drosophila*'s motion estimator. (**A**) Diagram of the HRC model. (**B**) We assessed motion estimation performance across an ensemble of naturalistic motions, each of which consisted of a natural image (***van Hateren and van der Schaaf, 1998***) and a velocity chosen from a normal distribution. (**C**) We quantified model accuracy by comparing the model response to the true velocity using the mean squared error. (**D**) We summarized the error with the correlation coefficient between the model output and the true velocity. (**E**) In previous work (***Clark et al., 2014***), we used a panoramic display and spherical treadmill to measure the rotational responses of *Drosophila* to visual stimuli. (**F**) We presented flies with binary stimuli called gliders (***Hu and Victor, 2010***), which imposed specific 2-point and 3-point correlations (***Clark et al., 2014***). (**G**) Flies turned in response to 3-point glider stimuli, but these responses cannot be predicted by the standard HRC. (**H**) Diagram of the converging 3-point correlator, which is designed to detect higher-order motion signals like those found in 3-point glider stimuli. (**I**) Adding the converging 3-point correlator to the HRC improved

*Figure 1. continued on next page*

*Figure 1. Continued*

motion estimation performance with naturalistic inputs. We optimized weighting coefficients to minimize the mean squared error over the ensemble of naturalistic motions and used cross-validation to protect against over-fitting. (**J**) This model predicted that *Drosophila* would weakly turn in response to 3-point glider stimuli.

our procedure with a simple example. The HRC does not account for 3-point glider responses because it is insensitive to 3-point correlations. Nevertheless, 3-point correlations are present in natural stimuli (*Clark et al., 2014*; *Nitzany and Victor, 2014*), and their use might facilitate accurate motion estimation. We can explore this hypothesis by summing the HRC with a motion estimator designed to respond specifically to 3-point correlations. For instance, the mirror anti-symmetric 'converging' 3-point correlator multiplies one high-pass filtered signal with two low-pass filtered signals (*Figure 1H*) and mimics the converging structure present in certain glider stimuli (*last two stimuli*, *Figure 1F*). We tune the model for motion estimation accuracy by choosing the weights of the HRC and the converging 3-point correlator to minimize the mean squared error ('Materials and methods'). The resulting model is more accurate than the HRC (*Figure 1I*) and it predicts that flies should respond to glider stimuli in the observed directions (*Figure 1J*, 'Materials and methods'). Nevertheless, this simple model underestimates 3-point turning magnitudes (*Figure 1J*), indicating a discrepancy between the fly's motion estimator and this performance-optimized model.

In this study, we apply this same basic model building procedure to a series of increasingly general model architectures. There are four benefits to this approach. First, each model incorporates a type of computation that was neglected by earlier models. Thus, we can compare model accuracies to quantify how important various computations are for naturalistic motion estimation. Second, each model has a distinct biological interpretation in terms of *Drosophila*'s motion estimation circuit (*Table 1*). This allows us to enumerate many directions for future experimental and computational research. Third, this set of models reveals several distinct principles of accurate naturalistic motion estimators, yet no single model illustrates every principle. Finally, by comparing the glider predictions of each model to behavioral data, we can gain insight into which principles underlie *Drosophila*'s known glider responses.

## Nonlinear preprocessing of HRC inputs improves estimation but poorly predicts responses to gliders

The HRC correlates pairs of photoreceptor signals (*Figure 1A*). We previously assumed that each photoreceptor's response was generated from incoming contrast signals through linear spatiotemporal filtering. However, real photoreceptors are linear only over a limited range of inputs (*Laughlin, 1981*; *Juusola and Hardie, 2001*) (*Table 1*). Our first model thus modifies the HRC by allowing the photoreceptor responses to become nonlinear (*Figure 2A*). More specifically, we consider models in which a static nonlinearity transforms the filtered contrast signals before a standard HRC is applied to the two input streams (*Figure 2A*, 'Materials and methods'). Since the nonlinearity occurs before the HRC, we refer to this model as the *front-end nonlinearity* model. By nonlinearly transforming the contrast signals, the front-end nonlinearity model is able to reshape natural sensory statistics. In particular, linear photoreceptor signals inherit complex non-Gaussian statistics from their natural inputs (*Figure 2B*), but front-end nonlinearities (*Figure 2C*) can produce transformed signals with alternate statistics (*Figure 2D*, 'Materials and methods'). Thus, optimal front-end nonlinearity models should reshape natural statistics into those that best suit the HRC. Previous studies have already demonstrated example front-end nonlinearity models that improve naturalistic motion processing by the HRC (*Dror et al., 2001*; *Brinkworth and O'Carroll, 2009*). Here we provide new theoretical insight into these improvements and their consequences for glider responses.

Although the statistics of natural images are complicated, the mean squared error between the HRC's output and the velocity of motion depends only on a few statistical quantities. Since the HRC is a 2-point correlator, the mean velocity signal decoded by an HRC is determined by the second-order statistics of the image ensemble (*Dror et al., 2001*). The variance of the motion signal comes from the square of a quadratic signal, and thus the noise statistics of the HRC depend on the fourth-order statistics of the image ensemble (Appendix 2). If the image ensemble is spatially uncorrelated, the situation simplifies further and

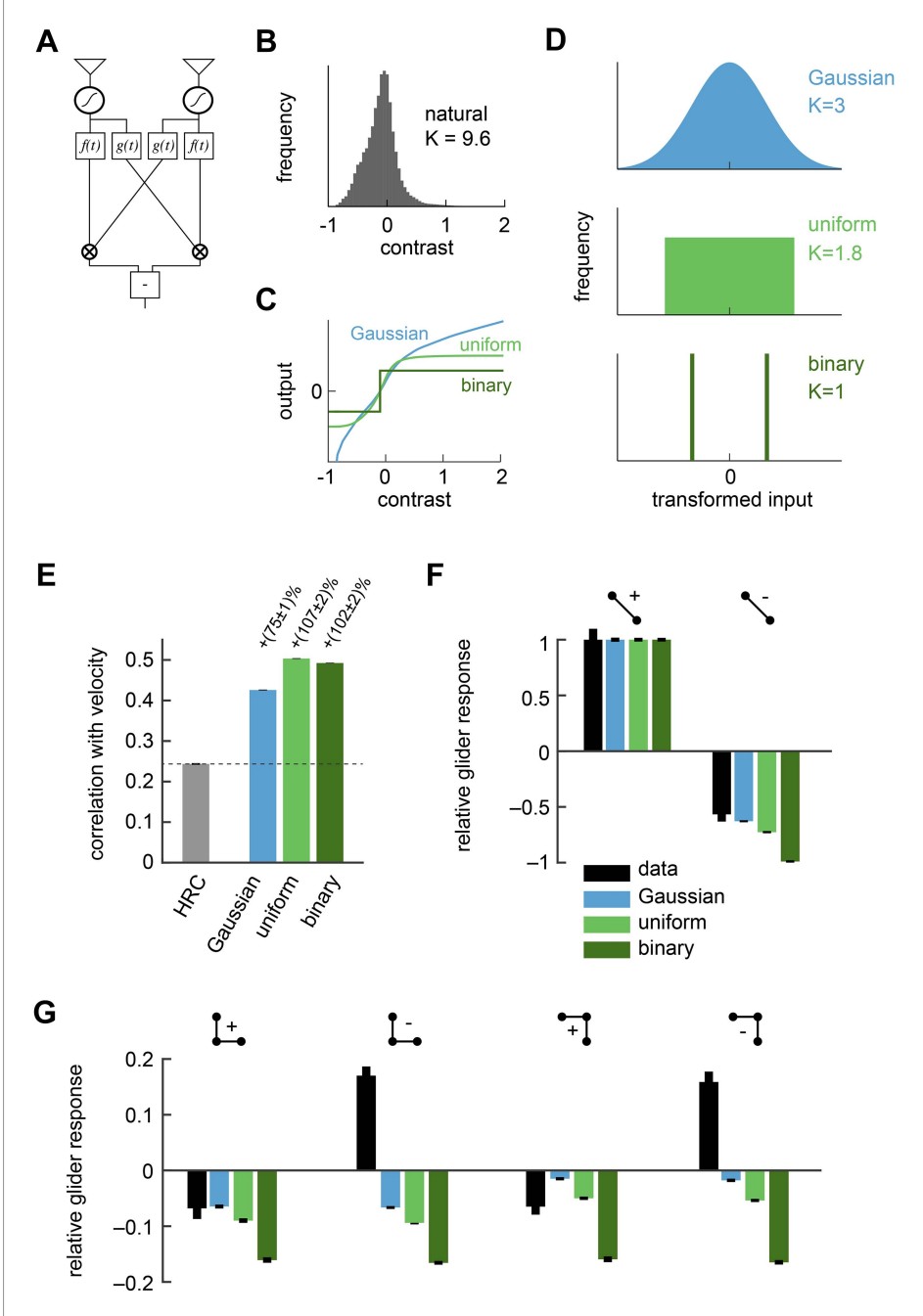

**Figure 2**. Front-end nonlinearities improved naturalistic motion estimation but did not reproduce the psychophysical results. (**A**) Diagram of the front-end nonlinearity model. The nonlinearity occurs after the spatiotemporal filtering of photoreceptors but before the temporal filtering of the HRC. (**B**) The distribution of contrast signals after photoreceptor filtering had a kurtosis of 9.6. The kurtosis of unfiltered pixels in the image database was 7.8. (**C**) Three different nonlinearities that transformed this input distribution into a Gaussian distribution, a uniform distribution, and a binary distribution. (**D**) After these transformations, the kurtosis of the contrast signal was reduced to 3, 1.8, and 1, respectively. (**E**) Each front-end nonlinearity model improved the HRC's estimation accuracy, and uniform output signals worked best. (**F**, **G**) The front-end nonlinearity models reproduced the sign of the negative 2-point glider psychophysical responses but did not reproduce the pattern of psychophysical responses to 3-point gliders.

*Figure 2. continued on next page*

*Figure 2. Continued*

The following figure supplement is available for figure 2:

**Figure supplement 1**. Front-end nonlinearities modify the correlations present in natural scenes.

the correlation between the estimated and true image velocity is determined entirely by the standardized fourth central moment of the input streams, a quantity known as kurtosis (Appendix 3). A larger kurtosis results in a larger error in the motion estimate. Note that some authors use 'kurtosis' to refer to the 'excess kurtosis', which shifts kurtosis values such that the Gaussian distribution has zero excess kurtosis. This shift is not relevant for our purposes. Because large positive contrasts are relatively probable, naturalistic inputs are highly kurtotic (kurtosis = 9.6 for the spatiotemporal filtering in our simulations) and are thus expected to hinder HRC performance (*Figure 2B*).

The Gaussian, uniform, and symmetric Bernoulli distributions have much lower kurtosis values (kurtosis = 3.0, 1.8, 1.0, respectively, *Figure 2D*). In fact, the symmetric Bernoulli distribution has the lowest kurtosis of any probability distribution (*DeCarlo, 1997*). When we transformed the HRC's inputs to have these statistics ('Materials and methods'), we found that each nonlinearity substantially improved the accuracy of the HRC (*Figure 2E*). The *contrast equalizing* nonlinearity, which produces uniform outputs, performed best and also plays a prominent role in efficient coding theory (*Laughlin, 1981*). It is interesting that contrast equalization improved the accuracy of the HRC more than binarization (*Figure 2E*), even though it produced outputs with greater kurtosis. The reason for this is that natural images are spatially correlated, and the accuracy of the HRC over a general image ensemble depends on the ensemble's spatial correlation structure (Appendix 2). Binarization attenuated spatial correlations more strongly than contrast equalization over the natural image ensemble (*Figure 2—figure supplement 1*), and spatial correlations can enhance the performance of the HRC (Appendices 4, 5). Designing a nonlinearity that optimally sculpts the correlation structure of natural images is not simple and goes beyond the scope of this study.

Each front-end nonlinearity model is sensitive to a variety of higher-order correlations (Appendix 6). We thus tested whether accurate front-end nonlinearity models would predict *Drosophila*'s glider response pattern. However, each front-end nonlinearity model performed poorly at this task (*Figure 2F,G*). None of the three models predicted that *Drosophila* would invert its response to positive and negative 3-point gliders (*Figure 2G*), even though they predicted that the 3-point glider responses would be nonzero. The simplest explanation for this observation is that the front-end nonlinearity models responded to fourth-order correlations that are common to the stimuli, rather than the third-order correlations that defined the glider stimuli and primarily drove the experimental response (*Clark et al., 2014*). Mechanistically, this result follows from the fact that the nonlinearities that reduced kurtosis (*Figure 2C*) were not strongly asymmetric around zero contrast (Appendix 6). The binarizing front-end nonlinearity model also failed to predict that *Drosophila* would respond less to negative 2-point glider stimuli than positive 2-point glider stimuli (*Figure 2F*). Since this effect was correctly predicted by the standard HRC (*Figure 1G*), this observation shows that accurate front-end nonlinearity models can distort the processing of 2-point correlations. Although the front-end nonlinearity model did not explain the phenomenon of fly glider perception, future work should investigate whether its merits make it functionally relevant for motion processing in other contexts or species.

## Separating ON and OFF signals improves motion estimation and predicts responses to gliders

Instead of a front-end nonlinearity, *Drosophila* could use an alternative non-Reichardtian motion estimation strategy that reflects natural sensory statistics, without necessarily requiring nonlinear preprocessing. Previous computational analyses show that motion estimation strategies that distinguish light and dark information can enhance motion processing with natural inputs (*Fitzgerald et al., 2011*; *Clark et al., 2014*; *Nitzany and Victor, 2014*), and recent experiments indicate that flies use separate channels to process the motion of light and dark edges (*Joesch et al., 2010*; *Clark et al., 2011*; *Behnia et al., 2014*; *Clark et al., 2014*; *Meier et al., 2014*; *Strother et al., 2014*) (*Table 1*). Our next model explores the hypothesis that *Drosophila* segregates ON and OFF signals in order to

facilitate naturalistic motion estimation (*Clark et al., 2014*) (*Figure 3A*, 'Materials and methods'). There are four ways to pair the ON and OFF components of the two filtered signals that enter the HRC's multiplier. For example, one possibility is to pair the ON component of the low-pass filtered signals with the OFF component of the high-pass filtered signal. Since each pairing restricts the HRC's multiplier to a single quadrant of the Cartesian plane, we refer to these four signals as HRC-quadrants. If the quadrants are summed with equal weights, then this model is mathematically identical to the HRC (*Hassenstein and Reichardt, 1956*; *Clark et al., 2011*). Unequal weighting coefficients enable the motion estimator to prioritize some quadrants over others, and here we select quadrant weightings that minimize the mean squared error between the model output and velocity (*Figure 3B*, 'Materials and methods'). More generally, we refer to any model that linearly combines the four HRC-quadrants as a *weighted 4-quadrant* model. The precise manner in which the four HRC-quadrants might map onto circuitry remains unclear; we do not suggest there exists separate circuitry for each quadrant. For instance, studies have identified only two motion-processing channels in the *Drosophila* brain, which might suggest that the fly only uses a subset of the quadrants (*Eichner et al., 2011*; *Joesch et al., 2013*; *Maisak et al., 2013*). On the other hand, each channel appears imperfectly selective for light vs dark signals (*Behnia et al., 2014*), which in principle enables these two channels to access all four quadrants (*Table 1*).

We began by examining how well individual quadrants predicted the velocity of motion. The four quadrants provided motion signals of strikingly different quality (*first four red bars*, *Figure 3C*). The most accurate quadrant correlated negative low-pass filtered signals with negative high-pass filtered signals ((− −) *bar*, *Figure 3C*). This isolated quadrant already outperformed the full HRC. The quadrant that correlated negative low-pass filtered signals with positive high-pass filtered signals also performed relatively well ((− +) *bar*, *Figure 3C*), whereas the quadrants that involved positive low-pass filtered signals performed poorly ((+ +) and (+ −) *bars*, *Figure 3C*). This shows that negative signals emanating from the low-pass filter better facilitate motion estimation, and the HRC's uniform weighting of all four quadrants is computationally detrimental.

We next considered all subsets of two, three, or four quadrants. The best subsets for each number of predictors were nested, and the quadrants were incorporated in the order (i) (− −); (ii) (− +); (iii) (+ +); (iv) (+ −). Although all four quadrants enhanced the accuracy of the weighted 4-quadrant model, the benefit of each added quadrant decreased with the number of quadrants (*Figure 3C*). It is possible to reparameterize the weighted 4-quadrant model in a form that isolates the contributions of various higher-order correlations to the model's accuracy (Appendix 7). Interestingly, this parameterization showed that nearly all the accuracy of the weighted 4-quadrant model can be obtained by supplementing the HRC with a set of odd-ordered correlations that account for the asymmetry between positive and negative low-pass filtered signals (*Figure 3—figure supplement 1*, Appendix 8). Principal component analysis (PCA) did not reveal this simple interpretation of the model's computation (Appendix 9).

The performance-optimized weighted 4-quadrant model also offered an interesting interpretation of *Drosophila*'s glider response pattern. First note that the model preserved the HRC's response pattern to 2-point glider stimuli (compare *left* subpanels of *Figure 3D* and *Figure 1G*). More interestingly, the model predicted behavioral responses to 3-point glider stimuli that matched the experimentally observed turning directions, and even the response magnitudes were similar between the model and the data (*right*, *Figure 3D*). Nevertheless, the model's predictions were imperfect. The primary qualitative discrepancy was that the model failed to predict that positive 3-point glider stimuli would generate smaller turning responses than negative 3-point glider stimuli. The simplest interpretation for this experimental result is that flies might incorporate both 3-point correlations and 4-point correlations into their motion estimation strategy. In particular, since the positive and negative 3-point glider stimuli have inverted 3-point correlations and matched 4-point correlations, third-order and fourth-order correlations would have the same sign for one parity and opposite signs for the other parity. This observation makes it easier to understand the glider predictions of the weighted 4-quadrant model. The optimized model does a good job accounting for the direction and approximate magnitude of the glider responses because it draws heavily on second-order and odd-order correlations, but it fails to predict the 3-point glider magnitude asymmetry because it finds little added utility in higher-order even correlations (*Figure 3—figure supplement 1*, Appendix 8). This failure stems from architectural limitations in the weighted 4-quadrant model (*Figure 3—figure supplement 2*), so it is important to consider alternate model classes.

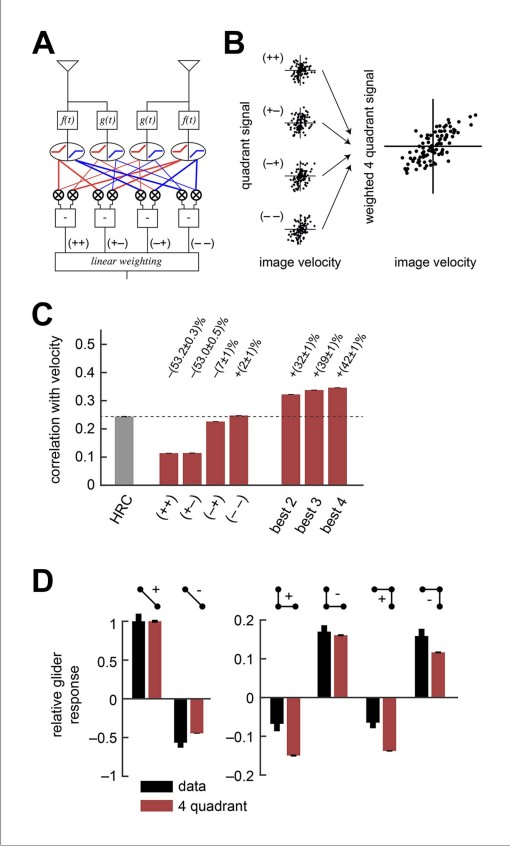

**Figure 3**. The weighted 4-quadrant model improved estimation performance and reproduced the directionality of psychophysical results. (**A**) Diagram of the weighted 4-quadrant model. Similar to ON/OFF processing in the visual system, the weighted 4-quadrant model splits the four differentially filtered signals into positive and negative components. As in the HRC, these component signals are paired, multiplied, and subtracted to produce four mirror anti-symmetric signals. We refer to these signals as HRC-quadrants. The model output is a weighted sum of the quadrant signals. We identify quadrants by whether they respond to the positive or negative components of each filtered signal and denote the four quadrants as (+ +), (+ −), (− +), and (− −). In this notation, the first index refers to the sign of the low-pass filtered signal (emanating from $f(t)$), and the second refers to the high-pass filtered signal (emanating from $g(t)$). (**B**) We measured the response of each quadrant to naturalistic motions and chose the quadrant weightings to minimize the mean squared error between the model output and the true velocity. (**C**) Comparison of the estimation performance of individual quadrants, multiple quadrants, and the HRC. The best two quadrants were (− −) and (− +); the best three also included (+ −). (**D**) The performance-optimized weighted 4-quadrant model reproduced the signs and approximate magnitudes of the psychophysical results.

*Figure 3. continued on next page*

## Drosophila circuitry contains additional elements that might facilitate motion estimation

The previous section suggested that the segregation of light and dark signals by *Drosophila*'s motion estimation circuitry might enhance naturalistic motion estimation in a manner that also generates the observed glider responses. In this section, we introduce three hierarchical models to investigate other features of *Drosophila*'s circuit that might have functional consequences for the processing of natural stimuli and gliders (*Table 1*). We refrain from modifying the temporal filtering of the motion estimator, and we focus on its nonlinear architecture.

The first of these models recasts the HRC and the weighted 4-quadrant model in a more general architecture. This model is the class of mirror anti-symmetric models that apply a 2-dimensional nonlinearity to the low-pass filtered signal from one point in space and the high-pass filtered signal from a neighboring point in space (*Figure 4A*). Since the observed glider responses indicate that flies use higher-order correlations of both even and odd order, we model this 2-dimensional nonlinearity as a fourth-order polynomial ('Materials and methods'). The HRC corresponds to the special case of this nonlinearity that multiplies the two inputs (*left*, *Figure 4B*). To emphasize how the model class in *Figure 4A* generalizes the HRC, we refer to it as the *non-multiplicative nonlinearity* model. In comparison, the weighted 4-quadrant model corresponds to a different nonlinearity that separately scales a pure multiplication in each quadrant of the Cartesian plane. Compared to the HRC, the optimized forms of both the weighted 4-quadrant model and the non-multiplicative nonlinearity model substantially attenuated positive low-pass filtered signals (*middle* and *right*, *Figure 4B*), though the non-multiplicative nonlinearity shows less attenuation. This model architecture provides enough flexibility to generate the glider response pattern (*Figure 4—figure supplement 1*).

The non-multiplicative nonlinearity model relaxes some restrictions of the 4-quadrant model. This is prudent because the exact nonlinear transformations implemented by neural circuits in the *Drosophila* brain remain poorly understood. For example, T4 and T5 are the first direction-selective neurons in the fly brain (*Maisak et al., 2013*), but the mechanism by which they become direction-selective is not yet known. Furthermore, neurons upstream of T4 and

*Figure 3. Continued*

The following figure supplements are available for figure 3:

**Figure supplement 1**. Separate ON and OFF processing improved motion estimation by supplementing the HRC with odd-ordered correlations.

**Figure supplement 2**. The weighted 4-quadrant model cannot reproduce the positive-negative parity asymmetry in the psychophysical data.

T5 imperfectly segregate light and dark information (*Behnia et al., 2014*) and show overlap between the two motion pathways (*Silies et al., 2013*), suggesting that ON/OFF segregation may not be crisply realized. We will discuss this model's estimation accuracy and glider performance in the next section.

*Drosophila*'s motion processing circuitry suggests two more generalizations of the non-multiplicative nonlinearity model. First, note that the non-multiplicative nonlinearity model inherits the HRC's assumption that each nonlinear unit only acts upon the low-pass filtered signal from one point in space and the high-pass filtered signal from the neighboring point (*Figure 4A*). In contrast, the converging 3-point correlator (*Figure 1H*) shows that the accuracy of motion estimation can sometimes be enhanced by nonlinearly combining both low-pass filtered signals (*Figure 1I*). Moreover, connectomic evidence conflicts with the non-multiplicative nonlinearity model's constraints, because each T4 cell receives synaptic connections from both the Mi1 cell and the Tm3 cells (T4's two major input channels) at overlapping points in space (*Takemura et al., 2013*). The *unrestricted nonlinearity* model removes this restriction of the non-multiplicative nonlinearity model by allowing a 4-dimensional nonlinearity to act on all four filtered signals (*Figure 4C*). Here, we again model this nonlinearity as a fourth-order polynomial ('Materials and methods'). The unrestricted nonlinearity allows the motion estimator to nonlinearly combine multiple temporal channels from the same point in space. Recent experiments indicate that the Mi1 and Tm3 cells alone are insufficient to account for the motion processing of the T4 channel (*Ammer et al., 2015*). Future work might generalize the unrestricted nonlinearity model to include three or more temporal channels at each point in space.

The models presented so far operate only on a pair of neighboring photoreceptors, and the final generalization incorporates a third point in space. Averaging EMDs over space improves the accuracy of whole-field motion estimation (*Dror et al., 2001*), but *Drosophila*'s neural circuitry suggests that it might adopt a more sophisticated strategy to combine signals across space. In particular, single T4 cells receive synaptic inputs from Mi1 cells and Tm3 cells from more than two retinotopic columns (*Takemura et al., 2013*). This arrangement could allow the circuit to incorporate higher-order correlations that are distributed across three or more spatial input channels. To explore whether this possibility has computational significance, we generalized the unrestricted nonlinearity model to provide unrestricted access to six temporal channels distributed across three points in space (*Figure 4D*). We refer to this model as the *extra input nonlinearity* model. We approximate its 6-dimensional nonlinearity as a fourth-order polynomial ('Materials and methods').

## Elaborated circuit architectures improve motion estimation without sacrificing glider responses

Having introduced the rationale behind the non-multiplicative, unrestricted, and extra input nonlinearity models, it is straightforward to examine their performance as motion estimators. First note that the polynomial non-multiplicative nonlinearity model was a better motion estimator (*Figure 4E*) than the weighted 4-quadrant model (*Figure 3C*). This implies that some useful signatures of naturalistic motion are not made accessible by simply segregating ON and OFF motion signals. Interestingly, this performance improvement is largely due to 3-point correlations, and models that exclude fourth-order polynomial terms still outperform the weighted 4-quadrant model (*Figure 4—figure supplement 2*). Third-order correlations are only useful for motion estimation because of light–dark asymmetries in natural stimulus statistics (*Fitzgerald et al., 2011*; *Clark et al., 2014*), so this result implies that ON/OFF segregation provides an imperfect way to account for the complexity of light–dark asymmetries found in the natural world. The non-multiplicative nonlinearity model also made novel use of low-order correlations to improve its motion estimate (Appendix 10).

The three models are hierarchical because the non-multiplicative nonlinearity model is a special case of the unrestricted nonlinearity model, which is itself a special case of the extra input

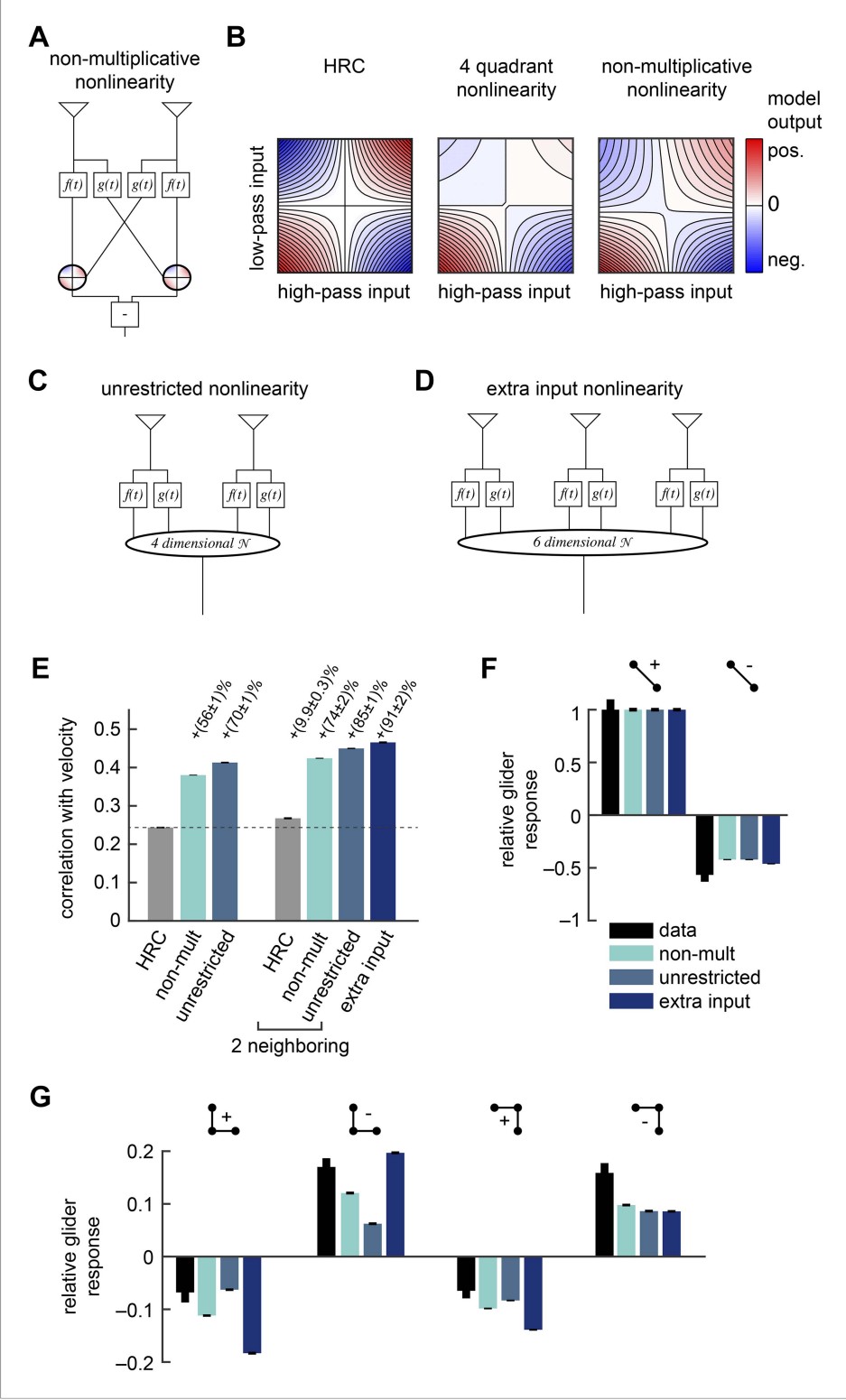

**Figure 4**. Several biologically motivated generalizations of the motion estimator further improved estimation performance without sacrificing glider responses. See *Table 1* for a description of the biological rationales behind these models. (**A**) The 'non-multiplicative nonlinearity' model substitutes a 2-dimensional nonlinearity for the pure multiplication of the HRC. Here, we approximated the nonlinearity with a fourth order polynomial. (**B**) Two-dimensional nonlinearities underlying the HRC, the weighted 4-quadrant model, and the non-multiplicative nonlinearity model. The latter models reflect optimized cases, in which the weighting coefficients maximized

*Figure 4. continued on next page*

*Figure 4. Continued*

estimation performance with natural inputs. Iso-output lines are shown in each plot, and the horizontal and vertical limits are chosen to include 95% of the naturalistic input signals. (**C**) Another generalization, the 'unrestricted nonlinearity' model allows all 4 input signals to be combined nonlinearly. We approximate this 4-dimensional nonlinearity with a fourth-order polynomial. (**D**) A final generalization, the 'extra input nonlinearity' model, relaxes the restriction that the motion estimator only uses 2 spatial inputs. We approximate this 6-dimensional nonlinearity with a fourth-order polynomial. (**E**) Comparison of the estimation performance of these models to the HRC. We compare the extra input nonlinearity model to the average of two neighboring motion estimators. (**F**, **G**) The three models correctly predicted the directions of psychophysical responses. The pattern of 3-point responses differed somewhat across the models, and the extra input nonlinearity model was the first to predict a large asymmetry between positive and negative 3-point glider responses.

The following figure supplements are available for figure 4:

**Figure supplement 1**. The non-multiplicative nonlinearity model can be tuned to account for the positive-negative parity asymmetry in the psychophysical data.

**Figure supplement 2**. The performance of the non-multiplicative nonlinearity model is plotted against the order of the fitted polynomial.

---

nonlinearity model. Thus, we expect each model to perform at least as well as its predecessor, but it is possible that some circuit elaborations will not introduce useful computational cues. Nevertheless, we found that that the unrestricted nonlinearity model performed better than the non-multiplicative nonlinearity model, and the extra input model performed better than the average of two neighboring unrestricted nonlinearity models (*Figure 4E*). Therefore, both models incorporated novel computational signatures with relevance for visual motion estimation. Although the relative improvements were fairly small, it's worth noting that the improvement from spatial averaging is also small, and it is possible that the fly brain builds an accurate motion estimator by combining a large number of weak predictors of motion.

Each of these three generalized models predicted 2-point glider responses (*Figure 4F*) that closely resembled the standard HRC (*left*, *Figure 1G*). Each model also correctly predicted the experimental turning directions to each of the 3-point glider stimuli (*Figure 4G*). The magnitudes of the 3-point glider turning responses did not unambiguously favor any of the three hierarchical models (*Figure 4G*) or the weighted 4-quadrant model (*right*, *Figure 3D*). Each model did better on some stimuli and worse on others. Nevertheless, the predicted glider responses did make several interesting points. First, the extra input nonlinearity model predicted a clear asymmetry between positive and negative 3-point gliders (*Figure 4G*). This shows that some of the even-ordered correlations found in 3-point glider stimuli have relevance for naturalistic motion estimation. Second, the observation that each model provides qualitatively similar glider response patterns illustrates that animals could use multiple nonlinear mechanisms to access ethologically relevant higher-order correlations. Future experiments should directly assess the functional relevance of the different models in the hierarchy. Finally, the qualitative agreement between all of these predictions and the experimental data supports the general hypothesis that glider responses could reflect underlying nonlinear mechanisms that facilitate motion estimation in natural environments.

## The extra input nonlinearity model contains the conceptual content of the other considered models

In this paper, we sequentially introduced several models in order to isolate specific ideas about the relationships between *Drosophila*'s behavior, its motion estimation circuit, and the statistical demands of accurate motion estimation in natural environments. The front-end nonlinearity model explored an interesting candidate principle for visual motion estimation, but it conflicted sharply with fly behavior (*Figure 2G*) and excluded the conceptual insights offered by other models. For example, the front-end nonlinearities we considered eliminated the asymmetry between light and dark contrasts (*Figure 2D*), removing the need for separate ON and OFF processing. However, the remaining models embodied ideas that are complementary rather than exclusive, and these models should not

be thought of as competitors. Instead, we will show here that the final, most general model incorporates the variety of conceptual points that were initially illustrated by specific models.

The structure of the non-multiplicative nonlinearity models can be directly plotted (*Figure 4B*), but is not easy to visualize the 6-dimensional nonlinearity that defines the extra input nonlinearity model. We therefore need an alternate technique to illustrate its computations. We proceed by leveraging three ideas. First, a wide variety of visual motion estimators can be expanded as an abstract series of multipoint correlators (e.g., see *Poggio and Reichardt, 1980*, *Fitzgerald et al., 2011*, and Appendices 6, 7, 11), and it is straightforward to pictorially represent a multipoint correlator (e.g., see *Figure 1A,H*, and more to come). In the extra input nonlinearity model, this expansion is immediate because we have already parameterized its 6-dimensional nonlinearity as a polynomial. Importantly, this expansion should be considered at the algorithmic level (*Marr and Poggio, 1976*), and we do not suggest that the wiring of brain circuits will reflect a large number of higher-order correlators. To the contrary, a large number of higher-order multipoint correlators may be implemented implicitly by high-dimensional nonlinearities suggested by *Drosophila*'s visual circuitry. Second, we note that certain multipoint correlators can be recombined into a 2-dimensional non-multiplicative nonlinearity that facilitates easy comparisons with the HRC and weighted 4-quadrant models (e.g., see *Figure 4B*). Taken together, these two points mean that we can represent the performance-optimized extra input nonlinearity model in terms of non-multiplicative nonlinearity models and multipoint correlators, each of which are easy to represent graphically.

This graphical representation could be unwieldy because of the shear number of higher-order correlators in the model. Thus the third and final point is that we need a way to identify a relatively small number of terms that substantially improve the accuracy of motion estimation and illustrate the conceptual content of the model. To achieve this, we used lasso regression (*Tibshirani, 1996*) to identify models with fewer multipoint correlators that still enabled accurate motion estimation ('Materials and methods'). This analysis revealed that fewer than half of all multipoint correlators were needed to account for the full accuracy of the extra input nonlinearity model (*rightmost bars*, *Figure 5A*). In fact, the accuracy of naturalistic motion estimation increased rapidly as the few correlators were sequentially added (*left bars*, *Figure 5A*), and a model that used 16 out of the 209 possible predictors was already able to produce 74% of the gain offered by the full extra input nonlinearity model (*red bar*, *Figure 5A*).

The leading 16 predictors compactly illustrated how the extra input nonlinearity model recapitulates the conceptual advances offered by the other models (*Figure 5B*). Four of the predictors combine to implement a mirror-symmetric non-multiplicative nonlinearity model that acts on the first and second points in space (*first term*, *Figure 5B*). The dominant contribution to the nonlinearity is the HRC's multiplier, but an additional third-order term breaks the symmetry between positive and negative low-pass filtered signals (*Figure 5—figure supplement 1*). Thus, the extra input nonlinearity model approximately correlates neighboring points in space, as the HRC would suggest, but it differentially weights positive and negative low-pass filtered signals, like the weighted 4-quadrant model. It also replicates the main insight from the non-multiplicative nonlinearity model: the best treatment of asymmetric light and dark information need not be as simple as pure ON/OFF segregation. The model used another eight predictors to construct two more non-multiplicative nonlinearity models, one that surveyed the second and third points in space and another that surveyed the first and third points (*first and second terms*, *Figure 5B*, *Figure 5—figure supplement 1*). These components make the previously highlighted conceptual points and add the observation that spatial averaging improves estimates.

The final four predictors implemented two mirror anti-symmetric multipoint correlators (*third and fourth terms*, *Figure 5B*). In particular, two predictors went towards implementing a converging 3-point correlator that spanned the first and third spatial points (*third term*, *Figure 5B*). This estimator made the model's asymmetric treatment of light and dark signals more nuanced than permitted by the non-multiplicative nonlinearity model, and it also incorporated motion signals that combine multiple temporal signals from the same point in space. This latter point was the main conceptual motivation for the unrestricted nonlinearity model. Finally, the last two predictors implemented a 4-point correlator that combined temporal signals from three distinct points in space (*fourth term*, *Figure 5B*). This component reinforces the conceptual motivation for the extra input nonlinearity model and gives a concrete example of a computationally relevant higher-order correlator that is distributed across three points in space. It's interesting that the leading fourth-order correlator

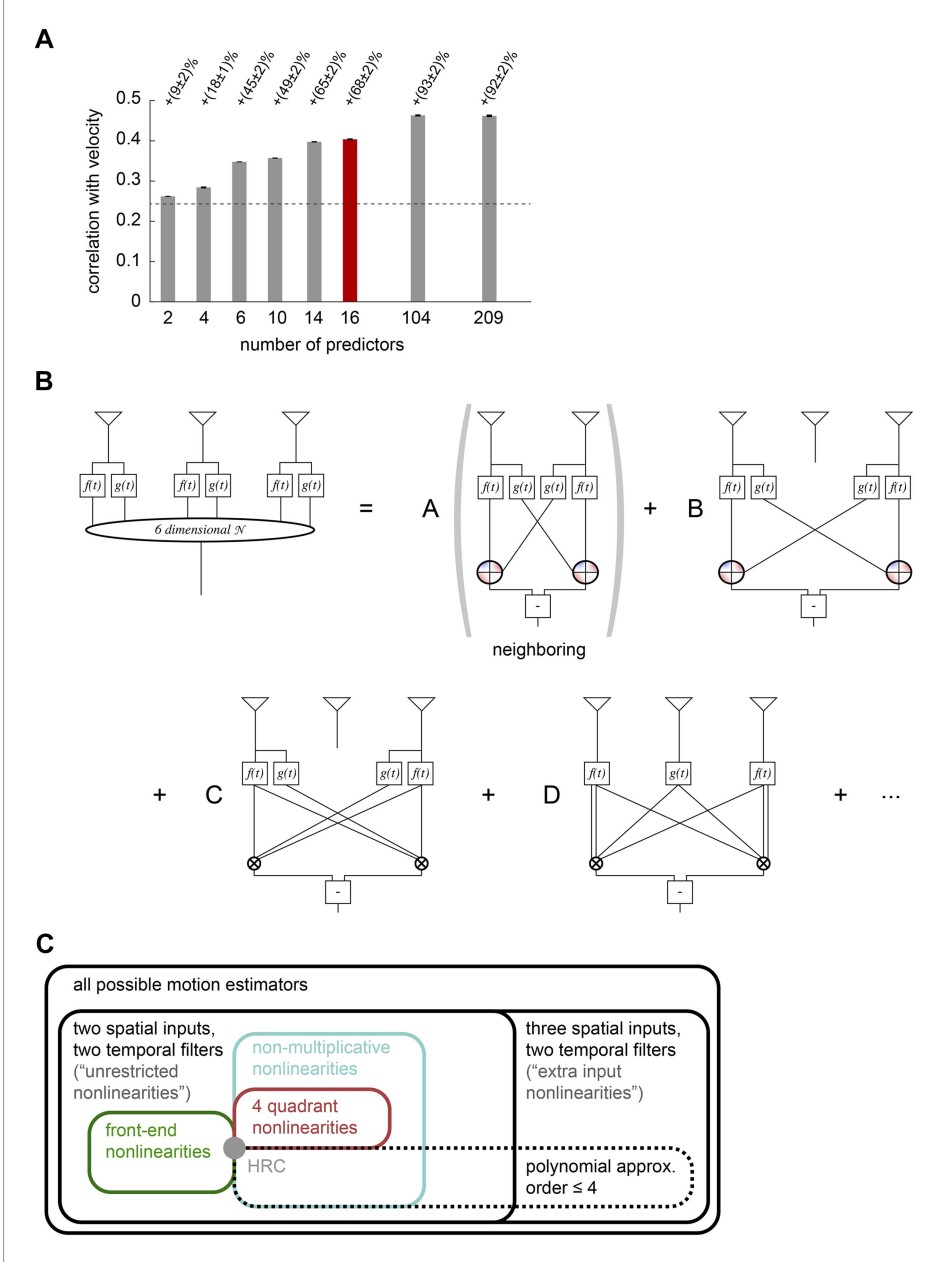

**Figure 5**. Computational interpretation of the extra input nonlinearity model. (**A**) We used lasso regression to select subsets of predictors that might enable accurate estimation (see 'Materials and methods'). With only 16 predictors, the model improved naturalistic performance over the HRC by 68%, and including fewer than half of the predictors improved it by the full 92%. The maximum number of predictors corresponds to the number of polynomial coefficients that were fit in the full model. (**B**) We visualized the 6-dimensional nonlinearity as the sum of several simpler computational modules. When only 16 predictors were used (red bar in (**A**)), the model used four distinct types of computations. In particular, the model included nearest-neighbor and next-nearest-neighbor non-multiplicative nonlinearities (*top row*). It also included a converging 3-point correlator from the two furthest photoreceptors and a 4-point correlator that combined three spatial inputs (*bottom row*). (**C**) Venn diagram illustrating the hierarchical nesting of models used in this paper. All models in this paper contain sets of parameters that reproduce the HRC (gray dot). The weighted 4-quadrant model is a subset of non-multiplicative nonlinearity models, which are themselves a subset of unrestricted nonlinearity models. The extra input nonlinearity encompasses all the models. When we approximated the nonlinearites with fourth order polynomials, we restricted them to a smaller portion of the model space. The 4-quadrant nonlinearities
*Figure 5. continued on next page*

*Figure 5. Continued*

only overlapped with the fourth-order polynomial approximation at the HRC, because the weighted 4-quadrant model is infinite order when expanded as a polynomial (see Appendix 7).

The following figure supplement is available for figure 5:

**Figure supplement 1**. Structure of non-multiplicative nonlinearities in the extra input model of *Figure 5B*.

spanned three spatial points, because the extra input nonlinearity model was the first performance-optimized model that generated a substantially asymmetric response to positive and negative 3-point gliders (*Figure 4G*).

This paper set out with the goal of exploring whether the statistical demands of naturalistic motion estimation could provide a useful lens for interpreting features of *Drosophila*'s behavior and neural circuitry that push beyond the canonical HRC. Although we have considered several interesting classes of visual motion estimators, the space of possible motion estimators is much larger (*Figure 5C*). For instance, these models have not explored the impact of temporal filter choice on naturalistic motion estimation. Nor have they assessed the possibility of more than two temporal filters, which is be suggested by anatomical (*Takemura et al., 2013*) and physiological (*Ammer et al., 2015*) experiments. More generally, the neural circuits contributing to *Drosophila*'s motion estimator are still incompletely known, and the extent to which the fly brain's biological complexity reflects computational sophistication remains an open question. Theoretical considerations will be critical for resolving that question and pinpointing the most relevant principles underlying visual motion estimation.

## Discussion

Ongoing research is providing an increasingly detailed picture of the anatomy and physiology of the visual circuitry that implements motion processing in *Drosophila*. Through the combination of genetic silencing experiments, connectomic analysis, and functional recordings, researchers have identified many individual neurons in the fly brain that contribute to visual motion processing (*Silies et al., 2014*). Although the HRC provided the initial theoretical impetus for these experiments, specific experimental outcomes have often been unanticipated. For instance, the fly brain contains multiple pathways that segregate different types of motion information (*Joesch et al., 2010*; *Clark et al., 2011*; *Silies et al., 2013*); its direction-selective neurons receive inputs from more than two neighboring points in visual space (*Takemura et al., 2013*); and the biological substrates for reverse-phi signals, which were fundamental to the formulation of the HRC, remain poorly understood (*Clark et al., 2011*; *Tuthill et al., 2011*; *Joesch et al., 2013*). Theoretical work to illuminate the computational significance of these various discrepancies is critical for understanding *Drosophila*'s motion estimator.

The results presented in this paper provide a new theoretical perspective on these experimental results. While previous research has addressed how neural circuits could use four quadrants to carry out algebraic multiplication, here, the recurring theme of our models was that motion-processing circuits should treat light and dark signals differently for functional reasons. We first showed that visual systems could use ON and OFF processing channels that separately correlate light and dark signals to improve the accuracy of motion estimation (*Figure 3*). This model was inspired by the experimental observation that *Drosophila*'s motion processing channels distinguish between light increments and decrements (*Joesch et al., 2010*; *Clark et al., 2011*), but this study is the first to explicitly demonstrate how such processing channels can improve the accuracy of motion estimation. Furthermore, our model shows that both the phi channels (i.e., the $(+\ +)$ and $(-\ -)$ and quadrants) and the reverse-phi channels (i.e., the $(+\ -)$ and $(-\ +)$ quadrants) can contribute productively to motion estimation in natural environments. Since many animals experience similar sensory statistics and ON and OFF visual processing channels are pervasive across visual systems (*Schiller, 1992*; *Westheimer, 2007*), these mechanisms might be very general. Ultimately, the performance gains from weighted quadrants were a consequence of statistical asymmetries between light and dark contrasts in natural images, and our models showed that neural circuits could perform even better if they made distinctions between light and dark signals that were subtler than simple ON/OFF segregation

(*Figure 4*). Recent experimental evidence indicates that the fly's motion processing channels are imperfectly selective for ON vs OFF information (*Silies et al., 2013*; *Behnia et al., 2014*; *Strother et al., 2014*), and it is important that future experiments characterize such subtleties in the computations performed by these circuits.

Our most general model contained three spatial inputs and showed that spatial averaging of local motion detectors was suboptimal (*Figure 4E*). Anatomy suggests that single T4 cells receive inputs from several different retinotopic columns, and also from multiple neuron types in a single retinotopic column (*Takemura et al., 2013*). Our modeling suggests that these two forms of circuit heterogeneity could enhance motion estimation by facilitating computations that go beyond averaging to compute higher-order correlations that are distributed across multiple points in space (*Figure 5B*). Overall, our results demonstrate how the subtleties of neural circuit nonlinearities can improve motion detection with naturalistic inputs. It therefore seems likely that some of the complexities of *Drosophila*'s circuitry are critical to its performance under natural conditions.

It is remarkable that our approximation of natural motion by the rigid translation of natural images revealed substantial utility for higher-order correlations in motion processing. Truly naturalistic motion would include spatial velocity gradients, occlusion, expansion, and contraction, yet the simplified naturalism we used to optimize our models already sufficed to account for many aspects of the fly's glider responses. This may be because the rotational optomotor response measured in the fly experiments is thought to be sensitive primarily to full-field rotations, which our naturalism emulates well. However, since other higher-order correlations may be associated with non-rigid translation (*Nitzany and Victor, 2014*), one might expect a different set of glider sensitivities to be optimal in the context of other motion-guided behaviors, such as looming responses (*Gabbiani et al., 1999*; *Tammero and Dickinson, 2002*; *Card and Dickinson, 2008*). Since a common elementary motion detector might underlie many or all motion-guided behaviors, incorporating more complex optic flow patterns may even diminish discrepancies between our models and *Drosophila*'s behavior.

The approach of this study is also relevant to vertebrate vision, where researchers typically model motion estimation using the motion energy model (*Adelson and Bergen, 1985*). Like the HRC, the motion energy model only responds to 2-point correlations in the visual stimulus. Consequently, many of the theoretical considerations in this paper apply directly to the motion energy model. Furthermore, each of our computational models can be straightforwardly generalized to the architecture of the motion energy model. For example, one could incorporate non-multiplicative nonlinearities by replacing the squaring operation of the motion energy model with a more flexible nonlinearity. Nevertheless, the numerical benefits offered by each modification to the motion energy model might differ from those found for the HRC because the motion energy model and HRC use distinct spatial and temporal filtering. Such differences could in principle manifest themselves as a different pattern of predicted glider responses (*Hu and Victor, 2010*; *Clark et al., 2014*), but comparative electrophysiology experiments in macaques and dragonflies currently suggest that similarities between primate and insect motion processing are abundant (*Nitzany et al., 2014*).

Our models make predictions that are testable with new experiments. Researchers hypothesize that the T4 and T5 neurons in the fly lobula nonlinearly combine visual inputs across space and time to become the first direction-selective neurons in *Drosophila*'s visual system (*Maisak et al., 2013*). In accordance with the HRC model, conventional wisdom says that these neurons will multiply their input channels. In contrast, we predict that T4 and T5 will combine their visual input streams with non-multiplicative nonlinearities that facilitate accurate motion estimation in natural sensory environments. It's crucial to note that subtle differences between biology's nonlinearity and a pure multiplication can correspond to substantial functional effects. In particular, the optimized nonlinearity that we found here (*Figure 4B*) is superficially similar a simple multiplication, yet its subtle distinctions manifest themselves by improving the local estimation accuracy of the HRC by an impressive margin (*Figure 2D*).

In this paper, we studied several simple models to most clearly illustrate the computational consequences of fundamental nonlinear circuit operations. Each of these operations individually provided a way for *Drosophila* to improve their motion estimation accuracy in natural environments, but they are not necessarily exclusive. For example, if a front-end nonlinearity does not fully remove the asymmetry between light and dark contrasts, then subsequent ON and OFF processing might further improve estimation accuracy. Similarly, non-multiplicative nonlinearities might enable an even better combination of ON and OFF signals for motion estimation. The general approach that we

adopted here is to restrict the space of candidate models to those that have immediate biological relevance and to identify interesting models by optimizing the model's estimation accuracy over naturalistic stimuli. Future models should incorporate more biological details to better emulate the specifics of *Drosophila*'s visual circuitry, which is rapidly being dissected through unprecedented anatomical, functional, and behavioral experiments (*Silies et al., 2014*).

# Materials and methods

## Simulated ensemble of naturalistic motions

We simulated the linear responses of neighboring photoreceptors to naturalistic motion using methods similar to previous work (*Clark et al., 2014*). We began with a database of natural images (*van Hateren and van der Schaaf, 1998*). We converted each natural image to a contrast scale, $C(\vec{x}) = (I(\vec{x}) - I_0)/I_0$, where $C(\vec{x})$ is the contrast at the spatial point $\vec{x}$, $I(\vec{x})$ is its intensity, and $I_0$ is the average intensity across the image. Since we only consider horizontal motion, we emulated the spatial blurring of *Drosophila*'s photoreceptors in the vertical dimension by filtering across rows with a Gaussian kernel (FWHM = 5.7°). We then took the central row of each filtered image to represent a one-dimension variant of the natural image, denoted $c(x)$. We applied reflective boundary conditions to generate images that covered 360° and down-sampled each resulting image to 1° pixels by averaging. Photoreceptor blurring from signals in the horizontal dimension depends on the velocity of motion. In particular, we model the response of the *i*th photoreceptor as

$$V_i(t) = \int dt' \, T(t') \int dx \, M(x - x_i) c(x - v(t - t')),$$

where $T$ is a causal exponential kernel (timescale = 10 ms), $M$ is a Gaussian kernel (FWHM = 5.7°), $x_i$ is the location of the *i*th photoreceptor, and $\nu$ is the velocity of motion.

Each naturalistic motion comprised a randomly selected one-dimensional natural image, an offset to set the initial location of the photoreceptors, and a velocity drawn from a zero-mean normal distribution with a standard deviation of 90°/s. In this manner, we simulated the responses of three horizontally adjacent photoreceptors (spaced by 5.1°) to $5 \times 10^5$ naturalistic motions (each with duration = 800 ms, time step = 5 ms). We then explicitly enforced left-right symmetry in the naturalistic ensemble by pairing each naturalistic motion with a new simulated motion, in which the natural image is reflected, the velocity is inverted, and the offset is chosen such that $\{V_1(t), V_2(t), V_3(t)\}$ in the new naturalistic motion is exactly $\{V_3(t), V_2(t), V_1(t)\}$ from its partner. The final symmetric ensemble thus consists of $10^6$ naturalistic motions.

## The HRC

The HRC applies two temporal filters to its photoreceptor inputs. We denote the kernels of the low-pass and high-pass filters as $f$ and $g$, respectively, such that the output of a local HRC is

$$R(t) = (f*V_1)(t)(g*V_2)(t) - (g*V_1)(t)(f*V_2)(t),$$

where * denotes convolution (*Figure 1A*). We consider the HRC's velocity estimate for a given naturalistic motion as its value at the final time point of the simulation. We model the filter kernels as

$$f(t) = t e^{-t/\tau}, t \geq 0$$

and

$$g(t) = \frac{df(t)}{dt},$$

where $\tau$ = 20 ms and $g(t)$ is comparable to lamina monopolar cell responses (*Clark et al., 2011*; *Behnia et al., 2014*). We built the alternate motion estimators considered in this work from the same four filtered signals, $\{(f*V_1), (g*V_1), (f*V_2), (g*V_2)\}$, always considering the estimator's output at the final time point as its velocity estimate. Thus, none of our models modified the spatial or temporal processing of the HRC, reflecting our emphasis on how nonlinear processing might be tuned for naturalistic motion estimation. The global output of an array of HRCs would be obtained by pooling

signals across space. Here we focus on spatiotemporally local strategies for motion estimation and at most pool motion signals across two neighboring motion detectors.

## Relationship between the mean squared error and the correlation coefficient

We evaluate motion estimators by the mean squared error between their output and the true velocity. To minimize the mean squared error of the HRC, we scale its output by $r^{(R)}\sigma_\nu/\sigma_R$, where $r^{(R)}$ is the correlation coefficient between the HRC's output and the velocity of motion, $\sigma_\nu$ is the standard deviation of the velocity distribution, and $\sigma_R$ is the standard deviation of the HRC's output. Once the HRC is scaled in this manner, its mean squared error is

$$\epsilon = \sigma_v^2 \left( 1 - \left( r^{(R)} \right)^2 \right).$$

More generally, this equation rewrites the mean squared error of any optimally scaled motion estimator in terms of its correlation coefficient with the velocity. All motion estimators considered in this paper are optimally scaled, and we find the correlation coefficient to be more intuitive than the mean squared error. We thus always report the performance of each motion estimator in terms of the correlation coefficient between the true and estimated velocity.

## Model fitting procedure

We fit the linear weighting parameters in the models of *Figures 1I, 3A, 4A,C,D* to maximize the estimation accuracy over a simulated ensemble of naturalistic motions. The formulas provided in subsequent sections of the 'Materials and methods' will cast each motion estimation scheme as a linear combination of a variety of motion predictors,

$$v_e = \sum_i w_i x_i,$$

where the $x_i$ are nonlinear combinations of $\{(f*V_1), (g*V_1), (f*V_2), (g*V_2), (f*V_3), (g*V_3)\}$ that depend on the model architecture, and the $w_i$ are associated weighting coefficients. We chose the weights to minimize the mean squared error between the true and predicted velocity, which is the standard scenario considered by ordinary least-squares regression. The same weights maximize the correlation coefficient between the true and predicted velocity, and we typically present model accuracies as correlation coefficients.

We used twofold cross-validation to protect against over-fitting. In particular, we randomly divided the ensemble of naturalistic motions into a training set of 500,000 symmetrically paired examples and a testing set of the remaining 500,000 examples. We determined the weighting coefficients by minimizing the empirical error over the training set, and we reported accuracies over the test set. To estimate error bars for each model's accuracy, we computed twenty random divisions of the naturalistic motion ensemble and calculated the standard deviation of the estimation accuracy.

## Model responses to glider stimuli

We generated 25 random instantiations of each glider stimulus considered by our previous experimental work (*Figure 1F*, duration = 3 s, update rate = 40 Hz, pixel size = 5°) (*Hu and Victor, 2010*; *Clark et al., 2014*). We evaluated the response of each model to these stimuli by averaging the outputs of 60 identical local motion estimators (each separated by 5.1°) over the last two seconds of visual stimulation. Glider predictions were equal and opposite for the left and right variants of the stimuli, so we pooled leftward and rightward stimuli in all figures (*Figure 1F* shows the rightward variants). We scaled each model's output such that the average response to the positive 2-point glider was 1. All figures associated with glider responses show the mean and standard error of each model's response across the 25 glider instantiations.

## Front-end nonlinearity model

The model in *Figure 2A* replaces the linear photoreceptor signals, $V_1$ and $V_2$, with nonlinear photoreceptor signals

$$y_i = h(V_i),$$

where $h$ is some nonlinear function. Thus, the motion estimate from the front-end nonlinearity model is

$$F = (f*y_1)(g*y_2) - (g*y_1)(f*y_2).$$

To implement the contrast equalizing nonlinearity, we replaced values of $V_i(t)$ by their rank-order (scaled and shifted to range between $-1$ and $+1$). Note that all $V_i(t)$ were sorted together (i.e., including all spatial points, temporal points, and simulated naturalistic motions). When multiple $V_i(t)$ had the same value, they were given the same rank. To implement binarizing nonlinearities, we again sorted the $V_i(t)$ and found the values corresponding to the threshold locations. For example, to calculate the binarizing nonlinearity with two steps (Appendix 4): (i) we found the $V_i(t)$ values corresponding to the 25th and 75th percentiles; (ii) signals below the 25th percentile or above the 75th percentiles were assigned the value of $-1$; and (iii) signals between 25th and 75th percentiles were assigned the value of $+1$. To implement the Gaussianizing nonlinearity, we again rank-ordered the $V_i(t)$ (scaled to range between 0 and 1) and applied the inverse Gaussian cumulative distribution function to these ranks. The HRC is the special case of this model where the front-end nonlinearity is linear.

## Weighted 4-quadrant model

The weighted 4-quadrant model in **Figure 3A** separately correlates bright and dark signals. Mathematically, it is

$$Q = \sum_{a \in \{+,-\}} \sum_{b \in \{+,-\}} w_{ab}^{(Q)} Q_{ab},$$

where $w_{ab}^{(Q)}$ are adjustable weights that parameterize the model,

$$Q_{ab} = [(f*V_1)]_a [(g*V_2)]_b - [(g*V_1)]_b [(f*V_2)]_a,$$

$[x]_+$ is equal to $x$ when $x$ is positive and zero otherwise, and $[x]_-$ is equal to $x$ when $x$ is negative and zero otherwise. The HRC is the special case of this model where $w_{++}^{(Q)} = w_{+-}^{(Q)} = w_{-+}^{(Q)} = w_{--}^{(Q)}$.

## Non-multiplicative nonlinearity model

The non-multiplicative nonlinearity model in **Figure 4A** replaces the HRC's multiplication step with a more flexible two-dimension nonlinearity. In particular, it is

$$N = \eta((f*V_1), (g*V_2)) - \eta((f*V_2), (g*V_1)),$$

where we approximate the nonlinearity, $\eta$, as a fourth-order polynomial

$$\eta(x,y) = \sum_{i=0}^{4} \sum_{j=0}^{4-i} w_{ij}^{(N)} x^i y^j,$$

and $w_{ij}^{(N)}$ are adjustable weights that parameterize the model. We include terms up to fourth order in this model to ensure that it is flexible enough to describe the published glider response data. In particular: (i) the second-order terms accommodate responses to 2-point glider stimuli; (ii) the third-order terms accommodate parity-inverting responses to 3-point glider stimuli; and (iii) the fourth-order terms enable the model to respond with unequal magnitude to positive and negative parity 3-point glider stimuli (**Figure 4—figure supplement 1**). Thus, this model has 14 parameters. The HRC is the special case of this model where only $w_{11}^{(N)}$ is nonzero.

## Unrestricted nonlinearity model

Here we model the 4-dimensional nonlinearity in **Figure 4C** as a fourth-order polynomial of the four filtered signals in the HRC. In general, this motion estimator is

$$S = \sum_{i=0}^{4} \sum_{j=0}^{4-i} \sum_{k=0}^{4-i-j} \sum_{l=0}^{4-i-j-k} w_{ijkl}^{(S)} (f*V_1)^i (g*V_1)^j (f*V_2)^k (g*V_2)^l,$$

where $w_{ijkl}^{(S)}$ are adjustable weights that parameterize the model, and we set $w_{0000}^{(S)} = 0$ because this term has no utility for naturalistic motion estimation. Thus, this model has 69 parameters. The HRC is the special case of this model where $w_{1001}^{(S)} = -w_{0110}^{(S)} \neq 0$, and all other parameters are zero.

## Extra input nonlinearity model

Here we model the 6-dimensional nonlinearity in *Figure 4D* as a fourth-order polynomial of the six filtered signals in two neighboring HRCs. In general, this motion estimator is

$$E = \sum_{i=0}^{4} \sum_{j=0}^{4-i} \sum_{k=0}^{4-i-j} \sum_{l=0}^{4-i-j-k} \sum_{m=0}^{4-i-j-k-l} \sum_{n=0}^{4-i-j-k-l-m} w_{ijklmn}^{(E)} (f*V_1)^i (g*V_1)^j$$
$$\times (f*V_2)^k (g*V_2)^l (f*V_3)^m (g*V_3)^n,$$

where $w_{ijklmn}^{(E)}$ are adjustable weights that parameterize the model, and we set $w_{000000}^{(E)} = 0$ because this term has no utility for naturalistic motion estimation. Thus, this model has 209 parameters. The average of two neighboring HRCs is the special case of this model where $w_{100100}^{(E)} = -w_{011000}^{(E)} = w_{001001}^{(E)} = -w_{000110}^{(E)} \neq 0$, and all other parameters are zero.

## Lasso regression for predictor selection

Lasso regression augments the squared error with an $L_1$ penalty on nonzero weighting coefficients that favors sparse solutions (*Tibshirani, 1996*). We used lasso regression to identify subsets of predictors that might enable accurate motion estimation (*Figure 5A*). Once we identified a predictor subset using lasso regression, we refit the nonzero model weights using ordinary least squares regression (i.e., without the weight penalty).

## Acknowledgements

The authors thank Ruben Portugues, Haim Sompolinsky, and Florian Engert for helpful conversations. JEF acknowledges fellowship support from the Swartz Foundation. DAC acknowledges support from a Searle Scholar Award, a Sloan Research Fellowship, and the Smith Family Foundation.

## Additional information

### Funding

| Funder | Grant reference | Author |
|---|---|---|
| Swartz Foundation | Postdoctoral Fellowship | James E Fitzgerald |
| Searle Scholars Program | Scholar Award | Damon A Clark |
| Richard and Susan Smith Family Foundation | Scholar Award | Damon A Clark |
| Alfred P. Sloan Foundation | Sloan Research Fellowship in Neuroscience | Damon A Clark |

The funders had no role in study design, data collection and interpretation, or the decision to submit the work for publication.

### Author contributions

JEF, Conceived of research, designed research, performed research analyzed data, and wrote paper; DAC, Conceived of research, designed research, and wrote paper

### Author ORCIDs

James E Fitzgerald, http://orcid.org/0000-0002-0949-4188
Damon A Clark, http://orcid.org/0000-0001-8487-700X

## Additional files

### Supplementary file

• Source code 1. Code used to generate figures.

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

## Appendix 1

### Visual signatures of motion.

The pattern of light that stimulates the retina encodes information about the relative motion between the retina and its visual environment. The manner in which this information is encoded depends on the geometry of the photoreceptor array, the statistics of self-motion, and the statistics of the visual environment. The principal goal of this paper is to illustrate several ways that the brain's nonlinear processing of visual motion signals might be tuned to reflect specific features of the natural visual environment. We thus begin by enumerating some computational signatures of visual motion in natural environments, thereby exposing a diversity of stimulus features that visual system nonlinearities might aim to extract.

In the real world, animals encounter visual environments that are intricately structured and far from random (*Appendix figure 1A*) (*Ruderman and Bialek, 1994*; *van Hateren and van der Schaaf, 1998*; *Geisler, 2008*). When an animal rotates with constant angular velocity through the environment, the spatiotemporal response profile of the photoreceptor array encodes the velocity of self-motion through the slope of oriented streaks in space-time (*front face*, *Appendix figure 1B*) (*Adelson and Bergen, 1985*). Thus, a visual system with a dense array of noiseless photoreceptors could extract the angular velocity of an arbitrary image by computing the ratio of temporal and spatial derivatives (*Potters and Bialek, 1994*). The statistics of the image ensemble become relevant once multiple interpretations of the sensory world become logically consistent with the photoreceptor data. In particular, the optimal motion estimator depends on the statistics of the image ensemble when photoreceptors have noise (*Potters and Bialek, 1994*; *Fitzgerald et al., 2011*), and a nonzero spacing between photoreceptors introduces ambiguity via aliasing (*Potters and Bialek, 1994*). In these cases, the animal can use prior information regarding the sensory environment and its motion in order to weigh the plausibility of each sensory interpretation.

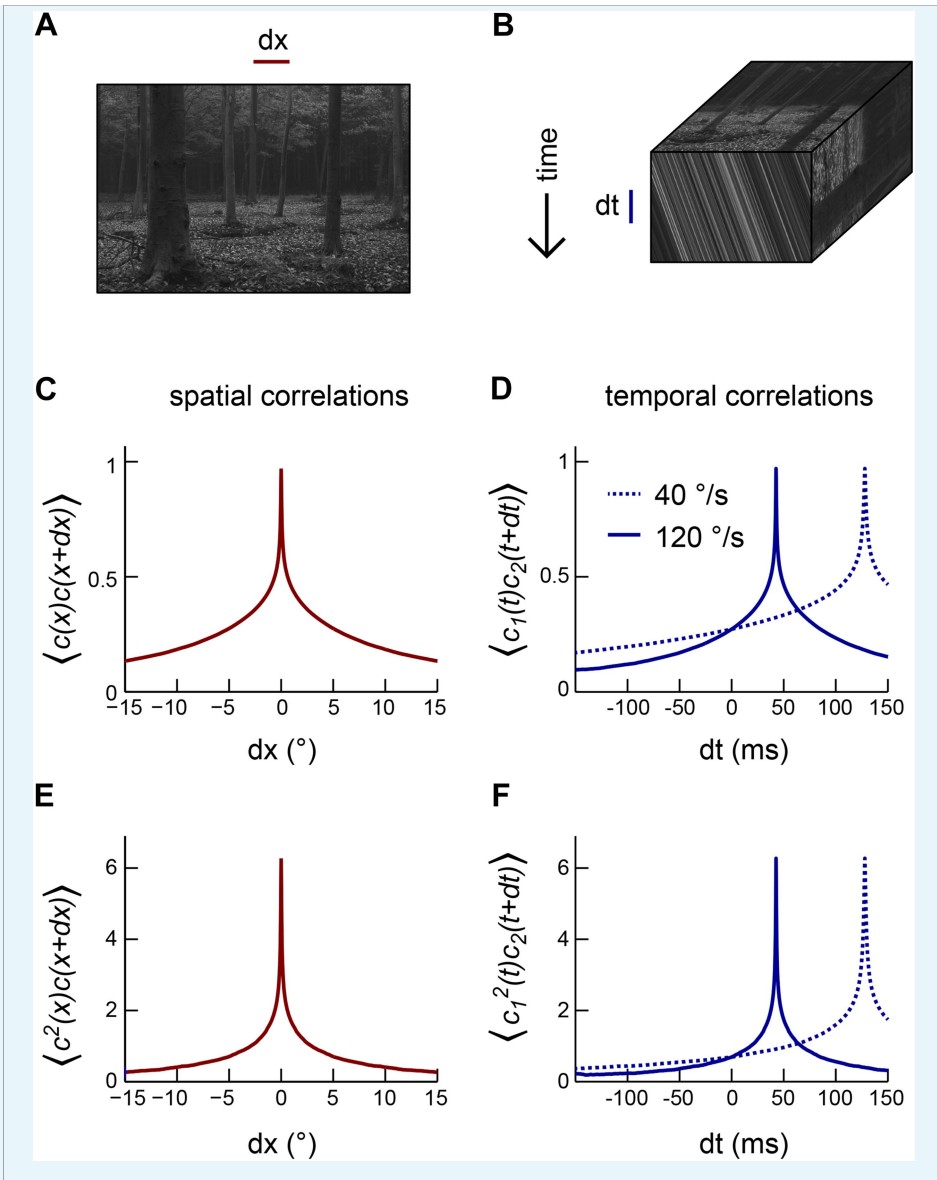

**Appendix figure 1**. Motion transforms spatial correlations into temporal correlations. (**A**) An example natural image (**van Hateren and van der Schaaf, 1998**). (**B**) When a natural image (*top face*) moves to the right, streaks in space-time (*front face*) indicate the direction and speed of the motion. Alternatively, motion influences the temporal correlation structure of visual signals (*side face*). (**C**) Second-order correlation function between pairs of spatially separated contrast signals (across the natural image ensemble [**van Hateren and van der Schaaf, 1998**]). (**D**) For constant velocity motion, the temporal correlation function between a pair of spatially separated points is shifted and stretched relative to the spatial correlation function. We separated the two points by *Drosophila*'s photoreceptor spacing (5.1°). (**E**) Example third-order spatial correlation function involving two points in space. (**F**) As with pairwise correlations, higher-order temporal correlations between spatially separated visual signals are shifted and stretched (relative to higher-order spatial correlation functions) in a manner that indicates the speed and direction of motion. DOI: 10.7554/eLife.09123.015

Full field motion transforms spatial features (*top face*, **Appendix figure 1B**) into temporal features (*side face*, **Appendix figure 1B**) in a manner that depends upon the velocity of motion. Consequently, one can also think about the visual signatures of motion in terms of spatiotemporal correlations between photoreceptors. The luminance contrast encoded by the $i$th photoreceptor is $C_i(t) = (I_i(t) - I_0)/I_0$, where $I_i(t)$ is the luminance intensity seen by the $i$th photoreceptor at time $t$ and $I_0$ is the average luminance intensity over the visual field. Thus, the

average contrast is zero, and the simplest correlation function corresponds to the product of two spatially separated contrast signals. Measured over an ensemble of natural images, this 2-point correlation function had a global maximum at zero spatial offset (**Appendix figure 1C**). Consequently, the velocity of motion is encoded by the peak of the temporal cross-correlation function between two neighboring photoreceptors, which occurs at the temporal offset that equals the photoreceptor spacing (5.1° for *Drosophila*) divided by the velocity of motion (**Appendix figure 1D**). Natural images also contain many higher-order correlations (**Ruderman and Bialek, 1994**; **Geisler, 2008**). For instance, the nonzero skewness of natural images implies that the third-order correlation that multiplies the contrast at one point with the squared contrast at a neighboring point also has a peak at zero spatial offset (**Appendix figure 1E**). Correspondingly, the peak of the temporal 3-point correlation function between neighboring photoreceptors encodes the velocity of motion (**Appendix figure 1F**). This argument generalizes to $n$th-order correlation functions when the ensemble of natural images has a nonzero $n$th moment. Note that this argument does not necessarily imply that a motion estimator would benefit from the incorporation of all nonzero correlation functions, because the velocity signals provided by one correlation function could be redundant with those provided by others.

Importantly, photoreceptor correlation functions also encode velocity information away from their peaks. For example, the velocity of motion influences the widths of the temporal cross-correlation functions between pairs of photoreceptors (**Appendix figure 1D,F**). To see this, note that the values of the temporal correlation functions at zero temporal offsets are velocity independent, whereas the peak locations are closer to zero for larger speeds (**Appendix figure 1D,F**). This implies a more rapid falloff for higher speeds. This fundamental effect occurs because nearby points are more correlated in natural environments and photoreceptors rapidly survey distant points when the speed of motion is high.

The description above illustrates how visual motion becomes encoded in photoreceptor correlations. A central goal of research in visual motion estimation is to understand how neural circuits invert (or decode) that encoding of velocity. Just as a broad class of functions can be represented as a power series, a broad class of motion estimators can be represented as a Volterra series (**Poggio and Reichardt, 1980**; **Fitzgerald et al., 2011**). Each term in the Volterra series can be interpreted as a multipoint correlator that decodes velocity information from a specific correlation function (**Fitzgerald et al., 2011**). For example, the HRC and the motion energy model are 2-point correlators that decode velocity from 2-point correlations, whereas the Bayes optimal motion estimator capitalizes on a wider variety of correlation functions (**Potters and Bialek, 1994**; **Fitzgerald et al., 2011**). Because multipoint correlators relate intuitively to measurable properties of the image ensemble, we will find that decomposing a motion estimator in terms of multipoint correlators is often illuminating. Moreover, we will use multipoint correlators as a common basis to compare the computations performed by mechanistically distinct models.

## Appendix 2

### Accuracy of 2-point correlators.

In this section we derive an expression for the accuracy of a general 2-point correlator in terms of the statistics of naturalistic motion.

We consider a general 2-point correlator that temporally correlates visual signals from the spatial points $i$ and $j$. Mathematically, this estimator has the form

$$v_e^{(2)}(t) = \int dt_1 \int dt_2 k_{i,j}^{(2)}(t_1, t_2) V_i(t - t_1) V_j(t - t_2), \tag{1}$$

where the 2-point kernel, $k_{i,j}^{(2)}(t_1, t_2)$, defines the correlator by specifying how each 2-point correlation contributes to the motion estimate. We model the response of the $i$th photoreceptor as

$$V_i(t) = \int d\tau T(\tau) \int d\theta M(\theta - \theta_i) c\left(\theta - \int_0^{t-\tau} dt' v(t')\right), \tag{2}$$

where $T$ is a temporal integration kernel, $M$ is the photoreceptor's spatial acceptance profile, $\theta_i$ is the location of the $i$th photoreceptor, $c(\theta)$ is the spatial contrast pattern of the visual world, and $\nu(t)$ is the time-dependent velocity. This formula simplifies to the formula in the 'Materials and methods' when $\nu(t)$ is time-independent. If $T$ is an invertible linear filter, then a more convenient representation of the photoreceptor signals is

$$U_i(t) = \mathscr{C}\left(\theta_i - \int_0^t dt'\, v(t')\right), \tag{3}$$

where $U_i = T^{-1} * V_i$, $\mathscr{C} = M * c$, and $*$ is the convolution operator (**Potters and Bialek, 1994**). We can rewrite the 2-point correlator in this representation as

$$v_e^{(2)}(t) = \int dt_1 \int dt_2 \kappa_{i,j}^{(2)}(t_1, t_2) U_i(t - t_1) U_j(t - t_2), \tag{4}$$

where

$$\kappa_{i,j}^{(2)}(t_1, t_2) \equiv \int dt_3 T(t_3) \int dt_4 T(t_4) k_{i,j}^{(2)}(t_1 - t_3, t_2 - t_4) \tag{5}$$

is the 2-point kernel that converts correlations in the $U$ variables to a velocity estimate.

Recall that we quantify the performance of visual motion estimators based on the mean squared error between the true and estimated velocities

$$\epsilon \equiv \left\langle \left(v_e^{(2)}(t) - v(t)\right)^2 \right\rangle = \sigma_v^2 - 2\langle v(t) v_e^{(2)}(t)\rangle + \left\langle \left(v_e^{(2)}(t)\right)^2 \right\rangle, \tag{6}$$

where $\sigma_\nu = 90°/s$ is the standard deviation of the velocity distribution. For estimators that are scaled to minimize their mean squared error ('Materials and methods'), this formula can be rewritten as

$$\epsilon = \sigma_v^2 \left(1 - r^2\right), \tag{7}$$

where

$$r = \frac{\langle v(t) v_e^{(2)}(t)\rangle}{\sqrt{\langle (v(t))^2 \rangle \left\langle \left(v_e^{(2)}(t)\right)^2\right\rangle}} = \frac{\langle v(t) v_e^{(2)}(t)\rangle}{\sigma_v \sqrt{\left\langle \left(v_e^{(2)}(t)\right)^2\right\rangle}} \tag{8}$$

is the correlation coefficient between the estimated and true velocities. Thus, minimizing the mean squared error is mathematically equivalent to maximizing the correlation coefficient if all motion estimators are correctly scaled. We find the correlation coefficient to be a more intuitive

error metric than the mean squared error, so many of our results will be presented in terms of correlation coefficients.

The numerator of the correlation coefficient is determined by the second-order statistics of the image ensemble,

$$
\langle v(t) v_e^{(2)}(t) \rangle = \int dt_1 \int dt_2 \kappa_{i,j}^{(2)}(t_1, t_2) \langle v(t) U_i(t-t_1) U_j(t-t_2) \rangle,
$$

$$
= \int dt_1 \int dt_2 \kappa_{i,j}^{(2)}(t_1, t_2) \langle v(t) \, \mathscr{C}^{(2)} \left( \Delta_{ij} + \int_{t_1}^{t_2} dt' \, v(t') \right) \rangle_v,
\tag{9}
$$

where $\Delta_{ij}$ is the angular separation between the $i$th and $j$th photoreceptors, and

$$
\mathscr{C}^{(2)}(\Delta) \equiv \langle \mathscr{C}(x) \, \mathscr{C}(x+\Delta) \rangle_{\mathscr{C}}
\tag{10}
$$

is the 2-point correlation function over the ensemble of spatially filtered natural scenes. Note that $\mathscr{C}^{(2)}$ is independent of $x$ because reasonable image ensembles are translationally invariant. Also note that the 2-point correlation function of filtered natural images is related to the correlation function of unfiltered images by

$$
\mathscr{C}^{(2)}(\Delta) = \int dx' M(x') \int dx'' M(x'') \langle c(x-x') c(x+\Delta-x'') \rangle = \left( (M \star M) \star C^{(2)} \right)(\Delta),
\tag{11}
$$

where $C^{(2)}(\Delta)$ is the correlation function of unfiltered images, and we've assumed that $M$ is a symmetric function. We model $M$ as Gaussian with FWHM of 5.7°, so $M \star M$ is also Gaussian with FWHM of $\sqrt{2} \times 5.7° = 8.1°$.

On the other hand, the denominator of the correlation coefficient is determined by fourth-order statistics of the image ensemble,

$$
\langle \left( v_e^{(2)}(t) \right)^2 \rangle = \int dt_1 \int dt_2 \int dt_3 \int dt_4 \kappa_{i,j}^{(2)}(t_1, t_2) \kappa_{i,j}^{(2)}(t_3, t_4)
$$
$$
\times \langle U_i(t-t_1) U_j(t-t_2) U_i(t-t_3) U_j(t-t_4) \rangle
$$
$$
= \int dt_1 \int dt_2 \int dt_3 \int dt_4 \kappa_{i,j}^{(2)}(t_1, t_2) \kappa_{i,j}^{(2)}(t_3, t_4)
$$
$$
\times \langle \mathscr{C}^{(4)} \left( \Delta_{ij} + \int_{t_1}^{t_2} dt' \, v(t'), \int_{t_1}^{t_3} dt' \, v(t'), \Delta_{ij} + \int_{t_1}^{t_4} dt' \, v(t') \right) \rangle_v,
\tag{12}
$$

where

$$
\mathscr{C}^{(4)}(\Delta_1, \Delta_2, \Delta_3) = \langle \mathscr{C}(x) \, \mathscr{C}(x+\Delta_1) \, \mathscr{C}(x+\Delta_2) \, \mathscr{C}(x+\Delta_3) \rangle_{\mathscr{C}}
\tag{13}
$$

is the 4-point correlation function of the ensemble of filtered natural images. Notice that the second argument of $\mathscr{C}^{(4)}$ in *Equation 12* lacks the additive factor of $\Delta_{ij}$ because $U_i(t-t_1)$ and $U_i(t-t_3)$ correspond to the same point in space. As above, $\mathscr{C}^{(4)}$ is related to the unfiltered 4-point correlation function through a fourfold application of the photoreceptor spatial acceptance filter.

The preceding analysis shows that only the second-order and fourth-order statistics of the natural image ensemble contribute to the correlation coefficient between an arbitrary 2-point correlator and the true velocity. The same quantities also determine the mean squared error. Thus, the second-order and fourth-order statistics of the image ensembles are the critical determinants of a 2-point correlator's motion estimation accuracy. Note that both the HRC and the motion energy model fall into this important class of visual motion estimators, so our analysis is also important for understanding visual motion estimation by vertebrates.

## Appendix 3

# Motion estimation without spatial correlations—the role of kurtosis on the accuracy of 2-point correlators.

In this section, we apply the results of Appendix 2 to the special case of normally distributed velocities and spatially uncorrelated image ensembles. This calculation reveals an important role for kurtosis in motion estimation, and we discuss how nonlinearities in the early visual system could cope with highly kurtotic naturalistic inputs.

In this section, we assume that the velocity is time-independent (i.e., $v(t) = v$) and normally distributed

$$P_v(v) = \frac{1}{\sqrt{2\pi\sigma_v^2}} e^{-v^2/(2\sigma_v^2)}. \tag{14}$$

We also assume that the image ensemble is spatially uncorrelated. By this, we mean that the luminance contrast at each point in space is statistically independent of the luminance contrast at all other points in space. Thus, the second-order correlation function is

$$\mathscr{C}^{(2)}(\Delta) = \sigma_C^2 \delta(\Delta), \tag{15}$$

where $\sigma_C$ is the standard deviation of the luminance contrast, and $\delta(\Delta)$ is the Dirac delta-function. The fourth-order correlation function is

$$\begin{aligned}\mathscr{C}^{(4)}(\Delta_1, \Delta_2, \Delta_3) = &\kappa_4 \sigma_C^4 \delta(\Delta_1)\delta(\Delta_2)\delta(\Delta_3) \\ &+ \sigma_C^4 (\delta(\Delta_1)\delta(\Delta_2 - \Delta_3) + \delta(\Delta_2)\delta(\Delta_1 - \Delta_3) + \delta(\Delta_3)\delta(\Delta_1 - \Delta_2)),\end{aligned} \tag{16}$$

where $k_4$ is the excess kurtosis of the contrast distribution. The excess kurtosis is zero for normally distributed contrasts. It can either be positive or negative for other contrast distributions. Note that we define the *kurtosis* of a probability distribution to be its fourth central moment normalized by the square of its second central moment. Thus, the kurtosis of a normal distribution is 3. We caution readers that some other sources use 'kurtosis' to refer to the excess kurtosis.

With these assumptions, the signal term represented by **Equation 9** is

$$\langle v(t) v_e^{(2)}(t) \rangle = \frac{\sigma_C^2 \Delta_{ij}}{\sqrt{2\pi\sigma_v^2}} \int dt_1 \int dt_2 \kappa_{i,j}^{(2)}(t_1, t_2) \frac{e^{-\Delta_{ij}^2/(2\sigma_v^2(t_2-t_1)^2)}}{(t_2-t_1)|t_2-t_1|}, \tag{17}$$

and the noise term represented by **Equation 12** is

$$\begin{aligned}\langle (v_e^{(2)}(t))^2 \rangle = &\frac{\sigma_C^4}{\Delta_{ij}\sqrt{2\pi\sigma_v^2}} \int dt_1 \int dt_2 \int dt_3 \int dt_4 \kappa_{i,j}^{(2)}(t_1,t_2)\kappa_{i,j}^{(2)}(t_3,t_4) \\ &\times \Big( e^{-\Delta_{ij}^2/(2\sigma_v^2(t_1-t_4)^2)} \delta((t_1-t_4)-(t_3-t_2)) + e^{-\Delta_{ij}^2/(2\sigma_v^2(t_1-t_2)^2)}\delta((t_1-t_2)-(t_3-t_4)) \\ &+ \kappa_4 \frac{|t_1-t_2|}{\Delta_{ij}} e^{-\Delta_{ij}^2/(2\sigma_v^2(t_1-t_2)^2)} \delta(t_3-t_1)\delta(t_4-t_2) \Big),\end{aligned} \tag{18}$$

where we've assumed that the 2-point correlator is mirror anti-symmetric,

$$\kappa_{i,j}^{(2)}(t_1,t_2) = -\kappa_{i,j}^{(2)}(t_2,t_1), \tag{19}$$

in order to ignore contributions from static signals. This mirror-symmetry assumption holds for the HRC and the motion energy model. Since the denominator of the correlation coefficient is set by $\sqrt{\langle (v_e^{(2)}(t))^2 \rangle}$, both the signal and the noise are proportional to $\sigma_C^2$. Thus, the only remaining dependence on the image ensemble is through the excess kurtosis. Note that

$$\frac{d\langle \left(v_e^{(2)}(t)\right)^2 \rangle}{d\kappa_4} = \frac{\sigma_C^4}{\Delta_{ij}^2 \sqrt{2\pi\sigma_v^2}} \int dt_1 \int dt_2 \left(\kappa_{ij}^{(2)}(t_1,t_2)\right)^2 |t_1 - t_2| e^{-\Delta_{ij}^2/\left(2\sigma_v^2(t_1-t_2)^2\right)} > 0. \tag{20}$$

Thus, the correlation coefficient is maximized by making $k_4$ as small as possible.

In conclusion, if the image ensemble is spatially uncorrelated (at second and fourth-order), then the image ensemble only affects the correlation coefficient between the velocity and a 2-point correlator through its kurtosis. The best accuracy is achieved when the kurtosis is minimized. In reality, the assumption that the image ensemble is spatially uncorrelated is clearly wrong. Natural images are strongly correlated, and even if they weren't, they'd become correlated once they are filtered by the photoreceptors' spatial acceptance filter. Nevertheless, *Figure 2E* empirically shows that introducing several front-end nonlinearities that decrease the kurtosis also improve the accuracy of naturalistic motion estimation. Thus, kurtosis provides a useful guide for the design of neuronal nonlinearities. On the other hand, *Figure 2D,E* demonstrate that it's too simplistic to assume that the kurtosis is the only relevant factor for the accuracy of a 2-point correlator. As we'll discuss in the next section, spatial correlations in the image ensemble also affect the accuracy of 2-point correlators.

## Appendix 4

### The HRC benefits from spatially correlated input signals.

When we applied a contrast-equalizing or binarizing nonlinearity to naturalistic inputs before evaluating the HRC, we found that both nonlinearities substantially improved the accuracy of the HRC (*Figure 2E*). Interestingly, contrast equalization improved the accuracy of the HRC more than binarization (*Figure 2E*), even though it produced outputs with greater kurtosis. The reason for this is that natural images are correlated (*Appendix figure 1*), and the accuracy of the HRC over a general image ensemble depends on the ensemble's spatial correlation structure (Appendix 2). Binarization attenuated spatial correlations more strongly than contrast equalization over the natural image ensemble (*Figure 2—figure supplement 1*), which leads us to hypothesize that correlations present in the natural image ensemble might benefit the HRC's performance. In Appendix 5 we will provide theoretical support for this idea. Here we begin with a less mathematical argument that also supports our hypothesis.

A comparison between the estimation performance of binarizing and equalizing front-end nonlinearities was complicated by the fact that the models produced outputs that differed in both their point statistics and their correlation structures. To gain more direct insight into how spatial correlations affect motion estimation performance, it would be helpful to compare front-end nonlinearity models that differ *only* through their output correlation structures. We implemented this comparison using a family of binarizing front-end nonlinearities that undergo multiple steps between +1 and −1 (*Appendix figure 2A*). Although these nonlinearities are not physiologically realistic, they are conceptually useful because they each produced a stimulus ensemble that minimized the kurtosis yet achieved distinct correlation structures (*Appendix figure 2B*). These nonlinearities thus allow us to assess directly whether spatial decorrelation of inputs degrades the motion estimation performance of the HRC. We found that each binarizing front-end nonlinearity model outperformed the original HRC (*Appendix figure 2C*). However, we found that the magnitude of the improvement decreased with the number of steps (*Appendix figure 2C*). Since spatial cross-correlations also decreased as a function of the number of steps (*Appendix figure 2B*), these results support our hypothesis that the correlations present in natural visual inputs aid the functionality of the standard HRC.

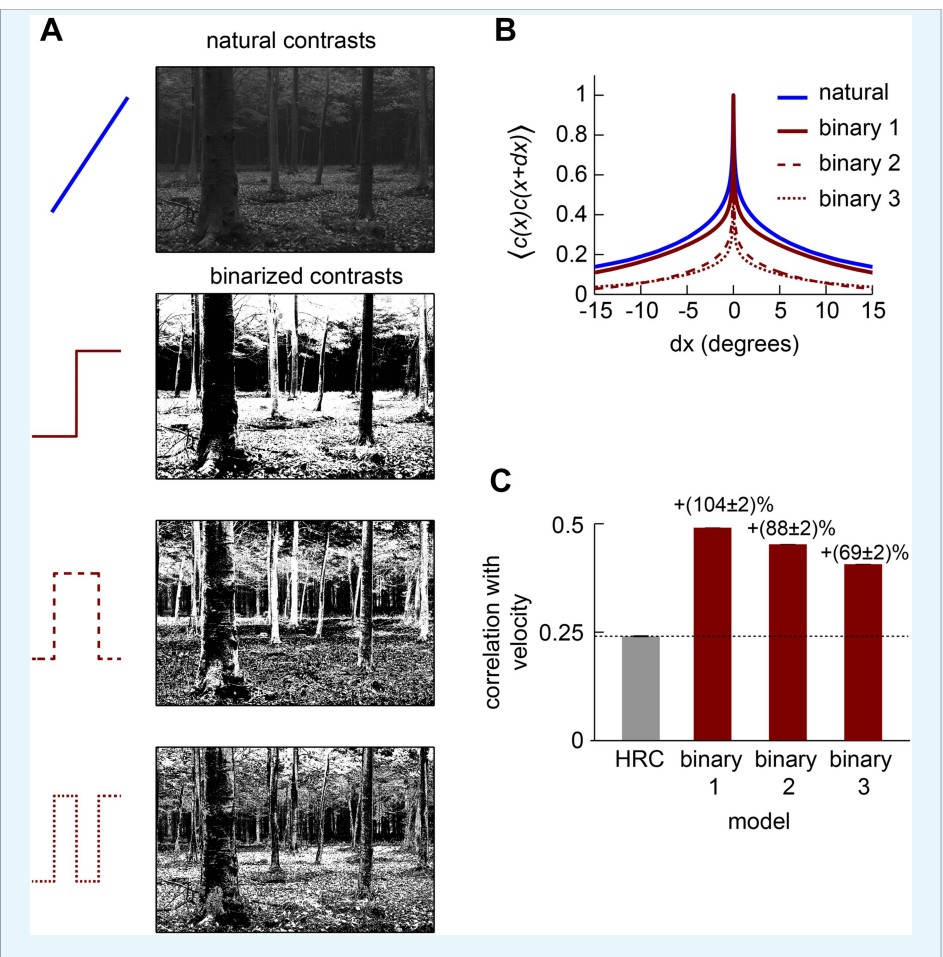

**Appendix figure 2**. Correlations in binarized natural images. (**A**) We transformed each image in the van Hateren natural image database (**van Hateren and van der Schaaf, 1998**) with several binarizing nonlinearities. To implement the simplest binarizing nonlinearity, we set all pixels to +1 or −1 depending on whether that pixel exceeded or fell below the median intensity in the image. For the nonlinearity with two steps, the thresholds were at the 25th and 75th intensity percentiles. For the nonlinearity with three steps, the thresholds were at the 25th, 50th, and 75th intensity percentiles. When a pixel intensity exactly equaled a threshold, we considered its value below threshold. Binary nonlinearities with a larger number of steps produced grainier images that indicate a spatial decorrelation of the transformed image. (**B**) We computed second-order spatial correlation functions across the nonlinearly transformed natural image ensemble. This confirmed that each step in the binarizing nonlinearity further decorrelated the image ensemble. (**C**) In addition to decreasing the spatial extent of correlations, a larger number of transitions also degraded the performance of the front-end nonlinearity model.

The HRC correlates two signals that are offset in space and differentially delayed in time. One intuition that researchers often apply to this computation is that the correlation operation effectively detects times when two signals that are offset in space and time are equal. However, a motion estimator that strictly obeyed this intuition would be agnostic to the spatial correlation structures present in the input signals, and our results show that the HRC is not (see also Appendix 5). Instead, the HRC also generates motion signals when its two input channels are imperfectly aligned, and these signals depend strongly on the correlation structure of the inputs (**Appendix figure 1D**). Our results thus show that the HRC's ability to detect imperfect coincidences contributes significantly to its performance as a motion estimator, as was suggested intuitively in Appendix 1.

## Appendix 5

### Motion estimation with Gaussian image statistics—the role of spatial correlations on the accuracy of 2-point correlators.

In this section, we apply the results of Appendix 2 to the special case of normally distributed velocities and normally distributed image ensembles. This model formalizes how spatial correlations in the natural world affect the accuracy of motion estimation by 2-point correlators and shows how spatial decorrelation can adversely affect estimation accuracy. For example, we'll show that the simplest HRC is unable to extract motion signals from high frequency components of the image ensemble, yet those components still lead to variability in the motion estimator. Thus, this HRC works best when the image ensemble is correlated in a manner that avoids high-frequency components in the signal, and spatial low-pass filtering at the photoreceptor level can help to eliminate the high-frequency image components that hurt the HRC's accuracy.

Here we use the same velocity distribution that we used in Appendix 3 (i.e., **Equation 14**). However, we now allow the two point correlation function to have arbitrary structure

$$\mathscr{C}^{(2)}(\Delta) = \sum_{k=0}^{\infty} S_k \cos(k\Delta),\tag{21}$$

where $S_k$ are the Fourier coefficients for $\mathscr{C}^{(2)}(\Delta)$, and we have noted that the image ensemble is $2\pi$-periodic. Note that $S_k$ is called the power spectrum of the image ensemble, and uncorrelated ensembles correspond to the special case where $S_k$ = constant. With these assumptions

$$\left\langle v(t)\, \mathscr{C}^{(2)}\left(\Delta_{ij} + \int_{t_1}^{t_2} dt'\ v(t')\right)\right\rangle_v = \sum_{k=0}^{\infty} S_k \langle v\cos\left(k\left(\Delta_{i,j} + v(t_2 - t_1)\right)\right)\rangle_v.\tag{22}$$

By evaluating the integral, we find that this velocity expectation is

$$\langle v\cos\left(k\left(\Delta_{i,j} + v(t_2 - t_1)\right)\right)\rangle_v = k(t_1 - t_2)\sigma_v^2 \sin\left(k\Delta_{ij}\right)e^{-\frac{1}{2}k^2(t_2 - t_1)^2\sigma_v^2}.\tag{23}$$

Thus, if we define

$$\gamma_k = k\sigma_v^2 \sin\left(k\Delta_{ij}\right)\int dt_1 \int dt_2 \kappa_{ij}^{(2)}(t_1, t_2)(t_1 - t_2)e^{-\frac{1}{2}k^2(t_2 - t_1)^2\sigma_v^2},\tag{24}$$

then

$$\langle v(t)v_e^{(2)}(t)\rangle = \sum_{k=0}^{\infty} \gamma_k S_k.\tag{25}$$

Each frequency component of the image ensemble linearly contributes to the correlation between the 2-point correlator's response and the velocity. The weight of each frequency component is determined by the structure of the 2-point correlator and the width of the velocity distribution.

We compute the fourth-order moment of the image ensemble using Wick's theorem for Gaussian moments, which says

$$\langle\,\mathscr{C}(x_1)\,\mathscr{C}(x_2)\,\mathscr{C}(x_3)\,\mathscr{C}(x_4)\rangle = \langle\,\mathscr{C}(x_1)\,\mathscr{C}(x_2)\rangle\langle\,\mathscr{C}(x_3)\,\mathscr{C}(x_4)\rangle + \langle\,\mathscr{C}(x_1)\,\mathscr{C}(x_3)\rangle\langle\,\mathscr{C}(x_2)\,\mathscr{C}(x_4)\rangle$$
$$+ \langle\,\mathscr{C}(x_1)\,\mathscr{C}(x_4)\rangle\langle\,\mathscr{C}(x_2)\,\mathscr{C}(x_3)\rangle.\tag{26}$$

This immediately implies that

$$\mathscr{C}^{(4)}(\Delta_1, \Delta_2, \Delta_3) = \mathscr{C}^{(2)}(\Delta_1)\,\mathscr{C}^{(2)}(\Delta_3 - \Delta_2) + \mathscr{C}^{(2)}(\Delta_2)\,\mathscr{C}^{(2)}(\Delta_3 - \Delta_1)$$
$$+ \mathscr{C}^{(2)}(\Delta_3)\,\mathscr{C}^{(2)}(\Delta_2 - \Delta_1). \tag{27}$$

Once again, it's convenient to rewrite this expression in the Fourier domain

$$\mathscr{C}^{(4)}(\Delta_1, \Delta_2, \Delta_3) = \sum_{k_1=0}^{\infty} \sum_{k_2=0}^{\infty} S_{k_1} S_{k_2} \big( \cos(k_1 \Delta_1) \cos(k_2(\Delta_3 - \Delta_2)) $$
$$+ \cos(k_1 \Delta_2) \cos(k_2(\Delta_3 - \Delta_1)) + \cos(k_1 \Delta_3) \cos(k_2(\Delta_2 - \Delta_1))\big). \tag{28}$$

With these assumptions

$$\left\langle \mathscr{C}^{(4)}\left(\Delta_{ij} + \int_{t_1}^{t_2} dt'\, v(t'), \int_{t_1}^{t_3} dt'\, v(t'), \Delta_{ij} + \int_{t_1}^{t_4} dt'\, v(t')\right)\right\rangle_v$$
$$= \sum_{k_1=0}^{\infty} \sum_{k_2=0}^{\infty} S_{k_1} S_{k_2} \big\langle \cos\big(k_1\big(\Delta_{ij} + v(t_2 - t_1)\big)\big) \cos\big(k_2\big(\Delta_{ij} + v(t_4 - t_3)\big)\big)$$
$$+ \cos(k_1 v(t_3 - t_1)) \cos(k_2 v(t_4 - t_2))$$
$$+ \cos\big(k_1\big(\Delta_{ij} + v(t_4 - t_1)\big)\big) \cos\big(k_2\big(\Delta_{ij} + v(t_2 - t_3)\big)\big)\big\rangle_v. \tag{29}$$

We evaluate the expectations over velocity by noting that each has the form

$$\langle \cos(k_1(\Delta + v\delta_1))\,\cos(k_2(\Delta + v\delta_2))\rangle_v = \frac{1}{2} e^{-\frac{1}{2}(k_1\delta_1 + k_2\delta_2)^2 \sigma_v^2} \big( \cos(\Delta(k_1 + k_2))$$
$$+ e^{2k_1 k_2 \delta_1 \delta_2 \sigma_v^2} \cos(\Delta(k_1 - k_2))\big) \tag{30}$$

for some spatial offset $\Delta$ and temporal offsets $\{\delta_1,\ \delta_2\}$. Thus, if we define

$$\Gamma_{k_1 k_2} = \int dt_1 \int dt_2 \kappa_{ij}^{(2)}(t_1, t_2) \int dt_3 \int dt_4 \kappa_{ij}^{(2)}(t_3, t_4)$$

$$\left(\frac{1}{2} e^{-\frac{1}{2}(k_1(t_2 - t_1) + k_2(t_4 - t_3))^2 \sigma_v^2} \left(\cos\big(\Delta_{ij}(k_1 + k_2)\big) + e^{2k_1 k_2(t_2 - t_1)(t_4 - t_3)\sigma_v^2} \cos\big(\Delta_{ij}(k_1 - k_2)\big)\right)\right.$$

$$+ \frac{1}{2} e^{-\frac{1}{2}(k_1(t_4 - t_1) + k_2(t_2 - t_3))^2 \sigma_v^2} \left(\cos\big(\Delta_{ij}(k_1 + k_2)\big) + e^{2k_1 k_2(t_4 - t_1)(t_2 - t_3)\sigma_v^2} \cos\big(\Delta_{ij}(k_1 - k_2)\big)\right)$$

$$\left. + \frac{1}{2} e^{-\frac{1}{2}(k_1(t_3 - t_1) + k_2(t_4 - t_2))^2 \sigma_v^2} \left(1 + e^{2k_1 k_2(t_3 - t_1)(t_4 - t_2)\sigma_v^2}\right)\right), \tag{31}$$

then

$$\left\langle \left(v_e^{(2)}(t)\right)^2 \right\rangle = \sum_{k_1=0}^{\infty} \sum_{k_2=0}^{\infty} \Gamma_{k_1 k_2} S_{k_1} S_{k_2}. \tag{32}$$

Power spectrum components contribute to the 2-point correlator's variance quadratically.

Putting these pieces together, the expected squared error achieved by a 2-point correlator is a quadratic function of the power spectrum

$$\epsilon = \sigma_v^2 - 2\sum_{k=0}^{\infty} \gamma_k S_k + \sum_{k_1=0}^{\infty} \sum_{k_2=0}^{\infty} \Gamma_{k_1 k_2} S_{k_1} S_{k_2}. \tag{33}$$

We're interested to know whether spatial correlations can enhance the accuracy of the 2-point correlator. This will be the case unless a uniform power spectrum minimizes $\epsilon$. Note that every physically meaningful power spectrum is non-negative

$$S_k \geq 0. \tag{34}$$

Thus, the minimum of $\epsilon$ either occurs at an extremum point or on the boundary of admissible solutions. If the minimum occurs on the boundary, then a subset of the $S_k$ are exactly equal to zero. In particular, the power spectrum would not be constant, which implies that the image ensemble would be spatially correlated. At an extremum point, we must find

$$0 = \frac{\partial \epsilon}{\partial S_k} = -2\gamma_k + 2 \sum_{k'=0}^{\infty} \Gamma_{kk'} S_{k'} \tag{35}$$

for every $k$. A uniform power spectrum can only satisfy this condition if

$$\gamma_k = \beta \sum_{k'=0}^{\infty} \Gamma_{kk'}, \tag{36}$$

where $\beta > 0$ is the (constant) value of each power spectrum component. This is generally not the case, so correlations exist that would help typical 2-point correlators.

For example, the simplest HRC, which replaces the low-pass and high-pass filters with pure time delays, is

$$\tilde{R} = A(U_1(t - \tau)U_2(t) - U_1(t)U_2(t - \tau)), \tag{37}$$

where $A$ is a constant with units of °/s. For this model,

$$\kappa_{1,2}^{(2)}(t_1, t_2) = A(\delta(t_1 - \tau)\delta(t_2) - \delta(t_1)\delta(t_2 - \tau)). \tag{38}$$

Substituting this expression into the above formulas, we find

$$\gamma_k = 2Ak\tau\sigma_v^2 \sin(k\Delta_0) e^{-\frac{1}{2}k^2\tau^2\sigma_v^2} \tag{39}$$

and

$$\Gamma_{k_1 k_2} = A^2 \left( 3 \sin(k_1\Delta_0) \sin(k_2\Delta_0) \left( e^{-\frac{1}{2}(k_1 - k_2)^2\tau^2\sigma_v^2} - e^{-\frac{1}{2}(k_1 + k_2)^2\tau^2\sigma_v^2} \right) \right.$$
$$\left. + (1 - \cos(k_1\Delta_0) \cos(k_2\Delta_0)) \left( 2 - e^{-\frac{1}{2}(k_1 - k_2)^2\tau^2\sigma_v^2} - e^{-\frac{1}{2}(k_1 + k_2)^2\tau^2\sigma_v^2} \right) \right), \tag{40}$$

where $\Delta_0$ is the spacing between adjacent photoreceptors. Note that

$$\lim_{k \to \infty} \gamma_k = 0. \tag{41}$$

On the other hand,

$$\lim_{k_2 \gg k_1} \Gamma_{k_1 k_2} = 2A^2(1 - \cos(k_1\Delta_0)\cos(k_2\Delta_0)). \tag{42}$$

This does not approach zero, even for large values of $k_1$. Therefore, $\sum_{k'=0}^{\infty} \Gamma_{kk'}$ diverges and $\gamma_k \neq \beta \sum_{k'=0}^{\infty} \Gamma_{kk'}$. In this model, high frequency components lack signal but contribute noise. It's helpful if these frequency components are absent from the image ensemble. Future work should more fully investigate the role of spatial correlations in naturalistic motion estimation.

## Appendix 6

### Front-end nonlinearities give the HRC access to higher-order correlations.

The response of the front-end nonlinearity model to a 3-point glider stimulus is determined by the higher-order correlations that it detects in the stimulus. Furthermore, we argued in Appendix 1 and *Figure 1I* that higher-order correlations can contribute to the accuracy of visual motion estimators. We now describe how front-end nonlinearities provide pair-correlation mechanisms with access to certain types of higher-order correlations.

We suppose that the front-end nonlinearity, denoted $h$, has a power series expansion:

$$h(x) = \sum_{n=0}^{\infty} h_n x^n. \tag{43}$$

Then the cross-correlation function between two non-linearly transformed input streams, denoted $y_1$ and $y_2$, is

$$\langle y_1(t) y_2(t+\tau) \rangle = \langle h(V_1(t)) h(V_2(t+\tau)) \rangle = \sum_{n,m=0}^{\infty} h_n h_m \langle V_1^n(t) V_2^m(t+\tau) \rangle, \tag{44}$$

where $V_1$ and $V_2$ are linear photoreceptor signals. This substitution explicitly demonstrates that the front-end nonlinear transformation enables pair correlation mechanisms to incorporate higher-order correlations of the form $\langle V_1^n(t) V_2^m(t+\tau) \rangle$. The choice of nonlinearity specifies the expansion coefficients, $h_n$, which in turn determines the pattern of higher-order correlations that the pair correlator incorporates into its velocity estimate. For example, sensitivity to odd-ordered correlations demands that $h_n$ be large for some even values of $n$. These expansion coefficients would manifest themselves in the structure of the front-end nonlinearity as asymmetries between positive and negative contrasts, but strong asymmetries were not needed to eliminate kurtosis in natural image ensembles (*Figure 2C*). Inversely, one could use this equation to determine whether a set of expansion coefficients exist that would implement a desired series of multipoint correlators. The preceding argument implies that strongly asymmetric front-end nonlinearities would be needed to account for the 3-point glider responses.

## Appendix 7

### Expansion of the weighted 4-quadrant model.

In this Appendix, we rewrite the weighted 4-quadrant model in a basis that isolates its dependence on 2-point correlations, on higher-even-ordered correlations, and on two types of odd-ordered correlations. In Appendix 8, we'll discuss the motion estimation performance of the weighted 4-quadrant model in this basis in order to gain insight into why performance-optimized weighted 4-quadrant models also predict 3-point glider responses that resemble *Drosophila* behavior.

The weighted 4-quadrant model supposes that the input signals are segregated into four separate streams:

$$
\begin{aligned}
Q_{++} &= [f{\star}V_1]_+[g{\star}V_2]_+ - [g{\star}V_1]_+[f{\star}V_2]_+ \\
Q_{+-} &= [f{\star}V_1]_+[g{\star}V_2]_- - [g{\star}V_1]_-[f{\star}V_2]_+ \\
Q_{-+} &= [f{\star}V_1]_-[g{\star}V_2]_+ - [g{\star}V_1]_+[f{\star}V_2]_- \\
Q_{--} &= [f{\star}V_1]_-[g{\star}V_2]_- - [g{\star}V_1]_-[f{\star}V_2]_-
\end{aligned}
\tag{45}
$$

where $Q_{ab}$ denotes the ($ab$) quadrant for $a, b \in \{+, -\}$, $[x]_+$ is $x$ for $x > 0$ and is zero otherwise, and $[x]_-$ is $x$ for $x < 0$ and is zero otherwise. The HRC is equal to

$$
R = Q_{++} + Q_{+-} + Q_{-+} + Q_{--}.
\tag{46}
$$

More generally, we suppose that *Drosophila* could estimate motion as any linear combination of these signals, and we define the weighted 4-quadrant model as

$$
Q = w_{++}^{(Q)} Q_{++} + w_{+-}^{(Q)} Q_{+-} + w_{-+}^{(Q)} Q_{-+} + w_{--}^{(Q)} Q_{--},
\tag{47}
$$

where $w_{++}^{(Q)}$, $w_{+-}^{(Q)}$, $w_{-+}^{(Q)}$, and $w_{--}^{(Q)}$ are linear weighting coefficients that specify the computation performed by the model. Since this section, and the next two, focus entirely on the weighted 4-quadrant model, we simplify notation by dropping the superscript ($Q$).

The weighted 4-quadrant model can be rewritten in an alternate form that facilitates an understanding of how various correlation types contribute to its motion estimates. We begin by noting that

$$
[x]_+ = \frac{x}{2}(1 + \mathrm{sgn}(x)),
\tag{48}
$$

$$
[x]_- = \frac{x}{2}(1 - \mathrm{sgn}(x)),
\tag{49}
$$

where $\mathrm{sgn}(x)$ is $+1$ for positive arguments and $-1$ for negative arguments. We thus see that

$$
\begin{aligned}
Q_{ab} &= [f{\star}V_1]_a[g{\star}V_2]_b - [g{\star}V_1]_b[f{\star}V_2]_a \\
&= \frac{(f{\star}V_1)(g{\star}V_2)}{4}(1 + a\,\mathrm{sgn}(f{\star}V_1) + b\,\mathrm{sgn}(g{\star}V_2) + ab\,\mathrm{sgn}(f{\star}V_1)\mathrm{sgn}(g{\star}V_2)) \\
&\quad - \frac{(g{\star}V_1)(f{\star}V_2)}{4}(1 + b\,\mathrm{sgn}(g{\star}V_1) + a\,\mathrm{sgn}(f{\star}V_2) + ab\,\mathrm{sgn}(g{\star}V_1)\mathrm{sgn}(f{\star}V_2)).
\end{aligned}
\tag{50}
$$

Therefore, the complete weighted 4-quadrant model is

$$
\begin{aligned}
Q = &\frac{w_{++} + w_{+-} + w_{-+} + w_{--}}{4}((f{\star}V_1)(g{\star}V_2) - (g{\star}V_1)(f{\star}V_2)) \\
&+ \frac{w_{++} + w_{+-} - w_{-+} - w_{--}}{4}((f{\star}V_1)\mathrm{sgn}(f{\star}V_1)(g{\star}V_2) - (g{\star}V_1)(f{\star}V_2)\mathrm{sgn}(f{\star}V_2)) \\
&+ \frac{w_{++} - w_{+-} + w_{-+} - w_{--}}{4}((f{\star}V_1)(g{\star}V_2)\mathrm{sgn}(g{\star}V_2) - (g{\star}V_1)\mathrm{sgn}(g{\star}V_1)(f{\star}V_2)) \\
&+ \frac{w_{++} - w_{+-} - w_{-+} + w_{--}}{4}((f{\star}V_1)\mathrm{sgn}(f{\star}V_1)(g{\star}V_2)\mathrm{sgn}(g{\star}V_2) \\
&- (g{\star}V_1)\mathrm{sgn}(g{\star}V_1)(f{\star}V_2)\mathrm{sgn}(f{\star}V_2))
\end{aligned}
\tag{51}
$$

This expression for the weighted 4-quadrant model groups the four weighting coefficients into four alternate terms. The first term is proportional to a standard HRC, which computes second-order correlations. We denote its associated coefficient as

$$w_{\text{even}=2} = \frac{1}{4}(w_{++} + w_{+-} + w_{-+} + w_{--}). \tag{52}$$

The second and third terms invert sign and retain magnitude under contrast inversion. Therefore, they only compute odd-ordered correlations:

$$w_{\text{odd}} = \frac{1}{4}(w_{++} + w_{+-} - w_{-+} - w_{--}), \tag{53}$$

$$w_{\text{odd}\star} = \frac{1}{4}(w_{++} - w_{+-} + w_{-+} - w_{--}). \tag{54}$$

The fourth term is unaffected by contrast inversion. Thus, it only computes even-ordered correlations. We'll soon see that the lowest-order contribution from this term is fourth-order, so we denote its coefficient as

$$w_{\text{even}>2} = \frac{1}{4}(w_{++} - w_{+-} - w_{-+} + w_{--}). \tag{55}$$

These four coefficients define the correlational basis considered in **Figure 3—figure supplement 1**. For example, note that **Figure 3—figure supplement 1A** shows the transformation defined by **Equations 52–55**.

Because $\text{sgn}(x)$ is a non-analytic function, it is still somewhat opaque how the weighted 4-quadrant model relates to specific higher-order correlations in the visual stimulus. We thus rewrite $\text{sgn}(x)$ as the limit of an analytic function:

$$\text{sgn}(x) = \lim_{\beta \to \infty} \text{erf}(\beta x), \tag{56}$$

where

$$\text{erf}(x) = \frac{2}{\sqrt{\pi}} \int_0^x dy \, e^{-y^2} \tag{57}$$

is the Gauss error function. The Gauss error function is entire, which means that it has a power series expansion for any value $x$. Also note that since real biological nonlinearities are not infinitely sharp, a more realistic weighted 4-quadrant model would fix $\beta$ at a finite value. We thus consider the follow approximation,

$$\text{sgn}(x) \approx \frac{2}{\sqrt{\pi}} \sum_{n=0}^{\infty} \frac{(-1)^n (\beta x)^{2n+1}}{n!(2n+1)} = \frac{2}{\sqrt{\pi}}\left(\beta x - \frac{(\beta x)^3}{3} + O\left((\beta x)^5\right)\right), \tag{58}$$

where $\beta \in (0, \infty)$. Although high-order terms might not be negligible in this expansion, the contributions of low-order correlations to visual motion estimation are set by low-order terms. In particular, the contributions of second, third, and fourth-order correlations to the weighted four quadrant model are determined by the leading terms in the expansion,

$$
\begin{aligned}
F = &\, w_{\text{even}=2}\left((f \star V_1)(g \star V_2) - (g \star V_1)(f \star V_2)\right) \\
&+ w_{\text{odd}}\frac{2\beta}{\sqrt{\pi}}\left((f \star V_1)^2(g \star V_2) - (g \star V_1)(f \star V_2)^2\right) \\
&+ w_{\text{odd}\star}\frac{2\beta}{\sqrt{\pi}}\left((f \star V_1)(g \star V_2)^2 - (g \star V_1)^2(f \star V_2)\right) \\
&+ w_{\text{even}>2}\frac{4\beta^2}{\pi}\left((f \star V_1)^2(g \star V_2)^2 - (g \star V_1)^2(f \star V_2)^2\right) + O(\beta^3 V^5).
\end{aligned} \tag{59}
$$

Thus, the third-order term associated with $w_{odd}$ squares the low-pass filtered signal and might help to account for light–dark asymmetries in the low-pass filtered signal. The third-order term associated with $w_{odd*}$ squares the high-pass filtered signal. Finally, note that this formula confirms that the lowest-order term associated with $w_{even>2}$ is fourth-order.

## Appendix 8

### The weighted 4-quadrant model improves motion estimation with odd-ordered correlations.

In the main text we quantitatively characterized the weighted 4-quadrant model by discussing its accuracy given various subsets of the four quadrants (**Figure 3C**). Here we consider the performance of the weighted 4-quadrant model in the correlational basis defined in Appendix 7 and **Figure 3—figure supplement 1A**. These results lead to a simple interpretation of the computation performed by performance optimized weighted 4-quadrant models.

Models that oriented all of their weight along the even = 2 axis outperformed models that focused their weight along any other correlational axis (**Figure 3—figure supplement 1B**). This reinforces the foremost importance of second-order correlations for motion estimation. In isolation, odd-ordered correlations were weaker predictors of motion than second-order correlations (**Figure 3—figure supplement 1B**). Nevertheless, the odd class well complemented the HRC, and the full accuracy of the weighted 4-quadrant model was obtained by linearly combining the even = 2 and odd correlation classes (*best 2 bar*, **Figure 3—figure supplement 1B**). This result suggests that the weighted 4-quadrant model has two relevant dimensions. In particular, accurate models combine an HRC with odd-ordered correlations that account for statistical light–dark asymmetries in the HRC's low-pass filtered branch.

Since the weighted 4-quadrant model only has four parameters, it's possible to exhaustively study its parameter dependence. We have in mind models that are correctly scaled, in which case the mean squared error is determined by the correlation coefficient (Appendix 2). Since the value of the correlation coefficient is unchanged when all four weighting coefficients are scaled by the same positive factor, it suffices to consider weighting coefficients drawn from the 3-sphere, such that $w_{++}^2 + w_{+-}^2 + w_{-+}^2 + w_{--}^2 = 1$. Because the 3-sphere has a finite volume, we were able to densely sample the correlation coefficient for all parameter values (**Appendix figure 3**). This function has one global maximum, corresponding to the optimal weight vector discussed in the main text. Its global minimum occurs on the polar opposite side of the 3-sphere, where the weighted 4-quadrant model is most strongly anti-correlated with the velocity. More generally, correlation coefficients corresponding to model parameters on opposite poles of the 3-sphere always have the same magnitude and opposite sign. Both models explain the same amount of variance about the velocity, and they become equivalent after they're correctly scaled. Thus, we henceforth focus our discussion on the hemisphere where the correlation coefficient was positive.

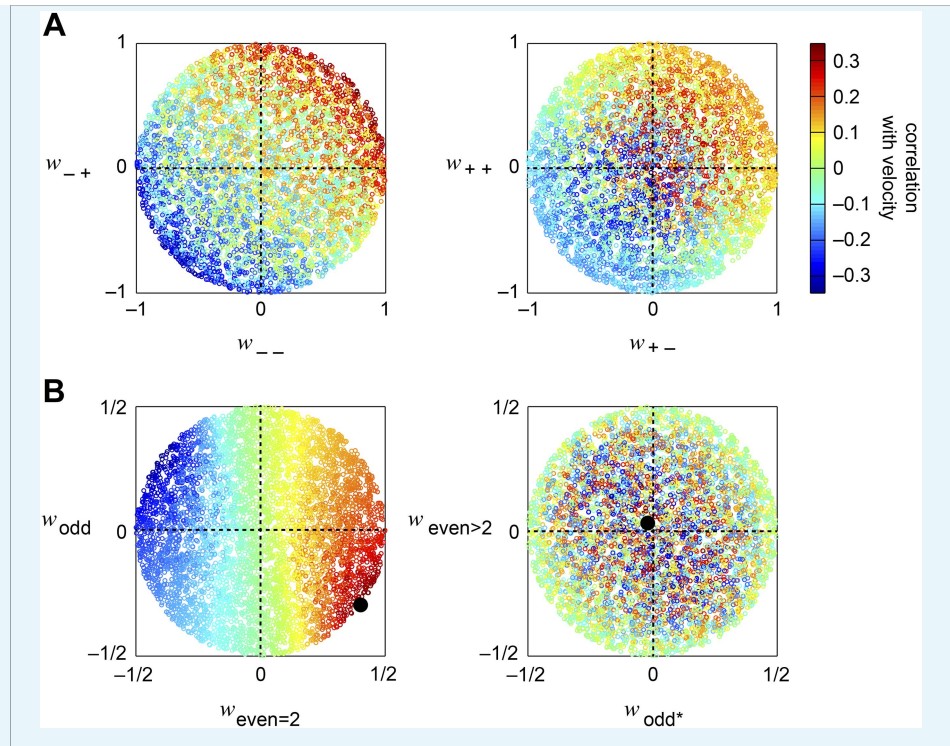

**Appendix figure 3**. Accuracy of the weighted 4-quadrant model across model parameters. (**A**, **B**) We computed the correlation coefficient between the velocity and the response of the weighted 4-quadrant model for all possible sets of model parameters. Since rescaling the weight vector does not affect the correlation coefficient, we assumed that all model parameters satisfy $\sum_{a,b\in\{+,-\}} (w_{ab}^{(Q)})^2 = 1$. We color-coded each set of model parameters by its accuracy and projected the parameter space onto various subspaces. (**A**) We first examined the quadrant basis by projecting onto the {(− −), (− +)} (*left*) and {(+ −), (+ +)} (*right*) subspaces. (**B**) We next examined the correlational basis by projecting onto the {even = 2, odd} (*left*) and {odd*, even >2} (*right*) subspaces. These project into different linear combinations of the original quadrant weightings. One of the projections is the pure HRC (even = 2), while the other projections contain only odd correlations, of two different types (odd and odd*), or only even correlations of order greater than 2 (even >2). These projections show that accurate weighted 4-quadrant models always put positive weight into 2-point correlations and negative weight into odd-ordered correlations. Note that the glider responses predicted by the weighted 4-quadrant model mirror this pattern (**Figure 3D**).

Weighted 4-quadrant models were most accurate when $w_{-+}$ and $w_{--}$ were large (**Appendix figure 3A**, *left*) and $w_{++}$ and $w_{-+}$ were small (**Appendix figure 3A**, *right*). In the correlational basis, the HRC is the model with maximum weight in $w_{even=2}$ and with zero weight in $w_{odd}$, $w_{odd*}$, and $w_{even>2}$. Thus, this basis makes it easy to compare the accuracy of the HRC to other weighted 4-quadrant models (**Appendix figure 3B**). Furthermore, this basis clearly sorts the weighted 4-quadrant models according to their accuracy and confirms that that the accuracy of a weighted 4-quadrant model is largely determined by $w_{even=2}$ and $w_{odd}$ (**Appendix figure 3B**, *left*). Higher even-ordered correlations and odd-ordered correlations that account for light–dark asymmetries in the high-pass filtered visual signals did not contribute prominently to the accuracy of the weighted 4-quadrant model (**Appendix figure 3B**, *right*). Interestingly, **Appendix figure 3A** shows that there is a diversity of ways to combine the four quadrants in order to improve the accuracy of the HRC, which translates into a diversity of correlational responses (**Appendix figure 3B**). Similarly, the HRC is only one of many models that achieve a comparable level of accuracy. Every other motion estimator that achieves the HRC's performance level incorporates higher-order correlations into its estimate.

## Appendix 9

### The weighted 4-quadrant model in the basis of PCs.

PCA is a popular method to reduce the dimensionality of neural population recordings. In this section, we conceptualize the four quadrants as a small neural population and study how each PC accounts for variance in the system and contributes to motion estimation. We show that most of the weighted 4-quadrant model's variance is due to two of the four PCs. Interestingly, most of this variance is not velocity-related, and we show that the two low-variance PCs are the ones that dominate motion estimation.

We began by directly applying PCA to the weighted 4-quadrant model. We computed the 4 × 4 covariance matrix of the four quadrants over the ensemble of simulated motions (**Appendix figure 4A**). The eigenvectors of the covariance matrix are called the PCs (**Appendix figure 4B**), and the associated eigenvalues specify the amount of variance accounted for by each PC (**Appendix figure 4C**). We found that the first two PCs accounted for 86.3% of the variance, whereas the third and fourth PCs each contributed about 7% of the variance (**Appendix figure 4C**). The high-variance eigenvectors roughly corresponded to a sum and a difference of the (+ +) and (+ −) quadrants, whereas the low-variance PCs roughly corresponded to a sum and a difference of the (− +) and (− −) quadrants (**Appendix figure 4B**). The (− −) and (− +) quadrants best facilitated motion estimation (**Figure 3C**). Thus, the low-variance PCs were most important for motion estimation.

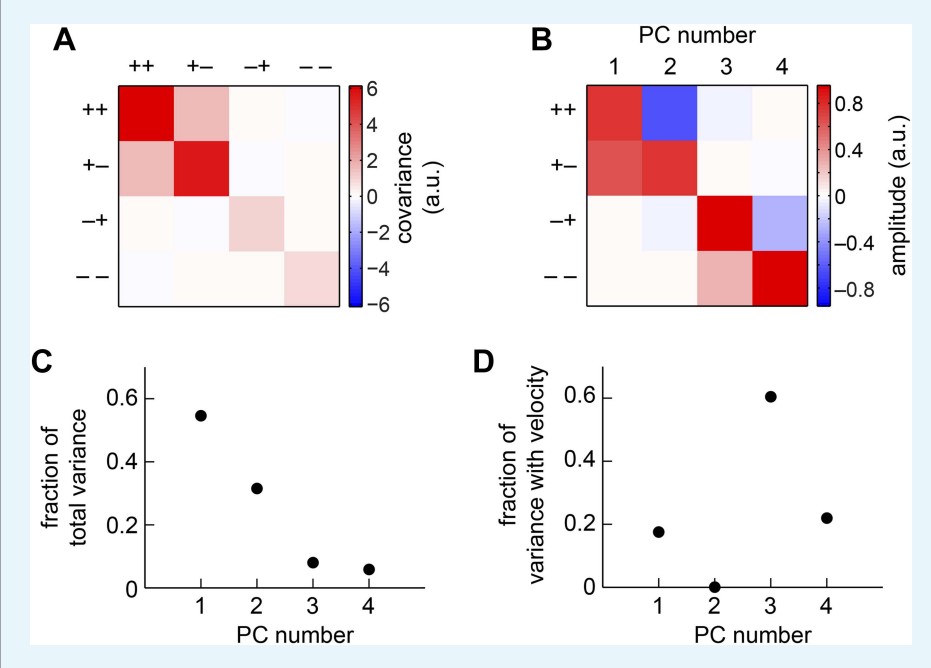

**Appendix figure 4**. The weighted 4-quadrant model in the basis of principal components (PCs). (**A**) We computed the covariance matrix of quadrant responses across the simulated ensemble of naturalistic motions. (**B**) The eigenvectors of the covariance matrix are called PCs. Signals from the (+ +) and (+ −) quadrants primarily comprised the first two PCs, whereas the (− +) and (− −) components comprised the third and fourth PCs. (**C**) The first two PCs accounted for the vast majority of the weighted 4-quadrant model's response variance. (**D**) Each member of the ensemble of naturalistic motions comprised a velocity and a natural image, and both components contributed variance to the model response. Although the first two PCs accounted for most of the variance, little of that variance was associated with the velocity of motion. Instead, the third and fourth PCs best aided motion estimation, because they contributed the vast majority of the velocity-associated variance.

This result is counter to one's usual intuition, but it is a straightforward consequence of the mathematics of linear regression and PCA. We want to linearly combine the PC signals to best predict the velocity:

$$\beta = \operatorname{argmin}\langle (v - \beta^T x)^2 \rangle, \tag{60}$$

where $\beta$ is a four-dimensional column vector of weights, $v$ denotes the velocity, the superscript $T$ denotes the matrix transpose, and $x$ is the 4-vector of PC signals. The solution to this problem is well-known from the theory of linear regression:

$$\beta = M^{-1} U, \tag{61}$$

where $M_{ij} = \langle x_i x_j \rangle$ is the covariance matrix of the predictors, and $U_i = \langle v x_i \rangle$ is the covariance of each predictor with the velocity. In practice, we estimate these expectations from the empirical data, and PCs are uncorrelated over the naturalistic motion ensemble by construction

$$M_{ij} = \lambda_i \delta_{ij}, \tag{62}$$

where $\lambda_i$ is the variance associated with $i$th PC, and $\delta_{ij}$ is the Kronecker $\delta$-function. Thus,

$$\beta_i = \frac{\langle v x_i \rangle}{\langle x_i^2 \rangle} = \frac{\sigma_v \sqrt{\lambda_i} r_i}{\lambda_i} = \frac{\sigma_v r_i}{\sqrt{\lambda_i}}, \tag{63}$$

where $\sigma_v$ is the standard deviation of the velocity signal, and $r_i$ is the correlation coefficient between the velocity and the $i$th PC.

It is also easy to calculate the correlation coefficient between the true velocity and the estimated velocity. First note that

$$\langle v \beta^T x \rangle = \beta^T \langle v x \rangle = \sum_i \frac{\sigma_v r_i}{\sqrt{\lambda_i}} \sigma_v \sqrt{\lambda_i} r_i = \sigma_v^2 \sum_i r_i^2, \tag{64}$$

$$\langle (\beta^T x)^2 \rangle = \sum_{i,j} \beta_i \beta_j \langle x_i x_j \rangle = \sum_i \frac{\sigma_v^2 r_i^2}{\lambda_i} \lambda_i = \sigma_v^2 \sum_i r_i^2. \tag{65}$$

Thus the square of the correlation coefficient between the true and estimated velocities is

$$r^2 = \frac{(\langle v \beta^T x \rangle)^2}{\langle v^2 \rangle \langle (\beta^T x)^2 \rangle} = \sum_i r_i^2. \tag{66}$$

Because the PCs are uncorrelated, each contributes independently to the motion estimator's accuracy. The amount that each PC contributes to the estimation accuracy is determined by its correlation with the velocity, and all dependence on the total amount of variance associated with the PC has dropped out entirely. These conclusions are also true when we look at the squared error directly

$$\epsilon = \langle (v - \beta^T x)^2 \rangle = \sigma_v^2 + \langle (\beta^T x)^2 \rangle - 2 \langle v \beta^T x \rangle = \sigma_v^2 \left( 1 - \sum_i r_i^2 \right). \tag{67}$$

As would be expected from this formula, the third and fourth PCs account for much more of the velocity-associated variance than the first and second PCs (**Appendix figure 4D**). Nevertheless, the first PC does account for a significant portion of the velocity-associated variance (**Appendix figure 4D**), so the basis of PCs does not fully reveal the structure that was apparent in the correlational basis (Appendix 8).

## Appendix 10

### Novel use of low-order signatures for motion estimation.

The non-multiplicative nonlinearity model (**Figure 4A**) relaxed the assumption that *Drosophila*'s motion estimator multiplies its inputs and substantially improved the accuracy of visual motion estimation (**Figure 4E**). Surprisingly, the non-multiplicative nonlinearity model slightly outperformed the HRC when we parameterized it as a second-order polynomial (**Figure 4—figure supplement 2**). This indicates that there are useful low-order correlations that the HRC neglects. In this section, we will explain how visual motion estimators can sometimes productively incorporate computational signatures that do not nonlinearly combine signals across space.

This section considers computational signatures that clash harshly with our usual intuition for visual motion estimation, and we need to unpack *how* the motion estimator in **Figure 4—figure supplement 2** works before we can understand *why* it works. The observed improvement results from a linear combination of the HRC

$$R = (f \star V_1)(g \star V_2) - (g \star V_1)(f \star V_2) \tag{68}$$

with a linear transformation of the photoreceptor signals

$$L = g \star V_1 - g \star V_2. \tag{69}$$

We thus must consider the motion estimator

$$v_e^{(\mathrm{low})} = \beta_R R + \beta_L L, \tag{70}$$

where $\beta_R$ and $\beta_L$ are the weighting coefficients that minimize the mean-squared error. Note that $L$ linearly combines signals from multiple points in space. Like the HRC, it is mirror anti-symmetric:

$$\{V_1(t), V_2(t)\} \mapsto \{V_2(t), V_1(t)\} \Rightarrow L \mapsto -L. \tag{71}$$

It is useful to take a detour to abstractly consider how motion estimation performance depends on the joint statistics of $R$, $L$, and the velocity of motion, $v$. All three of these quantities are zero mean. We denote their variances as

$$\sigma_R^2 = \langle R^2 \rangle, \ \ \sigma_L^2 = \langle L^2 \rangle, \ \ \sigma_v^2 = \langle v^2 \rangle \tag{72}$$

and their cross-correlation coefficients as

$$r^{(R)} = \frac{\langle vR \rangle}{\sigma_v \sigma_R}, \ \ r^{(L)} = \frac{\langle vL \rangle}{\sigma_v \sigma_L}, \ \ c^{(RL)} = \frac{\langle RL \rangle}{\sigma_R \sigma_L}. \tag{73}$$

The optimal weighting coefficients are determined by these quantities (see **Equation 61**):

$$\beta_R = \frac{\sigma_v \left( r^{(R)} - c^{(RL)} r^{(L)} \right)}{\sigma_R \left( 1 - \left( c^{(RL)} \right)^2 \right)}, \ \ \beta_L = \frac{\sigma_v \left( r^{(L)} - c^{(RL)} r^{(R)} \right)}{\sigma_L \left( 1 - \left( c^{(RL)} \right)^2 \right)}; \tag{74}$$

as is the correlation coefficient between the true velocity and $v_e^{(\mathrm{auto})}$:

$$r^{(\mathrm{low})} = \sqrt{\frac{\left( r^{(R)} \right)^2 + \left( r^{(L)} \right)^2 - 2 c^{(RL)} r^{(R)} r^{(L)}}{1 - \left( c^{(RL)} \right)^2}}. \tag{75}$$

Across the simulated ensemble of naturalistic motions we empirically found that $r^{(R)} \approx 0.24$, $r^{(L)} \approx -0.0017$, and $c^{(RL)} \approx -0.28$. Thus, we note that $|r^{(L)}| \ll |r^{(R)}|$ and approximate the correlation coefficient as

$$\frac{r^{(\text{low})}}{r^{(R)}} \approx \sqrt{\frac{1}{1 - \left(c^{(RL)}\right)^2}}. \tag{76}$$

Thus, we expect the inclusion of the linear term $L$ to improve the accuracy of motion estimation by about 4.3% (compare to **Figure 4—figure supplement 2**). Interested readers can find a complete derivation of these equations in section V of the supplemental materials for (**Clark et al., 2014**).

With this machinery in hand, we can start to understand the utility of the linear term. First, note that this term was only weakly correlated with the velocity across the simulated ensemble of motions. Furthermore, the correlation would have been *exactly* zero if $\langle v(g*V_1) \rangle$ had been equal to $\langle v(g*V_2) \rangle$, as would have been the case for an ensemble that was perfectly translationally invariant. So the small correlation we observed between $L$ and $v$ is nothing more than residual noise resulting from a finitely sized data sample that did not explicitly enforce translation invariance. Nevertheless, it's critical to realize that **Equation 76** treated $r^{(L)}$ as if it *were* zero, yet it still managed to account for the results of **Figure 4—figure supplement 2**. Thus, this residual sampling noise has nothing to do with the improvements offered by the hybrid estimator. As intuitively expected, the linear term is completely uncorrelated with the velocity of motion.

**Equation 76** suggests that a linear term, which is itself uncorrelated with the velocity of motion, can nevertheless help velocity estimation. However, this improvement demands that it be combined with another motion estimator that: (i) is correlated with the velocity (i.e., $r^{(R)} \neq 0$); and (ii) is correlated with the linear term (i.e., $c^{(RL)} \neq 0$). Our numerical results indicate that the HRC is an example of such a motion estimator. The HRC obviously satisfies the first condition. To examine the second condition, we note that correlation between the HRC and the linear term is nonzero if and only if

$$\begin{aligned}
\langle RL \rangle &= \langle (f*V_1)(g*V_1)(g*V_2) \rangle + \langle (g*V_1)(f*V_2)(g*V_2) \rangle \\
&\quad - \langle (f*V_1)(g*V_2)^2 \rangle - \langle (g*V_1)^2(f*V_2) \rangle
\end{aligned} \tag{77}$$

is nonzero. As long as the image ensemble is light–dark asymmetric, there are no symmetry principles that force this number to vanish for a general choice of $f$ and $g$. Our numerical results show that the associated correlation coefficient is far from zero for natural inputs and our choices of filters. Fundamentally, this correlation can be nonzero because the HRC's response depends on the pattern that is moving, as does the linear response. Because image-induced variability is partially shared between the HRC and the linear term, the latter can help to eliminate image-induced noise from the HRC, thereby improving the motion estimate.

Although our results indicate that a linear term can improve local motion estimation, its benefits do not sum over space. In particular, imagine an ensemble of elementary motion detectors that combine a local HRC and a local linear estimator:

$$v_{e,i}^{(\text{low})} = \beta_R \left( (f*V_i)(g*V_{i+1}) - (g*V_i)(f*V_{i+1}) \right) + \beta_L (g*V_i - g*V_{i+1}), \tag{78}$$

where $i$ indexes the first point in space surveyed by the $i$th local estimator. A whole field motion percept could be found by averaging these local motion signals over space

$$v_e^{(\text{low})} = \frac{1}{N} \sum_{i=1}^{N} v_{e,i}^{(\text{low})}, \tag{79}$$

where $N$ denotes the total number of local motion detectors. However, the second term in the linear estimator at point $i$ cancels the first term in the linear estimator at point $i+1$. Thus, spatial averaging eliminates most of the dependence on the linear term

$$v_e^{(\text{low})} = \frac{\beta_R}{N} \sum_{i=1}^{N} ((f \star V_i)(g \star V_{i+1}) - (g \star V_i)(f \star V_{i+1})) + \frac{\beta_L}{N}(g \star V_1 - g \star V_{N+1}).$$

(80)

All that remains of the linear term is a boundary term that depends on photoreceptor activity at the edges of the visual field. Furthermore, the magnitude of this contribution decreases with $N$. Thus, linear estimators have little utility for full-field motion estimation. Nevertheless, it's conceivable that such terms could play a role in *Drosophila*'s motion estimation circuit, because the same elementary motion detector is thought to underlie a wide variety of motion-guided behaviors, and the inclusion of this locally beneficial term is not detrimental to whole field motion estimation.

Finally we note that the principles discussed in the context of linear motion estimators also apply in other counterintuitive contexts. For example, consider an autocorrelator,

$$A = (f \star V_1)(g \star V_1) - (f \star V_2)(g \star V_2),$$

(81)

which correlates visual signals from the same point in space. Like the HRC, it is mirror anti-symmetric:

$$\{V_1(t), V_2(t)\} \mapsto \{V_2(t), V_1(t)\} \Rightarrow A \mapsto -A,$$

(82)

but it is uncorrelated with the velocity. Nevertheless, the autocorrelator's correlation with the HRC is determined by

$$\langle RA \rangle = \langle (f \star V_1)^2 (g \star V_1)(g \star V_2) \rangle + \langle (g \star V_1)(f \star V_2)^2 (g \star V_2) \rangle \\ - \langle (f \star V_1)(f \star V_2)(g \star V_2)^2 \rangle - \langle (f \star V_1)(g \star V_1)^2 (f \star V_2) \rangle$$

(83)

and need not be zero. Empirically, we find the relevant correlation coefficient to be −0.40 across the ensemble of naturalistic motions, so **Equation 76** implies that this autocorrelator would enhance the HRC by 8.9%. However, such improvements do not sum over space. Thus, autocorrelators might be relevant for local motion estimates, but not for motion estimates that average over space.

## Appendix 11

# Regarding the computational problem of visual motion estimation.

Throughout this paper, we have illustrated connections between the computations performed by our models and spatiotemporal correlations. These links are important for both practical and theoretical reasons. First, the many experimental successes of the HRC already suggest that the fly's computation of motion is organized around spatiotemporal correlations in the stimulus (*Silies et al., 2014*). Thus, by relating our models to spatiotemporal correlations, we were able to discern how each model generalizes this canonical model. For example, *Figure 3—figure supplement 1B* shows that the optimal weighted 4-quadrant model supplements the standard HRC with a specific subclass of odd-ordered correlations, an observation that both reiterates the importance of the HRC and highlights the most critical signals that it lacks. Second, spatiotemporal correlations provide a fundamental connection between the motion estimation strategies used by invertebrates and vertebrates (*Adelson and Bergen, 1985*; *van Santen and Sperling, 1985*). In particular, although the HRC and motion energy models differ in their architectural details, both models are ultimately driven by 2-point correlations in the stimulus. Therefore, general arguments framed in terms of spatiotemporal correlations are easy to investigate in the specific context of either the HRC or motion energy model. Third, an understanding of the spatiotemporal correlations computed by each model facilitates the design of psychophysical experiments that test the models. For example, glider stimuli (*Hu and Victor, 2010*) provide flexible experimental tools to probe how specific correlations contribute to motion percepts. Future work will lead to a variety of more realistic models that can also be characterized by the stimulus correlations that they detect. These models can be distinguished by carefully designed glider experiments.

From a theoretical point of view, correlation functions are important because they provide a mathematical basis in which to decompose neural computations (*Poggio and Reichardt, 1973, 1980*; *Fitzgerald et al., 2011*). David Marr famously proposed that neural computation must be understood at several levels (*Marr and Poggio, 1976*). He described his second level as "that at which the algorithms that implement a computation are characterized." Our emphasis on correlation functions is directed towards unraveling motion estimation at this algorithmic level. As illustrated concretely by *Figure 3—figure supplement 1*, it's possible for an algorithm to have a simple characterization in terms of correlation functions, even when the fundamental computational units (e.g., the quadrants) do not actually compute correlations. Furthermore, correlation functions intuitively relate the visual signatures of motion to measurable features of natural visual environments (*Appendix figure 1*). Nevertheless, it's possible that correlation functions will ultimately provide an inefficient basis for representing the algorithms of visual motion estimation. For example, although the weighted 4-quadrant model is well understood in terms of the correlations that it detects, it would be nontrivial to discern its underlying simplicity based solely on its responses to glider stimuli, because the constraints relating various higher order correlators would be *a priori* unknown. Overall, we consider correlation functions to provide a useful lens for characterizing and understanding the algorithms of visual motion estimation, but research should also consider visual motion estimation in alternate bases that might reflect the brain's biological substrates more directly (*Rust et al., 2006*).

Our characterization of visual motion estimation in terms of correlation functions provides an interesting perspective on the computational problem faced by *Drosophila*'s visual motion estimator in natural environments. Natural images contain many low and high-order correlations (*Geisler, 2008*), and this implies that the fly brain could in principle use a wide array of correlations for visual motion estimation (*Appendix figure 1*). However, each correlation is only weakly associated with the velocity of motion in naturalistic settings (*Dror et al., 2001*; *Clark et al., 2014*). The reason for this is that the specific structure of the scene that is moving

acts as a nuisance parameter that hinders the unambiguous assignment of a velocity to pattern of light input. For example, it's well known that the temporal frequency of a moving sinusoidal grating shapes the HRC's output (*Egelhaaf et al., 1989*), thereby conflating the velocity with the grating's spatial frequency. More generally, the variability of a multipoint correlator across an ensemble of moving scenes is determined by higher-order statistics of the image ensemble (e.g., see Appendix 2). The fact that the same natural image drives every multipoint correlator also implies that the correlators co-vary with each other across the naturalistic motion ensemble. This shared variability can sometimes enable higher-order multipoint correlators to compensate effectively for image-induced noise that contaminates the HRC (*Clark et al., 2014*).

Questions of how brains compute behaviorally relevant stimulus features from sensory inputs are central to neuroscience, but they are extraordinarily difficult to answer, even in principle. In the context of *Drosophila*'s visual motion estimator, the ensemble of photoreceptor signals contains many nonlinear cues that are weakly correlated with the stimulus velocity and with each other under naturalistic conditions. There are many ways to pool these signals into an improved motion estimate. The space of possible stimuli is astronomically large, so it is impossible for experiments to sample it completely. Nevertheless, synthetic laboratory stimuli can be designed to rule out specific algorithms that the brain might use to estimate motion. Thus, to deconstruct a neural computation, one must find ways to dramatically restrict the space of candidate models and to identify interesting models that can be experimentally ruled out. It's important to note that we did not construct our models to reproduce the behavioral data, even though this is a straightforward exercise (*Figure 4—figure supplement 1*). Instead we aimed for a predictive framework that can relate behavioral responses to the statistics of natural sensory inputs, the statistics of natural behavior, and the constraints imposed by the neural circuits that implement the computation. Such constructions are complicated and depend on features of neural circuits that are incompletely known. Nevertheless, we hope that this added complexity will eventually pay off in computational models that have a rational structure from the viewpoint of the stimulus, the animal, and the brain.

