## [Decision Letter]

Thank you for submitting your work entitled “Naturalistic visual motion estimation by *Drosophila*” for peer review at *eLife*. Your submission has been favorably evaluated by Eve Marder (Senior Editor), Matteo Carandini (Reviewing Editor), Jonathan Victor (peer reviewer), and two other reviewers.

The reviewers have discussed the reviews with one another and the Reviewing Editor has drafted this decision to help you prepare a revised submission.

This is an important computational and mathematical study of visual motion analysis: what kinds of motion signals are present in the natural environment, how they can be extracted by biologically-plausible neural circuitry, and the features that govern their performance. The point of reference for the paper is the Hassenstein-Reichardt correlator (HRC), the canonical model for motion signal detection, and the starting point for a great deal of important work on how motion computations are reduced to circuitry. The main point of the paper is that this model fails to account for some qualitative aspects of motion processing (responses to glider stimuli), and that relatively simple extensions of the model enable it to properly predict responses to glider stimuli, and also, substantially improve its accuracy for the naturalistic motion stimuli that are the focus of the paper.

The big-picture findings (which at the moment are very difficult to grasp, see below) are important for the community at large: simple multiplication is not the best approach, and deviations from multiplication – which are being revealed by circuit-level analysis – should likely be viewed as a feature, rather than a bug. This changes the way we view the analysis of the implementation of the HRC in circuitry – rather than attempting to understand how neurons carry out a multiplication, the focus is shifted into how they make use of nonlinearities to do a computation that is, perhaps surprisingly, even better for the task.

Essential revisions:

The paper is currently written in such a way that only an exquisitely trained and alert specialist can appreciate these points. It is essential that the paper be reorganized so that there are clear questions set in context of the literature, and the paper and its logic become easier to follow for someone who is not working on motion detection in fly.

The current organization of the paper, indeed, is obscure. The paper starts by introducing 4 models that are elaborations of the Reichardt detector: (1) multiple channels that explicitly construct higher order correlation statistics, (2) A static front end nonlinearity, (3) separate correlations between lights and darks, (4) nonlinear combination through operations other than multiplication. This is a lot to take in, and only the most motivated readers will get past this point. Most readers, instead, will stop there. It would be much better to introduce the Reichardt detector (and ideally an energy model), show their failings (e.g. with glider stimuli) and then motivate ways in which they could be improved, and introduce a few of the alternative models (do we need all 4?). Otherwise, introducing the 4 models right away feels like introducing solutions in search of a problem.

A key problem is that all the models considered seem to work better than the Reichardt model, with no compelling argument for one vs. the other. Possibly this should be regarded as generating a set of alternate hypotheses for physiologists and anatomists to investigate, but it would be useful to understand better what separates these models.

Because of this, and because the style of writing is opaque, it is currently difficult to discern the “bottom line” in this paper. It reads as a survey of many possible models and some of their virtues and discontents.

It is essential for the authors to decide what exactly they want the paper to convey, and set out a clear set of questions in Introduction, answer them in Results, and return to them in Discussion. The current paper, by contrast, seems to change its mind along the way as to what those questions are, and the reader is left grasping for a specific set of questions that need addressing. For instance, the Discussion (second paragraph) emphasizes processing in segregated ON and OFF channels. Is this the main take-home message? If so, it should be clearly set out as a question in the Introduction.

More generally, it seems essential that the authors introduce the problem in a broader context. As written, the paper uses computational methods to specifically examine some possible mechanisms of motion estimation in flies in view of the inability of the traditional Reichardt model to explain how animals use higher order motion cues. These are important issues, but to engage that broader interest it would be good to relate more substantially with the literature on higher order statistics in natural scenes and adaptation of circuit structure and perceptual phenomena to these statistics. Some suggestions appear in Reviewer 3's specific comments, appended below.

The language needs to become clearer. For instance, at the end of the subsection “Model responses to glider stimuli”, you state, “utility of higher-order correlations for naturalistic motion processing is not restricted to a specific neural circuit implementation.” What does this mean?

Also, while it is understandable that the authors may want to protect the reader from too much detail, at the moment the paper seems to put too much away into the appendices for the reader to be able to follow exactly what is going on. In turn, the material in the appendices is written in an extremely mathematical style, too much so for the readers of this journal – for example, “simply” is overused.

In addition to these key issues of organization and style, there are also some conceptual aspects that need work:

1) The paper seems to concern only rigid, constant, fronto-parallel motion. While this limitation needs to be explicitly stated, it also increases the strength of the findings. That is, there are many ways in which real motion estimation differ from this simplified scenario: for example, there are objects that may be moving independently of the visual flow, and objects may move in depth. Getting rid of these confounds will further complicate algorithms for extraction of visual flow, and influence what is “optimal”. So it is really quite remarkable that even with these real-world complications neglected, the authors still find that extraction of motion is benefited by mechanisms that make use of high-order nonlinearities.

2) The authors' choice of a way to evaluate performance, i.e. correlation of the inferred motion signal with the veridical motion, is problematic for two related reasons. Firstly, performance can be increased merely by attenuating large-magnitude outliers in the estimate (even simply by passing the output through a sigmoidal nonlinearity, independent of the stimulus). Secondly, the mathematical analysis is complicated because of the need to compute the denominator. There is no single best measure of performance, but for the above reasons, correlation is suboptimal. Two possible alternatives are: (a) to use the covariance (or, equivalently, mean-squared error) more extensively – this gets rid of the denominator problem, and may also simplify the understanding of the role of the autocorrelators; (b) to use mutual information between true and estimated velocity. The advantage of this is that it will not depend on any output transformation. There's a good argument that this is biologically appropriate, since after all, the fly does not need to know the velocity, it just needs to control its behavior – and there may well be nonlinear transformations that intervene between the motion signal output, and the motor command. In sum, it would be useful to know whether maximizing covariance, or maximizing information about velocity, would lead to the same conclusions as maximizing correlation. Note that the authors don't necessarily need to refit the models to maximize mutual information. Rather, all that is needed is to take the existing fitted models and calculate the mutual information from the scatter plot of true velocity vs. calculated velocity – a one dimensional calculation that can be done with binning in, say, 16 velocity bins of approximately equal occupancy.

3) The benefits of the autocorrelator are not clear, despite the paper's lengthy material on this point. To get off the ground: is it the case that the stimulus set is symmetric with respect to velocity – i.e., that each spatial profile is presented as moving both with positive and negative velocity? If this is not the case, then it would seem important to explain why this is justified, and of course, autocorrelators may help. And unless this asymmetry is present in natural stimuli, it would seem to be a flaw in the analysis that needs to be fixed.

Indeed, perhaps the stimulus set is actually not symmetric with respect to velocity. This is suggested by the sentence following [Disp-formula equ7] of Appendix 4: “The performance of the combined motion estimator is superior to the original direction selective estimator whenever the latter [i.e., the direction selective estimator] has a nonzero correlation with the autocorrelator.”

But if the stimulus set is symmetric in this regard, the result is quite puzzling. Let's say a particular estimator A that was optimal had the following behavior. For a particular stimulus moving with a positive velocity (say, S^+^, with signed velocity v(S^+^)) gave a result E^+^, and for the same stimulus moving with a negative velocity (say, S^-^, with signed velocity v(S^-^)=-v(S^+^)) gave a result E^-^. We could also construct an estimator A*, which first inverted the stimulus, and then applied A, and then inverted the result – so that A* would give the result of-(E-) for S^+^, and -(E^+^) for S^-^. If A is optimal, then (by symmetry) A* would also have to be optimal. But then it would seem that a new estimator, B=(A+A*)/2, would be better than both, unless A and A* are identical. The reason for this is that B removes any bias (towards either positive or negative velocities) that A or A* might have, and one can always decompose the error (across the entire ensemble) into the error for the summed velocity (v(S^+^)+v(S^-^)) and for the difference velocity (v(S^+^)-v(S^-^)), for each pair of stimuli. B reduces the error for the summed velocity to zero, and does not influence the error for the difference velocity. Is there a problem with this argument?

Assuming that the stimulus set is symmetric w.r.t. velocity, and, the above argument is correct, then it is hard to understand how linear addition of a signal Z that is generated in a spatially-symmetric fashion, including an autocorrelator, can improve the performance of an estimator if measured by covariance. But perhaps it might help by improving the correlation (see item 2 above) – and this should be clarified. On the other hand, using a symmetric signal Z in a nonlinear way (e.g., dividing by (1+Z^2^)) could improve the performance as measured by covariance or correlation, by reducing outliers. However, it is not clear that this is what the authors are doing.

In sum – the paper should clarify whether the stimulus set is symmetric with respect to velocity sign. If it is asymmetric, it should justify this asymmetry, and, be clear whether it is responsible for the utility of autocorrelators. If it is symmetric, then further explanation is needed as to why autocorrelators are useful.

4) Regarding the Discussion section concerning motion energy computations in vertebrates: In considering how the conclusions might apply to the motion energy model, it looks like the paper is suggesting that vertebrates might use different kinds of deviations from strict multiplication, because of the different linear structure of the motion energy model. In this regard, it would be useful to point out that cellular-level analysis in the macaque (Nitzany et al., Evolutionary convergence in computation of local motion signals in monkey and dragonfly. CoSyNe , 2014) shows that the consequences of these deviations is very similar in terms of the motion signal that is extracted – at least in terms of detection of three-point correlations, which are crucial to this paper.

5) Nesting of models. A diagram that shows the “nesting” relationships of the models – which ones are special cases of others, and that the HRC is a special case of all – would be helpful.

6) Static nonlinearities at the front end: the paper considers binarization and histogram-equalization. Might it be useful to add a front-end nonlinearity that converts the intensity distribution to a Gaussian? Kurtosis would be higher than either, but this is the distribution that maximizes information (entropy) for a given variance – so it would be interesting to see how it does.

7) The authors must make their program code for the results available either on the journal web site or on a publicly accessible data base. Please add details to the Method on this last point.

Reviewer #3:

1) At the end of the Introduction, the authors say that the perceptual measurements are consistent with only a subset of their models. It would be useful to be clear up front here about which models worked and which ones didn't and in what ways.

2) A comment about writing style. The paper frequently has expressions like “we hypothesize that biology tunes its motion estimators[…]” (subsection “Strategies for visual motion estimation”, first paragraph). Personally, I find it a bit grating to read this sort of broad (over-)generalization about “biology”. Are there motion detectors of some kind in plants, cyanobacteria and tube-worms at deep-sea vents? Maybe, but the authors are not saying anything that would convince me about how they work. There are many such occurrences of the “biology does X or Y” phrasing, which the authors would be well-advised to remove. (Another example is in the third paragraph, but there are many others.) In any case, the hypothesis of tuning of circuits to natural scenes statistics arising from ordinary behavior is hardly new, so a nod to the venerable history of this idea would be a good thing here.

3) In reading the third paragraph of the subsecton “Strategies for visual motion estimation” and Figure 1, I could not understand how the static front-end nonlinearity allowed the circuit compute higher order spatial correlations more easily. Or maybe that is not being implied, but it seemed to be. Later on it becomes more clear that the nonlinearity is supposed to remove kurtosis, but I was confused at this stage in the paper. More generally, it would be helpful to get some conceptual sense of why these particular models are considered – otherwise one gets a sense of a bit of a grab-bag, especially since many of these models could be combined with each other, no doubt giving improvements in each case.

4) In the subsection entitled “Each mechanisms outperforms the HRC[…]”, the authors show that their various models work better than the HRC. They then say that they can gain insight because their models are theoretical tractable. As an example, they say that their front-end nonlinearity does contrast equalization. But this is the first time they say anything about the nature of the nonlinearity, and they don't say why this kind of normalization helps. So I am afraid that the insight does not come through. Continuing in this vein, they say that “a large fraction of […] performance was afforded by a small number of correlation types”. But there has been no discussion in the text at this point of the possible correlation types and which ones are being used, so the comment remains opaque. Then they write “binarizing nonlinearities also offered certain […] advantages”. Again, this is the first time binarizing nonlinearities are mentioned and it is not clear what the advantages are. It seems clear from the opacity of the text that more details of the model variations should be described earlier, in order to make the paper easier to read. Looking at Figure 2 it is clear that everything outperforms the HRC. But the HRC really does poorly – a correlation coefficient of 0.25 or so. The best mechanism has a correlation coefficient of 0.5. This leads me to worry that none of these proposed mechanisms actually works that well. Also, a main message of this paper is that multi-point correlations are important to motion estimation. But it seems in this Figure that the front-end nonlinearity has the biggest effect. Why didn't the authors combine this nonlinearity with the other mechanisms they consider?

5) The authors cite their own work for the glider stimuli. Don't these stimuli and associated analysis come from the decades of work by J. Victor and collaborators? I might be mistaken about this – I am most familiar with the spatial stimuli created with gliders that I learned about from those papers. I understand how these stimuli are constructed from those works, but I suspect that the general reader will need a brief introduction at this stage in the paper, even though this is covered in previous works.

6) I appreciated the clear statement in the first paragraph of subsection “Model responses to glider stimuli” about what the HRC model fails to predict. In the next paragraph (and Figure 3) the authors discuss how the front-end nonlinearity increases accuracy compared to HRC broadly speaking, but fail to match the responses to negative 2-point correlations and some 3-point gliders. Then they show that models that explicitly compute higher-order correlations correctly predicted the sign of all glider responses, but did not predict the detailed response amplitudes. They try different architectures and find that several architectures show similar performance overall. Of course the output of any of these circuits need not be directly equal to the turning rate of the animal. Am I correct in understanding that the model is that a single gain parameter should relate all of the turning rates under different conditions to the output of the circuit, and that this single gain is fixed by normalizing the positive two-point glider? Also, in Figure 3, is the “equalized” model the same as the front-end nonlinearity model in Figure 1? In what sense is it “equalized”?

7) In the subsection “Improving motion estimation by accounting for natural light-dark asymmetries”, the authors describe results that show that accounting for bright-dark asymmetry improved the results of all of the mechanisms that got the sign of glider responses right. While the text has a technical discussion how various combinations of signals work, I did not understand from the text the conceptual reason why the bright-dark separation helps. At some broad level it is not surprising that adapting to the natural statistics helps with the detection of signals, and indeed it seems here that all the mechanisms are helped by incorporating bright-dark asymmetries. Is there a deeper insight here? Or maybe the point is that the authors are simply making a prediction that the higher order motion detection circuits in flies will be discovered to segregate ON and OFF pathways and then recombine them, independently of which detailed mechanism and nonlinearities are being used? Now in Figure 4, most of the quadrant models seem to do worse than HRC, and the (— —) quadrant seems not to be significantly better. And all of these models have a very low correlation with velocity. In view of this, could the authors please clarify why they are saying that accounting for bright-dark asymmetry in their model improves things?

8) In the subsection “Improving motion estimation by reducing kurtosis” the authors explain that their front end nonlinearities improved motion estimation by reducing the kurtosis in the inputs, even though they did not help with predicting glider responses. Would it help to have these nonlinearities along with the correlation detectors and the ON-OFF segregation discussed in previous sections?

9) As written, the paper uses computational methods to specifically examine some possible mechanisms of motion estimation in flies in view of the inability of the traditional Reichardt model to explain how animals use higher order motion cues. I agree with the authors that the work has potentially broader significance beyond the specific example discussed here. But in order to engage that broader interest it would be really good if the authors would engage more substantially with the literature on higher order statistics in natural scenes and adaptation of circuit structure and perceptual phenomena to these statistics. At the moment the engagement is cursory. Some suggestions appear below. None of the papers mentioned below is specifically about motion estimation and the suggestion to engage a bit more with all this is not intended to detract from the novelty of the present work; rather it will likely help readers to find the work situated better against this well-known context.

A) Light-dark asymmetry of natural stimuli and consequences for visual cicrcuits:

(i)Ratliff, Charles P., et al. “Retina is structured to process an excess of darkness in natural scenes.” Proceedings of the National Academy of Sciences 107.40 (2010): 17368-17373, and references therein on light-dark statistics;

(ii) Komban, Stanley Jose, Jose-Manuel Alonso, and Qasim Zaidi. “Darks are processed faster than lights.” The Journal of Neuroscience 31.23 (2011): 8654-865, and references therein.

B) Kurtosis and higher moments of the distribution of light:

(i) Bonin, Vincent, Valerio Mante, and Matteo Carandini. “The statistical computation underlying contrast gain control.” The Journal of neuroscience 26.23 (2006): 6346-6353, and references therein.

C) Higher order spatial correlations and perception in the “glider” formulation used here:

(i) A long history of studies by Victor, and specifically the recent work: Tkačik, Gašper, et al. “Local statistics in natural scenes predict the saliency of synthetic textures.” Proceedings of the National Academy of Sciences 107.42 (2010): 18149-18154; Hermundstad, Ann M., et al. “Variance predicts salience in central sensory processing.” *eLife* 3 (2014): e03722, and references therein;

(ii) Many papers by the Simoncelli group about visual textures.

---

## [Author Response]

*Essential revisions*:

*The paper is currently written in such a way that only an exquisitely trained and alert specialist can appreciate these points. It is essential that the paper be reorganized so that there are clear questions set in context of the literature, and the paper and its logic become easier to follow for someone who is not working on motion detection in fly*.

*The current organization of the paper, indeed, is obscure. The paper starts by introducing 4 models that are elaborations of the Reichardt detector: (1) multiple channels that explicitly construct higher order correlation statistics, (2) A static front end nonlinearity, (3) separate correlations between lights and darks, (4) nonlinear combination through operations other than multiplication. This is a lot to take in, and only the most motivated readers will get past this point. Most readers, instead, will stop there. It would be much better to introduce the Reichardt detector (and ideally an energy model), show their failings (e.g. with glider stimuli) and then motivate ways in which they could be improved, and introduce a few of the alternative models (do we need all 4?). Otherwise, introducing the 4 models right away feels like introducing solutions in search of a problem*.

In response to this suggestion and others below, we have fully restructured our paper. Instead of introducing all models simultaneously at the beginning of the paper, we now first discuss experimental and theoretical reasons that the pure HRC might be disfavored. This includes introducing the puzzling glider psychophysical results. We then show that front-end nonlinearities, while able to improve motion signals with natural inputs, are unable to recapitulate the glider response patterns. We next explore the ON/OFF model, which is probably most proximal to the field’s current conception of the circuit, and show that it improves natural motion estimation and matches experimental glider responses. However, it is important not to get stuck on the first model that fits the glider data reasonably well, because other aspects of *Drosophila*’s circuitry still conflict with the HRC’s hypotheses. We thus continue to explore the space of models with other plausible nonlinear modifications to the HRC: the non-multiplicative nonlinearity model, an unrestricted nonlinearity model, which is a circuit-motivated renaming of the former “explicit multipoint correlator model,” and an extra-input nonlinearity model. This last model is new to this revision, but we believe it is informative because it moves us further beyond the HRC and is motivated by EM reconstructions that show wide-field inputs to the fly’s motion detectors.

As hinted above, we now focus much more on explaining the biological motivation for each model. This is aided by a new table in the manuscript, which gives experimental justifications for each generalization. One of our goals in this paper is to move away from orthodox models by emphasizing the potential of more complex circuit computations. However, the models we consider are roughly hierarchical, moving to ever more general conceptualizations. This enables us to conclude the paper with our bottom line – the final model incorporates the variety of conceptual advances that were initially illustrated with specific models. This includes, but goes beyond, our point that many of our models improve natural motion estimation through asymmetric treatment of light and dark by the motion estimator. We believe this conceptual hierarchy of models clarifies why it is helpful to consider multiple models, rather than just one: the computational structure of the extra input nonlinearity model is too complicated to take in at once, and the other models in the hierarchy help to dissect it.

*A key problem is that all the models considered seem to work better than the Reichardt model, with no compelling argument for one vs. the other. Possibly this should be regarded as generating a set of alternate hypotheses for physiologists and anatomists to investigate, but it would be useful to understand better what separates these models*.

Our new structure, described above, provides a stronger rationale for each of the models, introducing them one by one and in a logical progression, from the most conceptually-proximal to the most generalized. Table 1 now gives experimental rationales for examining each model type. We do not intend to present our models as competitors. Rather, each illustrates something new about naturalistic motion estimation. As the reviewers say, each generates a set of hypotheses for experiments. Our new hierarchical organization should make this clearer.

*Because of this, and because the style of writing is opaque, it is currently difficult to discern the “bottom line” in this paper. It reads as a survey of many possible models and some of their virtues and discontents*.

*It is essential for the authors to decide what exactly they want the paper to convey, and set out a clear set of questions in Introduction, answer them in Results, and return to them in Discussion. The current paper, by contrast, seems to change its mind along the way as to what those questions are, and the reader is left grasping for a specific set of questions that need addressing. For instance, the Discussion (second paragraph) emphasizes processing in segregated ON and OFF channels. Is this the main take-home message? If so, it should be clearly set out as a question in the Introduction*.

As described above, we have restructured the paper to better tease out the bottom line of our paper. More generally, we have highlighted several points that are most critical to the paper: (a) In the motion estimation circuit, nonlinearities are a feature, not a bug, and can be tuned to improve motion estimation in natural scenes. Such tuning also accounts for puzzling psychophysical results. (b) We must explore the space of models to discern critical details from non-critical ones, and make sure our results aren’t the product of idiosyncracies of one particular model. (c) The critical detail of the models that reproduce the behavioral data is that they treat light and dark asymmetrically. (d) The models are conceptually hierarchical, and the most general model recapitulates the insights provided by its predecessors.

*More generally, it seems essential that the authors introduce the problem in a broader context. As written, the paper uses computational methods to specifically examine some possible mechanisms of motion estimation in flies in view of the inability of the traditional Reichardt model to explain how animals use higher order motion cues. These are important issues, but to engage that broader interest it would be good to relate more substantially with the literature on higher order statistics in natural scenes and adaptation of circuit structure and perceptual phenomena to these statistics. Some suggestions appear in Reviewer 3's specific comments, appended below*.

Thank you for this comment. Our revised manuscript rewrites the Introduction and parts of the Discussion to place our work in the broader contexts suggested by the reviewers.

*The language needs to become clearer. For instance, at the end of the subsection “Model responses to glider stimuli”, you state, “utility of higher-order correlations for naturalistic motion processing is not restricted to a specific neural circuit implementation.” What does this mean*?

In response to this comment we have tried to clarify and simplify our writing. (In this particular case, we meant that many models were sufficient to make use of HOCs to improve motion estimates. This is now clarified in the text.)

*Also, while it is understandable that the authors may want to protect the reader from too much detail, at the moment the paper seems to put too much away into the appendices for the reader to be able to follow exactly what is going on. In turn, the material in the appendices is written in an extremely mathematical style, too much so for the readers of this journal – for example, “simply” is overused*.

We have moved some of the material from the appendices to the main text to make the paper easier to follow. All mathematics remains in the appendices. We have gone through all of the appendices to clarify our arguments and simplify our prose. This word “simply” no longer appears. We have added explanatory prose to help ease the reader through the math. More importantly, we think it’s now possible for the reader to understand the paper without ever looking at the math, which is an important improvement.

*In addition to these key issues of organization and style, there are also some conceptual aspects that need work*:

*1) The paper seems to concern only rigid, constant, fronto-parallel motion. While this limitation needs to be explicitly stated, it also increases the strength of the findings. That is, there are many ways in which real motion estimation differ from this simplified scenario: for example, there are objects that may be moving independently of the visual flow, and objects may move in depth. Getting rid of these confounds will further complicate algorithms for extraction of visual flow, and influence what is “optimal”. So it is really quite remarkable that even with these real-world complications neglected, the authors still find that extraction of motion is benefited by mechanisms that make use of high-order nonlinearities*.

We agree that this is remarkable and interesting, and we have added a Discussion paragraph about this point.

*2) The authors' choice of a way to evaluate performance, i.e. correlation of the inferred motion signal with the veridical motion, is problematic for two related reasons. Firstly, performance can be increased merely by attenuating large-magnitude outliers in the estimate (even simply by passing the output through a sigmoidal nonlinearity, independent of the stimulus). Secondly, the mathematical analysis is complicated because of the need to compute the denominator. There is no single best measure of performance, but for the above reasons, correlation is suboptimal. Two possible alternatives are: (a) to use the covariance (or, equivalently, mean-squared error) more extensively – this gets rid of the denominator problem, and may also simplify the understanding of the role of the autocorrelators; (b) to use mutual information between true and estimated velocity. The advantage of this is that it will not depend on any output transformation. There's a good argument that this is biologically appropriate, since after all, the fly does not need to know the velocity, it just needs to control its behavior – and there may well be nonlinear transformations that intervene between the motion signal output, and the motor command. In sum, it would be useful to know whether maximizing covariance, or maximizing information about velocity, would lead to the same conclusions as maximizing correlation. Note that the authors don't necessarily need to refit the models to maximize mutual information. Rather, all that is needed is to take the existing fitted models and calculate the mutual information from the scatter plot of true velocity vs. calculated velocity – a one dimensional calculation that can be done with binning in, say, 16 velocity bins of approximately equal occupancy*.

We thank the reviewers for this comment. We emphasized the correlation coefficient because it is intuitive and has a general relationship with the mean- squared error. In particular, we’ve always been choosing parameters that minimize the mean-squared error, and we’re only using the correlation coefficient as a more intuitive metric to present those results. This was not clear in the previous draft. To make it clear in the new draft, we now introduce our optimization procedures in terms of the mean squared error, and we explicitly describe the relationship between the correlation coefficient and the mean squared error in the Methods and in Appendix 2. We also use this relationship to explicitly motivate our continued graphical reliance on the correlation coefficient. We hope that the situation is clear now. Simply put, everything in this paper is in terms of the mean squared error, and the correlation coefficient is merely an intuitive way to signify the mean squared error of correctly scaled models.

We have refrained from using mutual information to assess our motion estimators for both practical and conceptual reasons. Practically speaking, it is more difficult to tune high-dimensional models for mutual information than for the mean-squared error. The referees already allude to this practical problem by not asking us to refit the models. More importantly, we fear that the mutual information could be conceptually misleading in the present context. For example, the referees’ comment about autocorrelators clearly indicates that they would not consider the speed of motion to be a good estimator of its velocity. We agree. Nevertheless, the mutual information between the velocity of motion and its speed is the entropy of the velocity distribution minus one bit, where the “minus one” corresponds to the missing information about the motion’s direction. Thus, the mutual information between the speed and velocity can be huge for broad velocity distributions. For example, it can certainly be greater than 1 bit, in which case the mutual information metric would claim that the speed of motion is a better estimator of the velocity than the direction of motion. We think this is the wrong conclusion. As the referees say, what ultimately matters is how well the animal can control its behavior, and the direction of motion is clearly more useful for orienting behaviors than the speed of motion, independently of how broad the velocity distribution is. Furthermore, we would not consider the ensemble of raw photoreceptor signals to be a good estimator of the velocity of motion, but these signals have at least as much mutual information with the velocity as any motion estimator we can hope to dream up. Overall, we believe that both the mutual information and the mean squared error have their own domains of superiority, and we think that the mean squared error is a more appropriate fit for this paper. Both the mean squared error and the mutual information are standard metrics, and we hope the referees will permit us to emphasize the metric that we strongly prefer.

*3) The benefits of the autocorrelator are not clear, despite the paper's lengthy material on this point*.

Thank you for pointing out that this section was unclear. Your comments lead us to carefully revisit the utility of the autocorrelator, and these efforts have led us to clearer understanding. As described below, we still argue that autocorrelators have utility for motion estimation. We have fully rewritten the associated Appendix (now Appendix 10) to enhance its clarity in light of the referees’ specific concerns. Note that the new appendix discusses the mechanisms underlying the autocorrelator’s utility in a more general setting. The new paper organization leads us to emphasize a different computational example in Appendix 10, but we still comment on autocorrelators at the end. Below we will discuss Appendix 10 as if it had emphasized an autocorrelator example, because we want our response to most directly connect to the referees’ specific comments. We hope the combined concreteness and generality of the Appendix will help to clarify the situation for the referees and future readers.

*To get off the ground: is it the case that the stimulus set is symmetric with respect to velocity – i.e., that each spatial profile is presented as moving both with positive and negative velocity? If this is not the case, then it would seem important to explain why this is justified, and of course, autocorrelators may help. And unless this asymmetry is present in natural stimuli, it would seem to be a flaw in the analysis that needs to be fixed*.

The previous ensemble was statistically symmetric, in the sense that each spatial profile was equally *likely* to be presented with velocity *v* and –*v*. However, it is conceivable that our finite ensemble of simulated motions was too small to have achieved the desired level of left-right symmetry. To eliminate this possibility, we generated a new ensemble that explicitly enforced left-right symmetry: whenever a pattern with velocity *v* was randomly sampled, we reflected the pattern and presented it with velocity –v. We numerically confirmed that mirror-paired photoreceptor signals were exactly equal between the two simulations. This change didn’t affect the results, which suggests that our original ensemble was large enough to safely consider as left-right symmetric. Nevertheless, we now use the new ensemble with enforced left-right symmetry throughout the paper.

*Indeed, perhaps the stimulus set is actually not symmetric with respect to velocity. This is suggested by the sentence following*
[Disp-formula equ7]
*of*
Appendix 4*: “The performance of the combined motion estimator is superior to the original direction selective estimator whenever the latter [i.e., the direction selective estimator] has a nonzero correlation with the autocorrelator.”*

It is mathematically possible for the autocorrelator to be correlated with a direction- selective motion estimator (e.g. the HRC) without being correlated with velocity of motion. Indeed, this is what happens. The Appendix now discusses this initially counter-intuitive point at greater length. Basically, this result occurs because the HRC’s response depends on the pattern that is moving, as does the autocorrelator’s response. Because image-induced variability is partially shared between the HRC and the autocorrelator, the autocorrelator can help to eliminate image-induced noise from the HRC, thereby improving the motion estimate.

*Let's say a particular estimator A that was optimal had the following behavior. For a particular stimulus moving with a positive velocity (say, S*^*+*^*, with signed velocity v(S*^*+*^*)) gave a result E*^*+*^*, and for the same stimulus moving with a negative velocity (say, S*^*-*^*, with signed velocity v(S*^*-*^*)=-v(S*^*+*^*)) gave a result E*^*-*^*. We could also construct an estimator A*, which first inverted the stimulus, and then applied A, and then inverted the result – so that A* would give the result of-(E-) for S*^*+*^*, and -(E*^*+*^*) for S*^*-*^*. If A is optimal, then (by symmetry) A* would also have to be optimal. But then it would seem that a new estimator, B=(A+A*)/2, would be better than both, unless A and A* are identical. The reason for this is that B removes any bias (towards either positive or negative velocities) that A or A* might have, and one can always decompose the error (across the entire ensemble) into the error for the summed velocity (v(S*^*+*^*)+v(S*^*-*^*)) and for the difference velocity (v(S*^*+*^*)-v(S*^*-*^*)), for each pair of stimuli. B reduces the error for the summed velocity to zero, and does not influence the error for the difference velocity. Is there a problem with this argument?*

We think this argument is elegant and 100% correct.

*Assuming that the stimulus set is symmetric w.r.t. velocity, and, the above argument is correct, then it is hard to understand how linear addition of a signal Z that is generated in a spatially-symmetric fashion, including an autocorrelator, can improve the performance of an estimator if measured by covariance*.

Your argument helped us to realize that autocorrelators are only useful when they are combined in a mirror anti-symmetric way across space, but this is what they had been doing all along. For example, in Appendix 10 we discuss the usefulness of the autocorrelator: (f*V_1)(g*V_1) – (f*V_2)(g*V_2). When the referees consider this specific form of autocorrelator, we think they will understand why its utility doesn’t contradict their argument. Nevertheless, its utility might still be opaque, and we hope the material Appendix 10 will make its utility easier to understand.

*But perhaps it might help by improving the correlation (see item 2 above) – and this should be clarified*.

As described above, we do not think that our choice of error function is a cause for concern.

*On the other hand, using a symmetric signal Z in a nonlinear way (e.g., dividing by (1+Z*^*2*^*)) could improve the performance as measured by covariance or correlation, by reducing outliers. However, it is not clear that this is what the authors are doing.*

Part of the confusion regarding “what the authors are doing” probably relates to the fact that the old version of the Appendix had some discussion of autocorrelators that went beyond trying to explain the specific results that we observed. For example, we described some models that utilized autocorrelation functions in a nonlinear way. We have now deleted that material from the Appendix entirely.

*In sum – the paper should clarify whether the stimulus set is symmetric with respect to velocity sign. If it is asymmetric, it should justify this asymmetry, and, be clear whether it is responsible for the utility of autocorrelators. If it is symmetric, then further explanation is needed as to why autocorrelators are useful*.

We hope Appendix 10 has succeeded in this regard.

*4) Regarding the Discussion section concerning motion energy computations in vertebrates: In considering how the conclusions might apply to the motion energy model, it looks like the paper is suggesting that vertebrates might use different kinds of deviations from strict multiplication, because of the different linear structure of the motion energy model. In this regard, it would be useful to point out that cellular-level analysis in the macaque (Nitzany et al., Evolutionary convergence in computation of local motion signals in monkey and dragonfly. CoSyNe , 2014) shows that the consequences of these deviations is very similar in terms of the motion signal that is extracted – at least in terms of detection of three-point correlations, which are crucial to this paper*.

We thank the reviewers for this citation, which is now referenced appropriately in that Discussion paragraph.

*5) Nesting of models. A diagram that shows the “nesting” relationships of the models – which ones are special cases of others, and that the HRC is a special case of all – would be helpful*.

This was a fun diagram to think about, and we agree it is helpful. We have included it in Figure 5, where we discuss the most general model, and its relationship to the others. In fact, this conception of the hierarchy of models was key to reorganizing the paper.

*6) Static nonlinearities at the front end: the paper considers binarization and histogram-equalization. Might it be useful to add a front-end nonlinearity that converts the intensity distribution to a Gaussian? Kurtosis would be higher than either, but this is the distribution that maximizes information (entropy) for a given variance – so it would be interesting to see how it does*.

We simulated this Gaussianizing nonlinearity and found that it performed less well in natural scenes than the binarizing or equalizing nonlinearities. Like the other front-end nonlinearities, it didn’t generate glider responses that matched the data. This information is fully included in the results section about front-end nonlinearities and in Figure 2.

*7) The authors must make their program code for the results available either on the journal web site or on a publicly accessible data base. Please add details to the methods on this last point*.

We agree to provide code that will be posted on the journal website. However, we want to first ensure that the referees and editors are satisfied by our new paper organization. The process of assembling the code is time-consuming, so we want to do it only once in a way that reflects the final published paper’s organization.

*Reviewer #3*:

We thank Reviewer 3 for these specific comments. We have addressed them all in our revision. In particular, we have included an Introduction paragraph that puts our work in the broader context of natural scene statistics and visual processing.

*1) At the end of the Introduction, the authors say that the perceptual measurements are consistent with only a subset of their models. It would be useful to be clear up front here about which models worked and which ones didn't and in what ways*.

We think this should be clear in the revised manuscript, because we now go through the models more sequentially.

*2) A comment about writing style. The paper frequently has expressions like “we hypothesize that biology tunes its motion estimators[…]” (subsection “Strategies for visual motion estimation”, first paragraph). Personally, I find it a bit grating to read this sort of broad (over-)generalization about “biology”. Are there motion detectors of some kind in plants, cyanobacteria and tube-worms at deep-sea vents? Maybe, but the authors are not saying anything that would convince me about how they work. There are many such occurrences of the “biology does X or Y” phrasing, which the authors would be well-advised to remove. (Another example is in the third paragraph, but there are many others.) In any case, the hypothesis of tuning of circuits to natural scenes statistics arising from ordinary behavior is hardly new, so a nod to the venerable history of this idea would be a good thing here*.

We agree with this comment and have removed references to what biology does. In a few places, we have emphasized that evolution might tune specific parameters, and we have added appropriate citations for this idea.

*3) In reading the third paragraph of the subsecton “Strategies for visual motion estimation” and*
Figure 1*, I could not understand how the static front-end nonlinearity allowed the circuit compute higher order spatial correlations more easily. Or maybe that is not being implied, but it seemed to be. Later on it becomes more clear that the nonlinearity is supposed to remove kurtosis, but I was confused at this stage in the paper. More generally, it would be helpful to get some conceptual sense of why these particular models are considered – otherwise one gets a sense of a bit of a grab-bag, especially since many of these models could be combined with each other, no doubt giving improvements in each case.*

In the new structure of the paper, we believe it is clearer why we chose these particular models. We have added Table 1, which gives clear experimental rationales (and citations) for each of the models and generalized models we consider. We have also changed our Introduction to give more background for why these models are chosen.

*4) In the subsection entitled “Each mechanisms outperforms the HRC[…]”, the authors show that their various models work better than the HRC. They then say that they can gain insight because their models are theoretical tractable. As an example, they say that their front-end nonlinearity does contrast equalization. But this is the first time they say anything about the nature of the nonlinearity, and they don't say why this kind of normalization helps. So I am afraid that the insight does not come through. Continuing in this vein, they say that “a large fraction of […] performance was afforded by a small number of correlation types”. But there has been no discussion in the text at this point of the possible correlation types and which ones are being used, so the comment remains opaque. Then they write “binarizing nonlinearities also offered certain […] advantages”. Again, this is the first time binarizing nonlinearities are mentioned and it is not clear what the advantages are. It seems clear from the opacity of the text that more details of the model variations should be described earlier, in order to make the paper easier to read. Looking at*
Figure 2
*it is clear that everything outperforms the HRC. But the HRC really does poorly – a correlation coefficient of 0.25 or so. The best mechanism has a correlation coefficient of 0.5. This leads me to worry that none of these proposed mechanisms actually works that well. Also, a main message of this paper is that multi-point correlations are important to motion estimation. But it seems in this Figure that the front-end nonlinearity has the biggest effect. Why didn't the authors combine this nonlinearity with the other mechanisms they consider?*

To address this comment, we have first clarified our prose surrounding the introduction of the front-end nonlinearities, and we describe more details earlier in the new structure of the manuscript.

The second point about the low correlation coefficient of the HRC is worth exploring. Our models perform no averaging in time or space, so their performance is worse than one could do if one pooled signals over space. The HRC performance we measure is reminiscent of the noisy performance reported in [15]. The fact that the HRC doesn’t work very well for natural inputs is part of the reason that we’re looking for alternatives.

It is true that the front-end nonlinearity had the largest effect of any nonlinearity in improving natural motion estimates. Yet it poorly predicted the psychophysics. We agree that stringing models together may improve motion estimates, but our goal here was to first understand how the different mechanisms work in isolation. We believe that combining these nonlinearities is fodder for future work, but it is not appropriate in this paper, in part because of the combinatorial explosion expected when we start combining models, and in part because combining models makes it far more difficult to understand how they function to improve estimates. That said, we do now emphasize the hierarchical nature of our models, and the final figure of the paper should help future reader think about how these models might be combined.

It’s also interesting to note that the front-end nonlinearities considered here eliminate the light-dark asymmetry from their natural inputs. Thus, such non- linearly preprocessed signals cannot combine effectively with some of the other mechanisms that we describe, such as ON/OFF processing. We now point this out in the text. Our overall view is that tuning frontend nonlinearities for motion estimation is a good way to improve estimation performance, but it doesn’t provide a good way of understanding *Drosophila*’s computation and circuitry.

*5) The authors cite their own work for the glider stimuli. Don't these stimuli and associated analysis come from the decades of work by J. Victor and collaborators? I might be mistaken about this – I am most familiar with the spatial stimuli created with gliders that I learned about from those papers. I understand how these stimuli are constructed from those works, but I suspect that the general reader will need a brief introduction at this stage in the paper, even though this is covered in previous works*.

We have introduced the gliders more prominently in the new structure of the manuscript (Figure 1). We continue to cite Hu and Victor prominently as the originator of these spatiotemporal stimuli. We also now cite some of Victor’s references containing spatial gliders in the Introduction.

*6) I appreciated the clear statement in the first paragraph of subsection “Model responses to glider stimuli” about what the HRC model fails to predict. In the next paragraph (and*
Figure 3*) the authors discuss how the front-end nonlinearity increases accuracy compared to HRC broadly speaking, but fail to match the responses to negative 2-point correlations and some 3-point gliders. Then they show that models that explicitly compute higher-order correlations correctly predicted the sign of all glider responses, but did not predict the detailed response amplitudes. They try different architectures and find that several architectures show similar performance overall. Of course the output of any of these circuits need not be directly equal to the turning rate of the animal. Am I correct in understanding that the model is that a single gain parameter should relate all of the turning rates under different conditions to the output of the circuit, and that this single gain is fixed by normalizing the positive two-point glider? Also, in*
Figure 3*, is the “equalized” model the same as the front-end nonlinearity model in*
Figure 1*? In what sense is it “equalized”?*

In response to the first question, the answer is yes: we are assuming that the behavior is proportional to the model output. This is in line with typical models of the optomotor response, which were first modeled as proportional to the HRC output.

In response to the second question, the equalized model is a specific front-end nonlinearity that transforms the natural input distribution into a uniform distribution. This is in contrast to the binarizing nonlinearity, which turns the natural input distribution into a binary distribution. We have clarified this point when we introduce the front-end nonlinearity models. Furthermore, the new paper architecture avoids the specific issue that the referee is referring to.

*7) In the subsection “Improving motion estimation by accounting for natural light-dark asymmetries”, the authors describe results that show that accounting for bright-dark asymmetry improved the results of all of the mechanisms that got the sign of glider responses right. While the text has a technical discussion how various combinations of signals work, I did not understand from the text the conceptual reason why the bright-dark separation helps. At some broad level it is not surprising that adapting to the natural statistics helps with the detection of signals, and indeed it seems here that all the mechanisms are helped by incorporating bright-dark asymmetries. Is there a deeper insight here? Or maybe the point is that the authors are simply making a prediction that the higher order motion detection circuits in flies will be discovered to segregate ON and OFF pathways and then recombine them, independently of which detailed mechanism and nonlinearities are being used? Now in*
Figure 4*, most of the quadrant models seem to do worse than HRC, and the (*— —*) quadrant seems not to be significantly better. And all of these models have a very low correlation with velocity. In view of this, could the authors please clarify why they are saying that accounting for bright-dark asymmetry in their model improves things?*

We apologize that this section was unclear. In the new manuscript, the presentation of this material is very different from the previous draft. For example, the results that were compressed into Figure 4 are now more distributed throughout the narrative, which affords us the opportunity to discuss how each model incorporates light and dark information differently than prior models. We hope that our presentation is substantially clearer this time around.

When we say that accounting for light-dark asymmetry is helpful in the weighted 4- quadrant model, we meant to emphasize that the four quadrants can be combined in a better manner than the HRC predicts. We did not mean to say that the benefits of ON/OFF processing could be achieved by the isolated quadrants.

In terms of whether there is a deeper insight, we now include some discussion of the many different ways that the models utilized light-dark asymmetries. For example, we discuss how simple ON/OFF misses certain useful cues that are captured by other models. We also try to emphasize a broader set of conceptual points that go beyond the asymmetry between light and dark.

*8) In the subsection “Improving motion estimation by reducing kurtosis\ the authors explain that their front end nonlinearities improved motion estimation by reducing the kurtosis in the inputs, even though they did not help with predicting glider responses. Would it help to have these nonlinearities along with the correlation detectors and the ON-OFF segregation discussed in previous sections*?

As discussed previously (see response to comment 4), the frontend nonlinearities that we use will eliminate the benefit of subsequent ON/OFF processing. It is less obvious how effectively frontend nonlinearities could be combined with the non- multiplicative nonlinearity model. However, the situation will certainly be much more complicated, because the signal and noise of the non-multiplicative nonlinearity model depend on more than the second and fourth-order statistics of the image ensemble. We point out in our discussion that several of these models could be combined in interesting ways, though combinations would change the optimal weightings. We consider pursuing this to be beyond the scope of the current work.

9) As written, the paper uses computational methods to specifically examine some possible mechanisms of motion estimation in flies in view of the inability of the traditional Reichardt model to explain how animals use higher order motion cues. I agree with the authors that the work has potentially broader significance beyond the specific example discussed here. But in order to engage that broader interest it would be really good if the authors would engage more substantially with the literature on higher order statistics in natural scenes and adaptation of circuit structure and perceptual phenomena to these statistics. At the moment the engagement is cursory. Some suggestions appear below. None of the papers mentioned below is specifically about motion estimation and the suggestion to engage a bit more with all this is not intended to detract from the novelty of the present work; rather it will likely help readers to find the work situated better against this well-known context.

We thank the reviewer for the suggestion to add broader context. We have modified the Introduction and Discussion sections substantially to situate the reader within the broader context of natural scene regularities and their influence on visual processing. Through these changes, we have included many of the references suggested.